# Compensatory evolution to DNA replication stress is robust to nutrient availability

Mariana Natalino [ID] & Marco Fumasoni [ID] [✉]

## Abstract

Evolutionary repair refers to the compensatory evolution that follows perturbations in cellular processes. While evolutionary trajectories are often reproducible, other studies suggest they are shaped by genotype-by-environment (GxE) interactions. Here, we test the predictability of evolutionary repair in response to DNA replication stress—a severe perturbation impairing the conserved mechanisms of DNA synthesis, resulting in genetic instability. We conducted high-throughput experimental evolution on *Saccharomyces cerevisiae* experiencing constitutive replication stress, grown under different glucose availability. We found that glucose levels impact the physiology and adaptation rate of replication stress mutants. However, the genetics of adaptation show remarkable robustness across environments. Recurrent mutations collectively recapitulated the fitness of evolved lines and are advantageous across macronutrient availability. We also identified a novel role of the mediator complex of RNA polymerase II in adaptation to replicative stress. Our results highlight the robustness and predictability of evolutionary repair mechanisms to DNA replication stress and provide new insights into the evolutionary aspects of genome stability, with potential implications for understanding cancer development.

**Keywords** Compensatory Evolution; DNA Replication Stress; Nutrients; Genome Maintenance; *S. cerevisiae*
**Subject Categories** DNA Replication, Recombination & Repair; Evolution & Ecology; Metabolism

## Introduction

Compensatory evolution is a process by which cells mitigate the negative fitness effects of persistent perturbations in cellular processes across generations. This adaptation occurs through spontaneously arising compensatory mutations anywhere in the genome (Wright and John, 1964; Wright, 1977, 1982) that partially or fully alleviate the negative fitness effects of perturbations (Moore et al, 2000). The successive accumulation of compensatory mutations over evolutionary timescales progressively repairs the cellular defects, ultimately restoring fitness. Perturbations can affect virtually any cellular process and may arise from genetic disruptions due to mitotic errors, the presence of selfish genetic elements, or the cytotoxic effects of external agents such as drugs (LaBar et al, 2020). Consequently, compensatory evolution, often referred to as "evolutionary repair", is relevant for the evolution of organisms in nature (Natalino and Fumasoni, 2023), as well as in the context of viral (Debray et al, 2022), bacterial (Yang et al, 2020), and parasitic infections (McCutchan et al, 2004), or during cancer's somatic evolution (Persi et al, 2021). Predicting the evolutionary trajectories that lead to evolutionary repair can shed light on fundamental principles underlying the evolution of cell biology and inform clinical treatments.

Understanding evolutionary outcomes, however, is challenging (Papp et al, 2011; Lässig et al, 2017; Wortel et al, 2021), largely due to the polygenic nature of traits (Fagny and Austerlitz, 2021) and their interaction with the environment (Boyer et al, 2021). Genotype-by-environment (GxE) interactions are well-documented. Several studies on *E. coli* have demonstrated how different environments influence fitness and epistatic interactions among adaptive mutations in the Lenski Long-Term Evolution Experiment (Ostrowski et al, 2005, 2008; Flynn et al, 2013; Hall et al, 2019). Adaptive mutations in viral genomes similarly exhibit variable fitness effects across different hosts (Lalić and Elena, 2013; Cervera et al, 2016). Furthermore, interactions between mutations in the Plasmodium falciparum dihydrofolate reductase gene have been shown to predict distinct patterns of resistance to antimalarial drugs (Ogbunugafor et al, 2016). However, the role of environmental factors in shaping evolution within the context of compensatory adaptation, when fitness defects primarily arise from intracellular perturbations, remains much less explored. Szamecz and colleagues examined the evolutionary trajectories of 180 haploid yeast gene deletions over 400 generations (Szamecz et al, 2014). They found that, while fitness recovery occurred in the environment where evolution took place, the evolved lines often showed no improvement over their ancestors in other environments. This suggests that compensatory mutations beneficial in one environment often fail to restore fitness in others. Similarly, Filteau and colleagues found that the carbon source impacted the evolutionary trajectories available to recover from a mutation in the *LAS17* gene, which mimics the causal mutation of Wiskott–Aldrich syndrome in humans (Filteau et al, 2015). On the other hand, a number of studies in yeast have reported a high level of parallelism in evolutionary trajectories, when conserved cellular processes such as chromosome cohesion (Hsieh et al, 2020), segregation (Pavani et al, 2021), and cell polarization (Laan et al, 2015) are perturbed under a stable nutrient-rich environment.

Gulbenkian Institute for Molecular Medicine (GIMM), Lisbon, Portugal. ✉E-mail: marco.fumasoni@gimm.pt

For example, compensatory evolution to constitutive replication stress, an intracellular perturbation that challenges the faithful replication of the genome, was reported to reproducibly depend on adaptive changes in DNA replication, chromosome segregation and cell cycle modules (Fumasoni and Murray, 2020, 2021). Replication stress is considered an early hallmark of cancer (Macheret and Halazonetis, 2015), deriving directly from oncogene activation, and sustaining genetic instability throughout cancer progression (Gaillard et al, 2015). Cancer cells experiencing DNA replication stress can divide in widely variable nutrient environments depending on the site of the primary lesion, its metastatic derivatives, and the level of tumor vascularization (Torrence and Manning, 2018). Several studies in recent years have linked nutrient availability and metabolic processes orchestrated by the Target of Rapamycin (TOR) pathway to genome maintenance processes such as DNA replication (Lamm et al, 2019; Silvera et al, 2017; Schonbrun et al, 2013) and repair (Shen et al, 2007; Shimada et al, 2013; Ferrari et al, 2017). Here, we leverage a well-characterized system to induce and monitor adaptation to constitutive DNA replication stress in *Saccharomyces cerevisiae* budding yeast to explore the impact of different physiologically relevant levels of nutrients on evolutionary repair.

We evolved 96 parallel populations of budding yeast, organized into 12 replicate lines, across four conditions of glucose availability (from starvation to abundance) with or without replication stress. Glucose is the most abundant monosaccharide in nature and represents the preferred source of energy for most cells. Constitutive replication stress was induced by deleting *CTF4*, a non-essential replication fork component responsible for coordinating helicase progression with other fork-associated activities (Villa et al, 2016; Gambus et al, 2009; Samora et al, 2016; Yuan et al, 2019). Glucose availability significantly impacted the growth rate, cell cycle progression, and cell size of replication stress mutants (*ctf4Δ*). Notably, glucose starvation restored some cell cycle traits to WT levels, and improved the competitive fitness of *ctf4Δ* mutants. Unlike WT controls, replication stress mutants evolved faster under higher glucose concentrations. Nevertheless, whole-genome sequencing of evolved *ctf4Δ* populations revealed high genetic robustness and parallelism across glucose conditions. Importantly, we identified a novel module involved in adaptation to replication stress through transcriptional regulation. Overall, our findings support the predictability and robustness of compensatory evolution to constitutive replication stress and challenge the idea that environmental constraints inevitably lead to distinct evolutionary outcomes. These findings have significant implications for predicting evolutionary trajectories during evolutionary repair, impacting our understanding of the evolution of cellular processes and of medical conditions such as cancer progression.

## Results

### Glucose availability impacts cell physiology and fitness in the presence of DNA replication stress

The impact of nutrient availability on yeast cell cycle and growth dynamics has been well-documented (Talavera et al, 2024; Brauer et al, 2008; Johnston et al, 1977; Beck and von Meyenburg, 1968; Carter and Jagadish, 1978; Slater et al, 1977; Johnston et al, 1980).

Reduced nutrient availability typically decreases population growth rates and prolongs the G1 phase, due to the checkpoint controlling entry into the S phase (Johnston et al, 1977; Alberghina et al, 1998; Turner et al, 2012). However, it is still unclear whether these effects are similarly observed in cells experiencing significant intracellular defects, or if they are overshadowed by the cell cycle disruptions these defects cause. In the absence of Ctf4, cells exhibit multiple defects commonly associated with DNA replication stress, such as single-stranded DNA gaps and altered replication forks (Fumasoni et al, 2015), leading to basal cell cycle checkpoint activation (Poli et al, 2012). These defects result in severe and persistent growth impairments, cell cycle delays, elevated nucleotide pools and chromosome instability (Miles and Formosa, 1992; Kouprina et al, 1992; Poli et al, 2012), making *ctf4Δ* mutants an ideal model for studying the cellular consequences of general and constitutive replication stress over evolutionary time.

To investigate how glucose availability affects the growth and physiology of *S. cerevisiae* experiencing constitutive replication stress, we grew WT and *ctf4Δ* cells in varying glucose concentrations to induce distinct physiological states. Low glucose levels (0.25% and 0.5%) induce caloric restriction and ultimately glucose starvation (Lin et al, 2000; Smith et al, 2009). These conditions elicit increased respiration (Lin et al, 2002), sirtuins expression (Guarente, 2013), autophagy (Bagherniya et al, 2018), DNA repair (Heydari et al, 2007), and reduced recombination at the ribosomal DNA locus (Riesen and Morgan, 2009) ultimately extending lifespan in several organisms (Kapahi et al, 2017). In contrast, standard laboratory conditions typically use 2% glucose, promoting a rapid proliferation environment to which strains have been adapted since laboratory domestication (Lindegren, 1949). Finally, elevated glucose concentrations (such as 8%) result in higher ethanol production (Lin et al, 2012) and reactive oxygen species (ROS) levels (Maslanka et al, 2017).

As the initial glucose concentration decreased, WT cells showed progressively lower growth rates (Fig. 1A), longer doubling times (Fig. EV1A), and reduced maximum optical density (Fig. EV1B). *ctf4Δ* cells exhibited a similar, albeit smaller, decline (Fig. 1A, right), despite their growth rates in standard conditions (2% glucose) being already significantly lower than the minimum observed in starved WT cells. In WT cells, nutrient-dependent reductions in growth rates are linked to changes in cell cycle progression and cell size (Johnston et al, 1979; Turner et al, 2012). We asked how these growth differences affect cell physiology and cell cycle progression in cells experiencing DNA replication stress. Previous studies showed that *ctf4Δ* cells present an altered cell cycle profile, characterized by a prominent G2 arrest and an almost absent G1 phase (Tanaka et al, 2009; Fumasoni et al, 2015). The extended G2 phase results from the constitutive activation of the DNA damage checkpoint, which delays anaphase to allow for DNA lesion repair (Fumasoni and Murray, 2020; Tanaka et al, 2009; Poli et al, 2012). The shortened G1 phase is usually attributed to the large size of cells arrested in mitosis, leading to premature satisfaction of the size checkpoint during the subsequent G1 phase (Johnston et al, 1977). Similar to WT cells (Fig. 1B, left), decreasing glucose availability in *ctf4Δ* cells led to changes in cell cycle profile (Fig. 1B, right), increasing the length of G1 (Fig. 1C), reducing that of G2 (Fig. EV1C), and resulting in smaller cell sizes (Fig. 1D). Notably, glucose starvation (0.25%) alleviated these physiological defects in replication stress mutants, leading to a G1 length and a

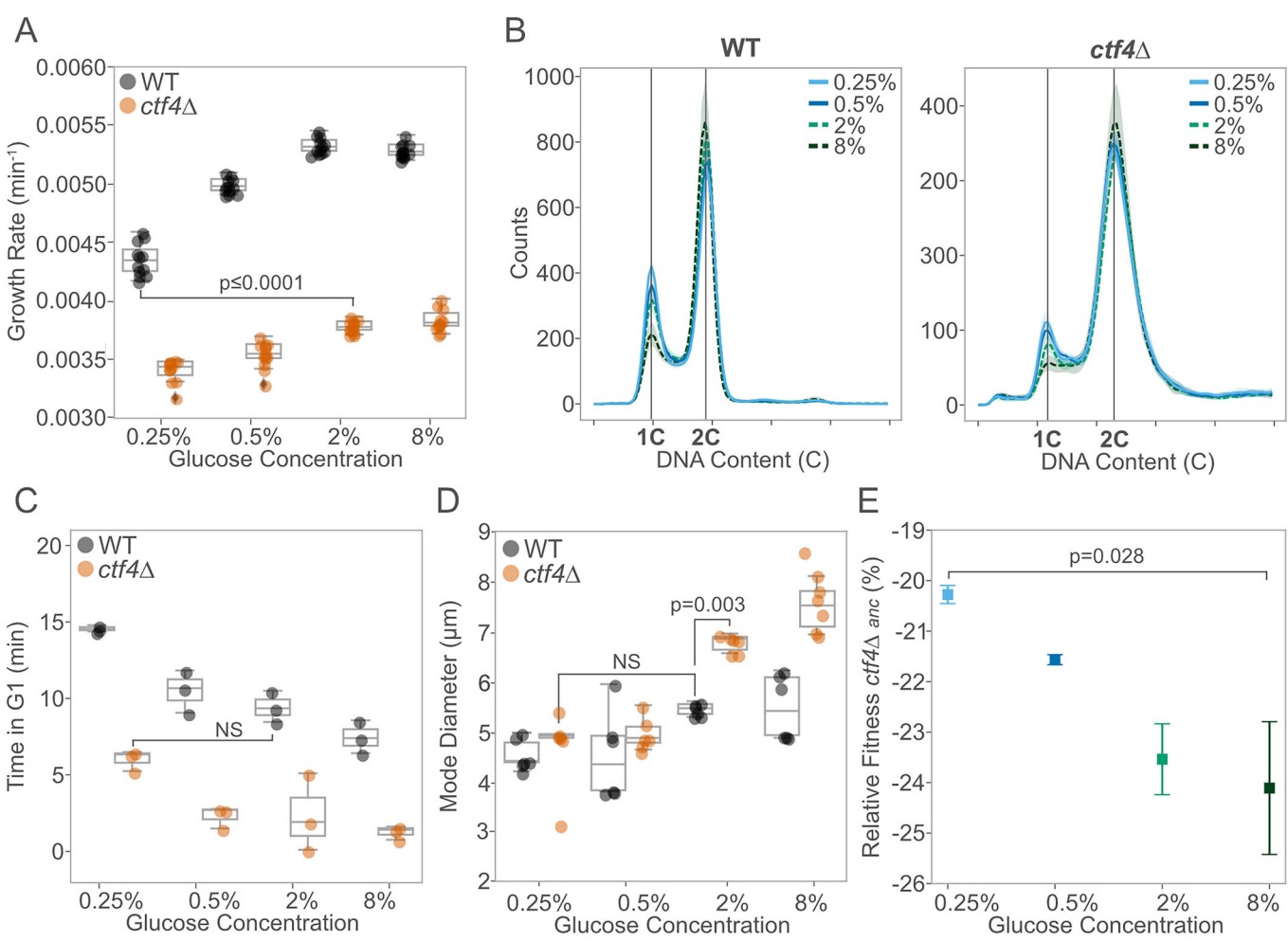

**Figure 1. Glucose concentration impacts cell physiology in the presence of DNA replication stress.**

(A) Population growth rates (min$^{-1}$) of ancestral WT (black) and *ctf4Δ* mutant (orange) across different glucose concentrations ($n = 12$ biological replicates, Mann–Whitney $U$ test with BH correction) (WT 0.25% vs *ctf4Δ* 2%, $P$ value $= 6.47 \times 10^{-5}$). (B) Cell cycle profiles of ancestral WT (left) and *ctf4Δ* (right) across glucose concentrations. Colors represent different glucose concentrations: blue refers to glucose starvation (light blue for 0.25%, dark blue for 0.5%), while green refers to glucose abundance (light green for 2%, dark green for 8%). Bold lines indicate mean profiles, and shaded areas represent standard deviation (SD) ($n = 3$ biological replicates). 1C and 2C indicate DNA content in G1 and G2/M phases, respectively. (C) Time spent (minutes) in G1 phase for ancestral WT and *ctf4Δ*, across different glucose concentrations, estimated from DNA content and doubling times (see "Methods", $n = 3$ biological replicates, ANOVA Tukey's HSD). (D) Mode cell diameter of ancestral WT and *ctf4Δ* across different glucose concentrations ($n = 6$ biological replicates, ANOVA Tukey's HSD). (E) Mean relative fitness of ancestral *ctf4Δ* relative to reference WT across different glucose concentrations. Colors represent glucose concentration. Error bars represent SD ($n = 4$ biological replicates, Mann–Whitney $U$ test). Box plots in (A, C, D) represent the median (center line), 25th and 75th percentiles (lower and upper bounds of the box), and whiskers extending to the smallest and largest values within 1.5× the interquartile range (IQR) from the lower and upper quartiles, respectively. Data points beyond whiskers are shown as outliers. Detailed statistical analysis and underlying data for this figure are provided in Source Data. Source data are available online for this figure.

cell size indistinguishable from those of WT cells under standard glucose abundance ($6.33 \pm 0.7$ min vs. $7.40 \pm 1.1$ min and $4.61 \pm 1.5$ μm vs. $4.68 \pm 0.4$ μm, respectively).

We assessed the impact of these cell cycle changes on cellular fitness using competition assays, in which two genetically distinct lineages compete for nutrients in the same media over multiple generations (see "Methods" for details). As previously reported, *ctf4Δ* cells exhibited severe fitness defects (Fumasoni and Murray, 2020, 2021). However, their relative fitness improved compared to the WT reference as the initial glucose levels in the competition media decreased (Fig. 1E). In 0.25% glucose, *ctf4Δ* cells showed approximately a 4% fitness advantage over those competed in 8% glucose (Fig. 1E). These results highlight how changes in glucose

availability impact the population growth and cell physiology of *ctf4Δ* cells, mitigating the fitness defects introduced by constitutive DNA replication stress.

## Replication stress mutants display higher adaptation rates and fitness gains under glucose abundance

Can glucose availability influence the evolutionary adaptation to DNA replication stress? The glucose-dependent effects on fitness and cell physiology reported above alter the selective pressures experienced by populations. Over multiple generations, these could influence the cells' ability to evolutionarily repair DNA replication stress. To test this hypothesis, we evolved 12 parallel populations

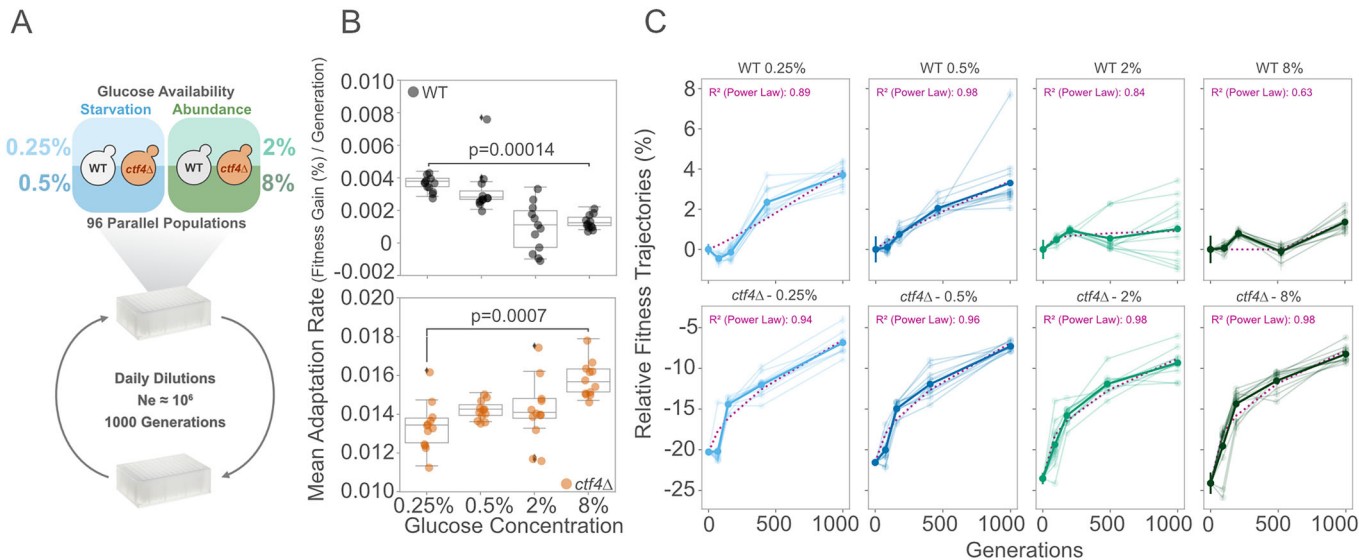

**Figure 2. Glucose availability impacts the dynamics of fitness recovery.**

(A) Schematic of experimental layout. 48 isogenic clones of ancestral *ctf4Δ* and 48 WT clones were inoculated in either glucose starvation (0.25% and 0.5%) or abundance (2% and 8%) on deep 96-well plates. Clones were grown to saturation and diluted daily until reaching 1000 generations. The bottleneck was adjusted to maintain Ne within the same order of magnitude throughout the experiment. (B) Mean adaptation rate (% fitness gain per generation) for ancestral WT (top) and *ctf4Δ* mutant (bottom) across glucose concentrations. Adaptation rate was calculated as the fitness difference between the evolved population and their ancestor (Δ), divided by the total number of generations elapsed (1000). Box plots represent the median (center line), 25th and 75th percentiles (lower and upper bounds of the box), and whiskers extending to the smallest and largest values within 1.5× the IQR from the lower and upper quartiles, respectively. Data points beyond whiskers are shown as outliers ($n = 3$ replicas per population, Mann–Whitney $U$ test with BH correction). (C) Fitness trajectories of WT (upper panel) and *ctf4Δ* (lower panel) populations evolved across varying glucose concentrations. Individual population data are shown as shaded lines, while mean fitness values are displayed as solid lines, with error bars representing standard deviations (SD) ($n = 3$ replicas per population). Individual trajectories were modeled using a power law function (see "Methods"). The dashed purple line depicts the power law fit of the mean trajectory. Line colors correspond to glucose concentrations as follows: light blue (0.25%), dark blue (0.5%), light green (2%), and dark green (8%). Detailed statistical analyses, underlying data, and estimated parameters are provided in Source Data. Source data are available online for this figure.

each of haploid *ctf4Δ* mutants and WT controls over 1000 generations through daily serial dilutions in rich media (YP) under glucose starvation, or glucose abundance (Fig. 2A). Importantly, we adjusted the number of cells passaged and the daily number of generations to maintain the effective population size (Ne) at the same order of magnitude throughout the experiment (Fig. EV2A). Maintaining a stable Ne is crucial, as large differences can influence evolutionary dynamics by affecting genetic drift and the population's access to beneficial mutations, potentially confounding the impact of nutrient levels in adaptation (Wright, 1931; Crow and Kimura, 1970; Lynch et al, 1995; Silander et al, 2007; Desai et al, 2007; Van den Bergh et al, 2018; Schenk et al, 2022).

By generation 1000, both WT and *ctf4Δ* evolved lines achieved, on average, slightly higher fitness in low glucose compared to high glucose conditions (Fig. EV2B). However, in *ctf4Δ* lines, the proportion of the initial fitness defect recovered remained constant across conditions (Fig. EV2C). As a result, *ctf4Δ* lines displayed an opposite trend to WT, with increasing absolute fitness gains throughout the experiment as glucose concentration rose (Fig. EV2B vs S2D). The different absolute fitness gains over the same number of generations highlight distinct mean adaptation rates (Fig. 2B). These differences are evident when examining the evolutionary dynamics of the evolved lines over time (Fig. 2C). In addition, we approximated the fitness trajectories using the power law function (Fig. 2C, dashed purple lines), previously proposed to describe long-term evolutionary dynamics in constant environments (Wiser et al, 2013). The parameter b in this formula

determines the curve's steepness, and can be used to quantify the global adaptation rate over generations (Fig. EV2E). Collectively, these analyses demonstrate that, unlike WT cells, *ctf4Δ* lines adapt faster in the presence of high glucose. This evidence aligns with the declining adaptability observed in other studies (Moore et al, 2000; Kryazhimskiy et al, 2014; Couce and Tenaillon, 2015), where low-fitness strains consistently adapt faster than their more fit counterparts (Fig. EV2F).

Overall, these results demonstrate that cells can recover from fitness defects caused by constitutive DNA replication stress regardless of the glucose environment. However, adaptation rates under DNA replication stress exhibit opposing trends compared to WT cells, with faster adaptation yielding greater fitness gains in higher glucose conditions.

## Genotype, and not the environment, shapes the mutational profile in evolved populations

During experimental evolution, populations accumulate various types of mutations: adaptive mutations that confer a fitness benefit and neutral or slightly deleterious mutations that spread due to genetic drift or hitchhiking alongside adaptive mutations. Comparative analysis of the identity and frequency of these mutations can shed light on the genetic basis of the evolutionary processes across different environments. We whole-genome sequenced the final evolved populations as well as the ancestral clones (see "Methods"), to examine whether genotype, environment, or their

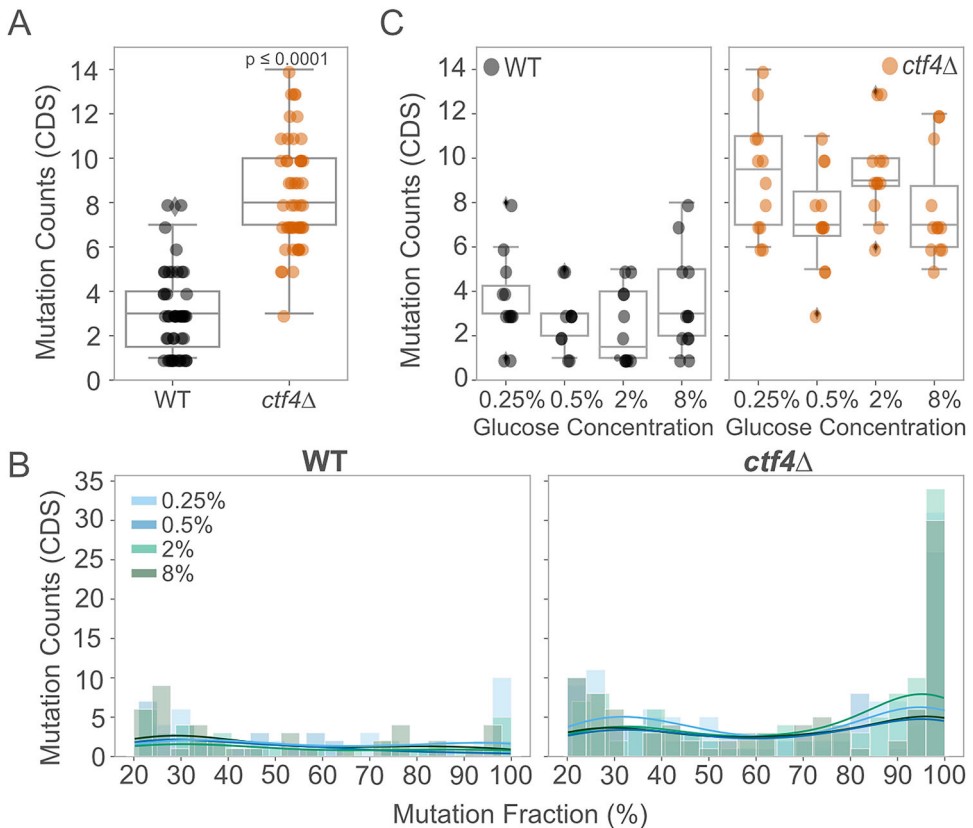

**Figure 3.  The mutational profile is mainly influenced by the genotype.**

(A) Total detected mutations in CDS per evolved WT (black) and *ctf4Δ* (orange) populations, at generation 1000 ($n = 12$ for individual populations, Mann–Whitney *U* test was used to compare mutational counts (CDS)) (WT vs *ctf4Δ*, *P* value $= 1.38 \times 10^{-14}$). (B) Distribution of mutated read fractions across glucose concentrations for WT (left) and *ctf4Δ* (right) evolved populations, used as a proxy for clonality. Mutation fraction (%) was calculated as the fraction of reads from whole-population sequencing that contained a particular mutation in CDS. Colors represent glucose concentrations: light blue (0.25%), dark blue (0.5%), light green (2%), dark green (8%). Histograms are overlaid with a kernel density estimate (KDE, colored lines) to illustrate frequency distributions. Kolmogorov–Smirnov (KS) test was used to compare read fraction distributions between glucose concentrations and genotypes. (C) Total mutations detected in CDS across glucose concentrations for WT (left) and *ctf4Δ* (right) populations at generation 1000 ($n = 12$ for individual populations). Kruskal–Wallis test was performed to assess the effect of glucose concentration on mutation counts for WT and *ctf4Δ*. Box plots in (A, C) represent the median (center line), 25th and 75th percentiles (lower and upper bounds of the box), and whiskers extending to the smallest and largest values within 1.5× the IQR from the lower and upper quartiles, respectively. Data points beyond whiskers are shown as outliers. Detailed statistical analysis and underlying data for this figure are provided in Source Data. Source data are available online for this figure.

interaction (GxE) affected the mutational landscape of the evolved lines. Evolved *ctf4Δ* lines exhibited approximately three to four times more mutations in coding regions (CDS) than WT lines (Fig. 3A). This finding suggests that replication stress mutants either accumulate mutations at a higher frequency than WT or that a larger fraction of mutations were beneficial and thus were retained until generation 1000. To distinguish between these scenarios, we examined the frequency of synonymous mutations, which are less likely to be subject to selection during evolution. We found no significant differences in the numbers of synonymous mutations detected in evolved populations in WT and *ctf4Δ* populations (Fig. EV3A). These results support the hypothesis that replication stress in *ctf4Δ* lines favors the retention of beneficial mutations, rather than simply increasing the overall mutation rate. We detected a higher median mutation fraction (Fig. EV3B) and significantly different distribution ($P \leq 0.0001$) in *ctf4Δ* compared to WT evolved populations, with a strong skew towards higher-frequency mutations (Fig. 3B). This indicates a higher level of clonality in *ctf4Δ* populations, consistent with a greater frequency

of adaptive mutations reaching fixation. Despite significantly affecting cell physiology and fitness dynamics, glucose concentrations did not influence mutation counts (Fig. 3C) or population clonality (Fig. EV3C) in either WT or *ctf4Δ* evolved lines.

We then asked whether glucose concentrations influenced the occurrence of putative adaptive mutations within each genotype. We identified candidate adaptive mutations by determining which genes were mutated more frequently than expected by chance in independent populations (see "Methods" for details). In WT, 13 mutations across 48 evolved populations exhibited signs of selection, with 7 of these mutations occurring exclusively under glucose starvation (Fig. 4A, upper panel). Only *MSS11*, encoding a transcription factor involved in regulating filamentous growth, was found to be mutated across all glucose conditions (Fig. 4A, upper panel). All positively selected GO terms, including genes involved in nutrient sensing and cell growth—such as those in the Mitogen-Activated Protein Kinase (MAPK) signaling pathway—were exclusively found in populations evolved under low glucose, but not in other glucose concentrations (Fig. 4A, bottom panel). In *ctf4Δ* populations, we similarly identified more putatively adaptive genes

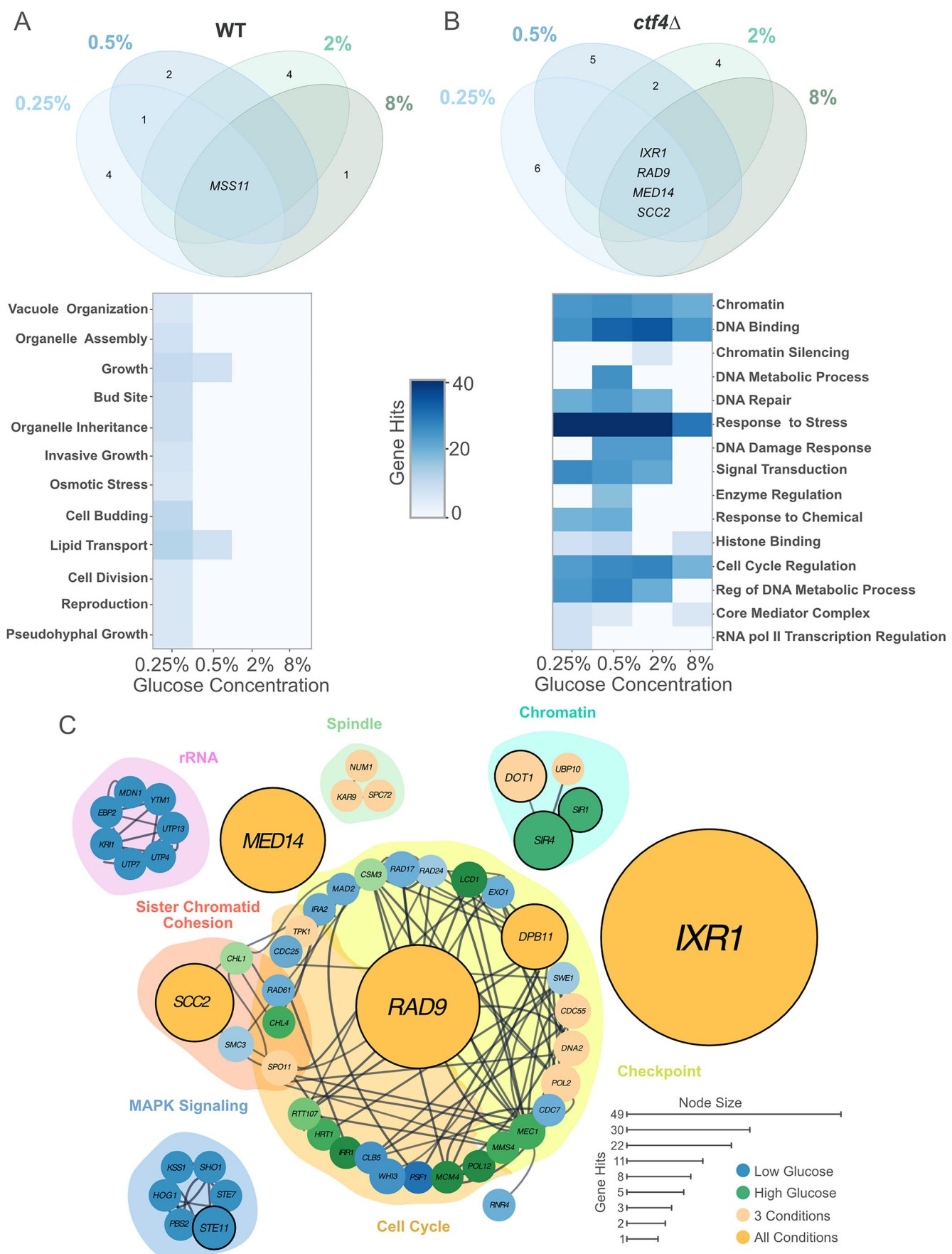

**Figure 4.   Environment impacts the genetic basis of adaptation of WT but not the replication stress mutant.**

(A) Venn diagram of putative adaptive genes mutated in evolved WT populations across glucose concentrations (upper panel). Colors represent glucose concentrations: light blue (0.25%), dark blue (0.5%), light green (2%), and dark green (8%). Numbers represent counts of putative adaptive genes (excluding zero counts). Gene names in the center are shared across conditions. GO terms enriched across glucose concentrations in WT (bottom panel). Heatmap illustrates the total number of gene hits for significant GO terms. Fisher's Exact Test was used to assess the significance between pairwise glucose conditions. (B) Venn diagram of putative adaptive genes mutated in evolved *ctf4Δ* populations across glucose concentrations (upper panel) and corresponding GO term enrichment heatmap (bottom panel). (C) Simplified interaction network (curated in Cytoscape, (Shannon et al, 2003)) of mutations detected in *ctf4Δ* evolved populations. Dark gray lines represent known genetic and physical interactions from the literature (STRING database, (Szklarczyk et al, 2023)). Node diameter corresponds to the total number of mutations (hits) detected in the coding region of each gene. Nodes are color-coded: blue for mutations in low glucose (0.25% and 0.5%), green for high glucose (2% and 8%), light orange for both high and low glucose, and dark orange for all conditions. Nodes with a bold outline indicate putative adaptive genes in at least one condition. Shaded clusters represent Gene Ontology (GO) term enrichment for biological processes obtained using STRING database. Detailed statistical analysis and underlying data for this figure are provided in Source Data. Source data are available online for this figure.

uniquely mutated under glucose starvation (0.25% and 0.5%) compared to abundance (2% and 8%) (11/20 vs 4/20, Fig. 4B, upper panel). However, GO term enrichment analysis revealed a different pattern compared to WT lines. While only a few modules involved in rRNA regulation and MAPK signaling were exclusively selected under glucose starvation, most selected GO terms —such as those involved in cell cycle, transcription regulation and genome maintenance—were found enriched across all glucose conditions (Fig. 4B, bottom panel, and 4C).

Four genes, here defined "core genetic adaptation", were consistently mutated (three SNPs/INDELS and one gene amplification) in response to replication stress, regardless of glucose availability (Fig. 4B and Source Data). Three of these genes had been previously identified to affect DNA replication (*IXR1*), the DNA damage checkpoint (*RAD9*), and chromosome cohesion (*SCC2*) in response to constitutive DNA replication stress (Fumasoni and Murray, 2020, 2021). Importantly, we identified a novel fourth module, represented by mutations in *MED14* (also known as *RGR1*), a gene implicated in transcriptional regulation (Sakai et al, 1988; Warfield et al, 2022). Overall, our results indicate that the mutational profile of replication stress mutants is primarily shaped by the intracellular stress imposed by the *CTF4* deletion. The genotype, rather than the glucose environment, played a dominant role in determining the mutational profiles. In addition, while glucose availability influenced adaptive mutations in WT, this effect was negligible under DNA replication stress, where most selected modules were consistently targeted across all conditions.

## A single amino acid substitution in mediator complex subunit 14 alleviates replication stress defects

We observed a remarkable level of parallelism targeting the Med14 subunit of the transcription mediator complex in *ctf4Δ* evolved populations. A single amino acid substitution of histidine with proline in the C-terminal of the protein (*med14*-H919P) was detected at high frequency in 21 out of 48 independently evolved populations (Fig. 5A). The reconstruction of this mutation in the *ctf4Δ* ancestor demonstrated causality by leading to approximately a 4% fitness advantage (Fig. 5B).

The mediator complex is essential for the initiation of eukaryotic gene transcription (Thompson and Young, 1995; Myers et al, 1998) and is highly conserved across eukaryotes (Bourbon, 2008). It interacts with the C-terminal domain of RNA polymerase II (RNA pol II) and transcription factors, facilitating transcription initiation (Kim et al, 1994). In yeast, the mediator complex comprises 25 subunits, organized into three distinct modules: kinase, core

mediator, and the tail. The tail module plays a role in gene-specific transcription regulation by interacting with transcription activators (Soutourina, 2018). Recent structural studies have shown the dimerization of yeast's pre-initiation complex upon binding to divergent promoters (Fig. 5C). Data ref: (Gorbea Colón et al, 2023, Protein Data Bank 7UIO). The C-terminal domain of Med14, known as Tail Interaction Domain (TID, residues 705–1082), links the tail module to the rest of the mediator complex. The adaptive mutation *med14*-H919P is located within the TID, specifically in an alpha-helix that connects Med14 with the core mediator complex (Fig. 5C, left). We simulated Med14's amino acid substitution using the Rotamer tool in ChimeraX (Meng et al, 2023), which predicted steric clashes with adjacent amino acids, Asp 917 and Tyr 918 (Fig. 5C, right). In addition, the substitution of histidine with proline is predicted to destabilize the alpha-helix due to proline's inability to form hydrogen bonds with neighboring residues, suggesting a possible disruption of Med14's structural integrity. To explore possible functional consequences of this mutation, we analyzed the publicly available RNA-seq data resulting from degron-mediated removal of the Med14 TID, Data ref: (Warfield et al, 2022). Our analysis identified 90 significantly downregulated genes, with an enrichment in energy and nucleotide metabolism pathways (Dataset EV1). The list included a subunit of the ribonucleotide reductase (Rnr1), the enzyme responsible for producing the deoxyribonucleotides (dNTPs) required for DNA replication (Fig. EV4A). The mediator complex was also recently shown to recruit the cohesin loader Scc2 to transcribed genes through an interaction with the Med14 subunit (Mattingly et al, 2022). Based on this, we propose three non-mutually exclusive hypotheses to account for the beneficial roles of *med14*-H919P in alleviating DNA replication stress: (i) By alleviating transcription genome-wide *med14*-H919P could lower the frequency of collisions of the transcription machinery with replication forks. (ii) Alternatively, *med14*-H919P could slow and stabilize replication forks by lowering the dNTP pools through decreased RNR1 expression. (iii) Finally, *med14*-H919P could enhance the recruitment of Scc2, whose excess is beneficial in *ctf4Δ* cells (Fumasoni and Murray, 2020). To distinguish between these hypotheses, we examined the genetic interactions of *med14*-H919P with mutant alleles of *DUN1*, involved in dNTP regulation (Zhao and Rothstein, 2002), Ribonuclease H1 and H2 (*RNH1* and *RNH201*, respectively), and *SEN1*, implicated in RNA metabolism (Appanah et al, 2020; Aguilera and García-Muse, 2012), and *CHL1*, required for cohesion establishment during DNA replication (Skibbens, 2004). Interestingly, *med14*-H919P showed slightly positive genetic interaction with all the alleles

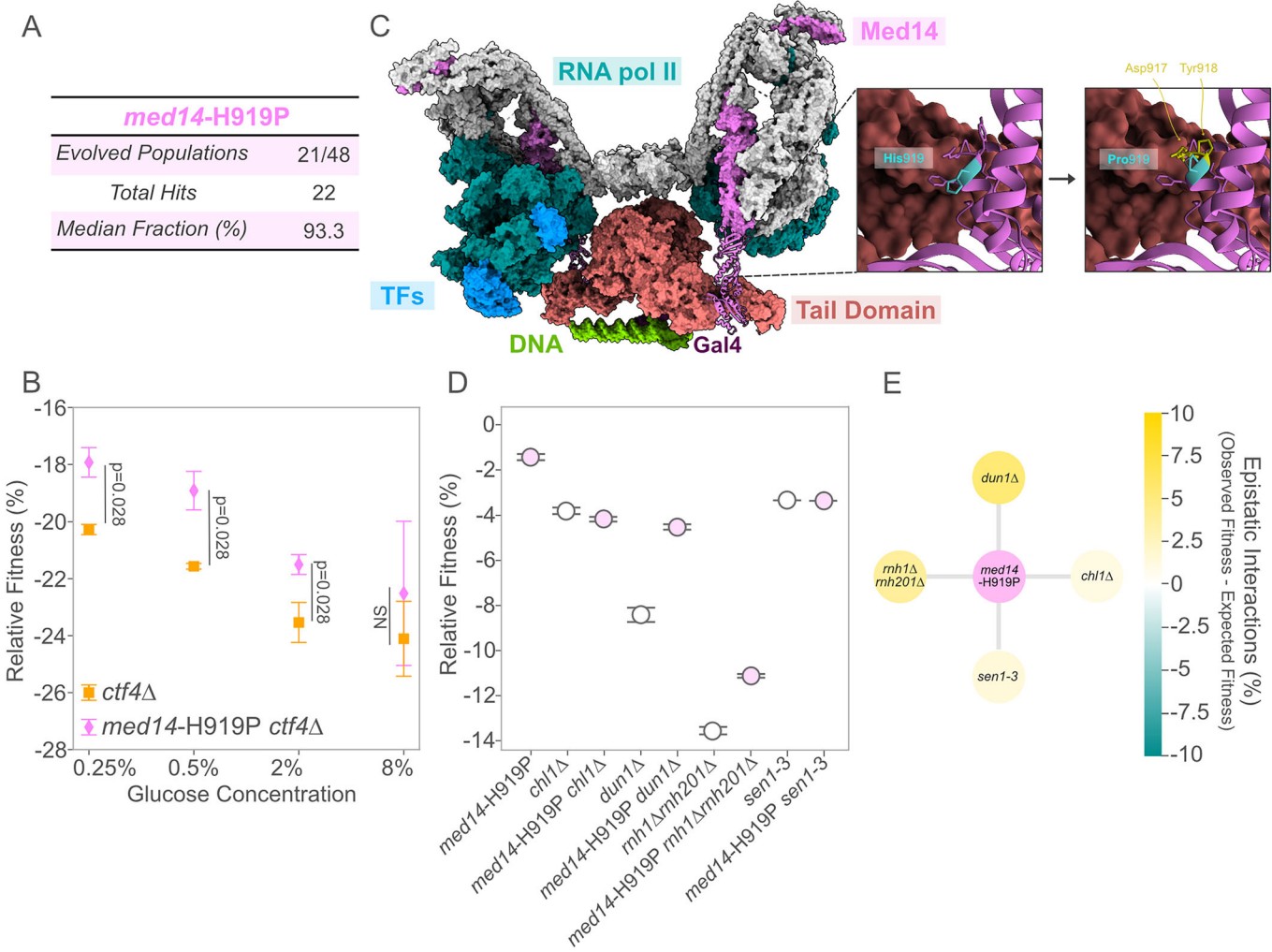

**Figure 5. Adaptive fitness and structural insights of Med14 mutation in replication stress mutants.**

(A) Schematic representation of the prevalence of the *med14*-H919P point mutation in evolved *ctf4Δ* populations across glucose conditions. (B) Mean relative fitness of *ctf4Δ* ancestor (orange) and *ctf4Δ* carrying reconstructed mutation *med14*-H919P (pink). Error bars represent SD ($n = 3$ biological replicates, Mann–Whitney $U$ test with BH correction). (C) Composite model of the transcription pre-initiation complex of RNA pol II with mediator complex forming a dimer to act on a distal promoter (Gal4-activated, PDB: 7UIO). Tail components, RNA pol II, transcription factors (TFs), DNA, and regulatory Gal4 are color-coded. Med14 is highlighted in pink, with secondary structures shown for the C-terminal domain (705–1082 aa). Right panel: top view of HIS 919 (blue) and surrounding amino acids, with a simulation of the His to Pro substitution at site 919. Structural clashes were identified using ChimeraX (affected residues in yellow). (D) Mean relative fitness of mutants in chromatin cohesion (*chl1Δ*), nucleotide production (*dun1Δ*), and transcription replication conflicts (*rnh1Δ rnh201Δ* and *sen1-3*) alone (white) or in combination with *med14*-H919P mutation (light pink). Error bars represent SD ($n = 4$ biological replicates). (E) Genetic interaction network centered on *med14*-H919P. Node color represents the sign and strength of epistasis. Epistasis was calculated as the difference between observed and expected fitness (additive model). Detailed statistical analysis and underlying data for this figure are provided in Source Data. Source data are available online for this figure.

(Fig. 5D,E). Positive epistasis with both *rnh1Δrnh201Δ* and *sen1-3*, which exacerbate replication-transcription conflicts by failing to process R-loops, supports a role of *med14*-H919P in alleviating these conflicts by lowering transcription. The positive genetic interaction with *dun1Δ* which leads to a decrease in dNTP pools, argues instead against a role for *med14*-H919P in further lowering dNTPs. Finally, the positive interaction manifested with *chl1Δ* is compatible with a role of med14-H919P in facilitating cohesion establishment. Overall, these results show how an amino acid substitution in the Med14 subunit of the mediator complex, putatively affecting transcription, is strongly selected, and advantageous, in the presence of constitutive DNA replication stress.

## Core genetic adaptation to replication stress is robust to nutrient availability

The signs of positive selection towards the four genes belonging to the core genetic adaptation do not necessarily imply that their fitness benefit does not depend on the glucose environment. To test this point, we engineered loss-of-function alleles mimicking the mutations affecting *IXR1* and *RAD9*, as well as the *SCC2* amplification and *med14*-H919P substitution, into the ancestral WT and *ctf4Δ* cells. We competed all reconstructed strains individually against a reference WT strain and assessed their relative fitness under the different glucose environments. In the WT

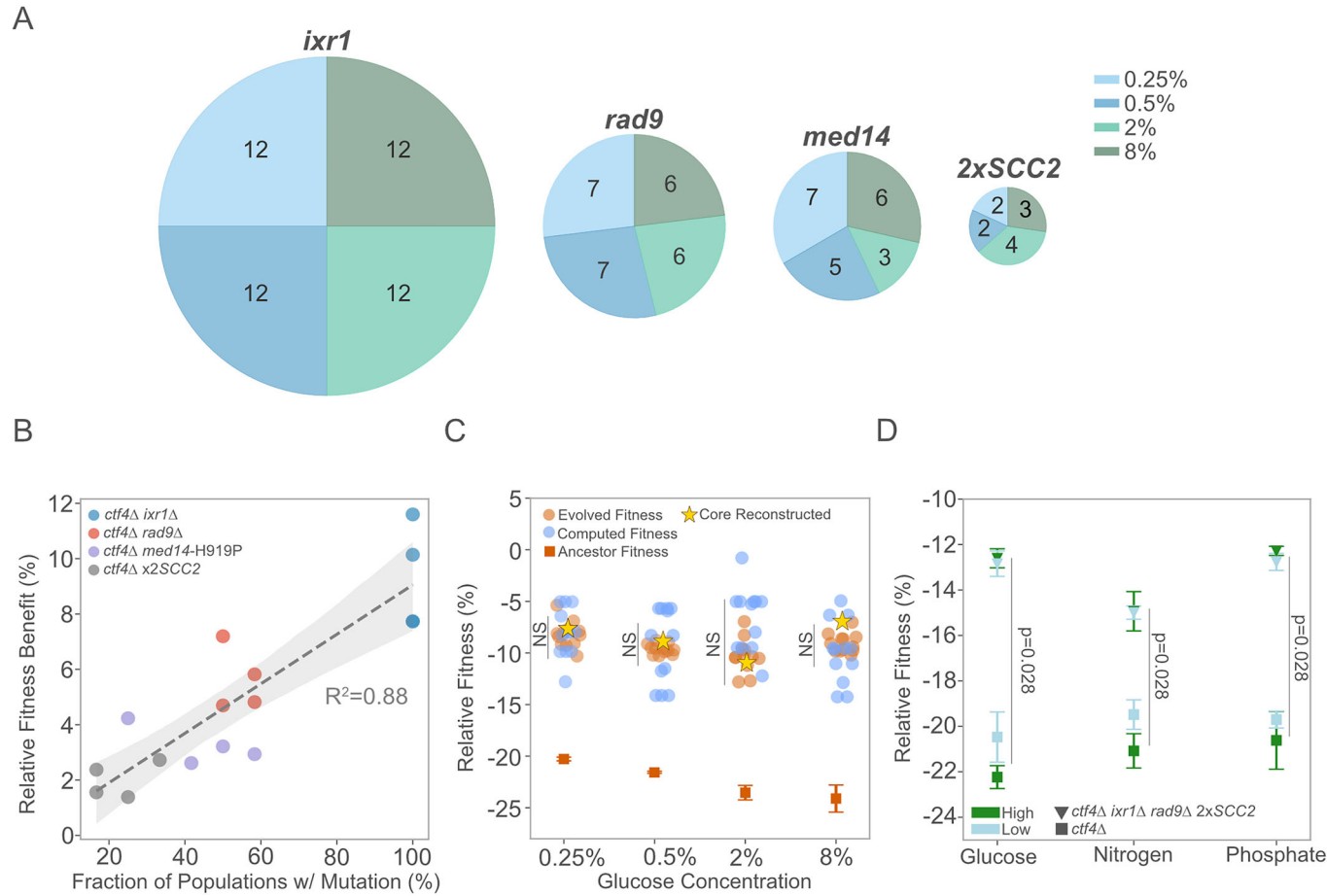

**Figure 6. Robustness in core genetics of adaptation to replication stress.**

(A) Distribution of core adaptive mutations in *IXR1*, *RAD9*, *MED14* and *SCC2* genes across glucose conditions. Pie chart sizes are proportional to the number of populations carrying each mutation in the gene. Numbers indicate the mutations detected across glucose concentrations, color-coded as light blue (0.25%), dark blue (0.5%), light green (2%), and dark green (8%). (B) Correlation between the fraction of populations in each glucose condition carrying a specific adaptive mutation and the fitness benefit of the mutation in a *ctf4Δ* background under the same glucose condition. Colors refer to genes: *ixr1Δ* (blue), *rad9Δ* (red), *med14*-H919P (purple), and *2xSCC2* (gray). A positive correlation ($R^2 = 0.88$) is observed, with the shaded area representing the 95% confidence interval of the linear regression. (C) Comparison of ancestral, reconstructed core mutations, evolved, and computed fitness for *ctf4Δ* lines. For each evolved population, if a reconstructed gene was found mutated, its fitness effect in the respective glucose concentration was added to the *ctf4Δ* ancestor to calculate computed fitness (blue). Relative fitness of evolved *ctf4Δ* populations (orange, $n = 3$ replicates per population), reconstructed quintuple (*ixr1Δ rad9Δ 2xSCC2 med14*-H919P *ctf4Δ*, yellow star, $n = 4$ biological replicates) and ancestral *ctf4Δ* (dark orange, $n = 4$ biological replicates) are shown. Error bars represent SD. *P* values indicate the likelihood that evolved fitness is higher than core reconstructed mutants (one-sided *t* test). (D) Mean relative fitness of *ctf4Δ* and the reconstructed strain (*ctf4Δ ixr1Δ rad9Δ 2xSCC2*). Error bars represent standard deviation (SD) ($n = 3$ biological replicates). Each point shows the mean fitness for a specific nutrient and concentration (low or high), with marker shapes distinguishing genotypes (squares for *ctf4Δ* and inverted triangles for the reconstructed strain) and colors indicating nutrient concentration levels (light blue for low, dark green for high). Mann–Whitney *U* test used to compare *ctf4Δ* and reconstructed in each nutrient. Detailed statistical analysis and underlying data for this figure are provided in Source Data. Source data are available online for this figure.

background, all mutations were nearly neutral, with only minimal deleterious or advantageous effects on fitness depending on glucose concentrations (Fig. EV5A). When introduced in a *ctf4Δ* background, all mutations led to fitness benefits compared to the ancestral cells. However, with the exception of *IXR1* deletion, competition assays performed in the different glucose conditions resulted in comparable fitness benefits with no statistical difference between glucose starvation or abundance (Fig. EV5B and Source Data EV5B). Similarly, we did not detect differences in the frequency of occurrence ($\chi^2$ tests) or average fractions (ANOVA test) achieved by the mutations in the populations evolved under different glucose environments (Figs. 6A and EV5C and Source Data). The presence of all mutations in the final evolved lines

correlated with their fitness benefits, suggesting how their selection in all glucose conditions was mostly dictated by their relative fitness benefits, rather than the environment (Fig. 6B).

Is the combined effect of the mutations belonging to this core genetic adaptation sufficient to recapitulate the final fitness of the evolved *ctf4Δ* populations? (Fig. EV2B). To address this question, we engineered a quintuple mutant carrying all alleles that mimic the core adaptive mutations in a *ctf4Δ* background. Across glucose concentrations, the fitness of this reconstructed strain closely approximated that of the final evolved populations. We also computed the cumulative fitness benefit of the core adaptive mutations present in each evolved population (Fig. 6C). For instance, in glucose 8%, population 11, where all four core

mutations had fixed, exhibited an observed and computed fitness of ~−8% and ~−5%, respectively. Under the same condition, the core reconstructed strain showed a fitness of ~−6%. Overall, these results highlight the robustness of core genetic adaptations to DNA replication stress. The four recurrently selected mutations provide significant fitness benefits across conditions, and their nearly additive effects largely account for the fitness gains observed in populations evolved over 1000 generations. A previous study in yeast showed how evolved lines that compensate for detrimental defects of gene deletions in standard laboratory conditions often failed to show fitness benefits compared to their ancestor when tested in other environments (Szamecz et al, 2014). We thus investigated the extent to which the core genetic adaptation to DNA replication stress was beneficial under alternative nutrient conditions. We reconstructed a quadruple mutant in the ancestral *ctf4*Δ background carrying a frequent combination of adaptive mutations (*ixr1*Δ, *rad9*Δ, and 2x*SCC2*). We then compared its relative fitness across conditions of starvation or abundance of the essential macronutrients' glucose, nitrogen, and phosphate. Across all conditions, the reconstructed strain showed a significant fitness recovery compared with ancestor *ctf4*Δ (Fig. 6D, *P* value = 0.031, Wilcoxon rank-sum test). Altogether, these results demonstrate how the core genetic adaptation we identified is responsible for a large extent of fitness benefits across a wide range of macronutrient availability and, thus, largely robust to nutrient environments.

## Discussion

Gaining a deeper understanding of gene-by-environment (GxE) interactions within the context of compensatory evolution is crucial for elucidating how cells adapt and maintain functionality in the face of genetic and environmental pressures. In this work, we investigated how changes in glucose availability, a common environmental variable, shaped the evolutionary repair of DNA replication stress. Our findings demonstrate that while glucose availability significantly affects the physiology and adaptation speed of cells under replication stress, it does not alter the fundamental genome-wide compensatory mutations that drive fitness recovery and evolutionary repair.

### Glucose availability affects cell physiology, fitness, and evolutionary dynamics

Glucose availability significantly impacts the growth dynamics of both WT and *ctf4*Δ cells. In our study, glucose starvation partially restored cell cycle and the cell size of *ctf4*Δ cells to levels similar to WT, potentially explaining the proportional decline in fitness of *ctf4*Δ ancestor as glucose levels increased. Several explanations could account for this observation. One possibility could be the reduction in the nucleotide (dNTP) pool under glucose starvation. The reduced activity of the ribonucleotide synthetase, the enzyme responsible for dNTP production has been shown to slow down replication fork progression (Poli et al, 2012), and improve the overall fitness of *ctf4*Δ cells (Poli et al, 2012; Fumasoni and Murray, 2020). During glucose starvation, the levels of Rnr1, a subunit of the ribonucleotide reductase, decrease substantially (Corcoles-Saez et al, 2019), likely causing a subsequent drop in dNTP levels. In addition, *IXR1* loss-of-function, which is strongly selected in evolved *ctf4*Δ lines, results in reduced dNTP pools through reduced expression of *RNR1* (Tsaponina et al, 2011). Interestingly, we observed a significant reduction in the

fitness benefits of *IXR1* deletion under glucose starvation (Fig. EV5B), suggesting that the fitness effects of *ixr1*Δ may overlap with those of glucose starvation. To substantiate this hypothesis, we show that overexpression of an *RNR1* allele refractory to feedback inhibition (*rnr1*-D57N) (Chabes et al, 2003; Chabes and Stillman, 2007) reduces fitness in *ctf4*Δ cells and abolishes the fitness benefits of *IXR1* deletion in a *ctf4*Δ background (Fig. EV4B). Another possible explanation involves the alleviation of ribosomal DNA (rDNA)-related replication stress. The rDNA locus is particularly susceptible to DNA replication stress (Salim et al, 2017), with replication fork stalling and collapsing observed, even in WT conditions (Takeuchi et al, 2003). Glucose starvation dramatically reduces rDNA origin firing by 80% in extreme glucose starvation (0.05%) (Kwan et al, 2013). We speculate that the fitness recovery observed in *ctf4*Δ mutants under starvation could result from a combination of a reduced dNTP pool and reduced fork collapse at the rDNA locus.

Beyond the immediate physiological effects, glucose availability also influenced the evolutionary dynamics of replication stress mutants. Specifically, we found that *ctf4*Δ lines evolved more rapidly during the early generations under glucose-rich conditions. Our results are consistent with declining adaptability, as evidenced by the reduced rates of adaptation observed both between *ctf4*Δ and WT lines and among *ctf4*Δ lines evolved in different glucose conditions (Fig. EV2F). However, despite this accelerated gain in fitness, there were no significant glucose-dependent differences in their mutational profiles. This suggests that initial fitness levels, rather than glucose availability itself, may have driven the differences in adaptation rates (Couce and Tenaillon, 2015).

### A novel mechanism of adaptation to DNA replication stress through the transcription mediator complex

A single amino acid substitution at position 919 of Med14—an essential component of the mediator complex—was selected in 22 out of the 48 *ctf4*Δ populations across glucose conditions. This mutation became fixed in most populations by generation 1000, indicating strong positive selection (Figs. 5A and EV5C). The recurrence of this mutation and the fitness advantage conferred suggest a significant role for *med14*-H919P in coping with replication stress. Further analysis predicted that this amino acid substitution could lead to perturbations in the structure of Med14 TID. Based on existing literature, and our analysis of the transcriptome dataset for the degron-mediated depletion of the Med14 TID (Fig. EV4A), we tested three non-mutually exclusive hypotheses for the beneficial effect of *med14*-H919P: (i) reducing replication-transcription conflicts, (ii) lowering dNTP pools, and (iii) facilitating cohesion establishment. Interestingly, *med14*-H919P exhibited a slightly positive genetic interaction with mutant alleles implicated in all three processes (Fig. 5D,E). Since hypothesis ii predicts a negative interaction with *dun1*Δ, which also reduces dNTP levels, our findings allow us to reject the possibility that the fitness benefits of *med14*-H919P arise from altered deoxyribonucleotide metabolism. Instead, our results support hypotheses i and iii, which predict the alleviation of defects caused by defective R-loop metabolism (*rnh1*Δ *rnh201*Δ and *sen1-3*) and premature sister chromatid separation (*chl1*Δ).

Based on these results, we speculate that the fitness benefits of *med14*-H919P in alleviating DNA replication stress stem from its combined pleiotropic effects, including reduced replication-transcription conflicts and enhanced cohesion establishment. However, future mechanistic studies will be necessary to elucidate

the molecular details underlying these effects. Notably, dysregulation of the mediator complex has been implicated in multiple cancer types, and human Med14 has been specifically found to be downregulated in lymphoma (Syring et al, 2016).

## Reproducibility of evolutionary repair

Is compensatory evolution shaped by the environment where cells grow and divide? Our study suggests that adaptive strategies to cope with the severe intracellular cellular stress caused by replication perturbations are both predictable and largely independent of nutrient availability. This finding is particularly surprising considering the several reported links between starvation, the TOR pathway, and genome stability (He et al, 2021; Weisman et al, 2014). In addition, the effects of glucose on the cell physiology and fitness of replication stress mutants reported here could have suggested different selective pressures during evolution. What could explain the discrepancies between our results, and previous studies on evolutionary repair highlighting the role of the environment in shaping evolutionary trajectories (Filteau et al, 2015), and the heterogeneous behavior of evolved lines in various environments (Szamecz et al, 2014)? We propose several non-mutually exclusive hypotheses. First, environmental variability can influence population size, affecting access to and the spread of beneficial mutations. In our experiment, we maintained a consistent effective population size across conditions to control for this variable. Second, qualitative changes in the environment, rarely encountered in nature, may impose more severe constraints on evolutionary trajectories than the quantitative and physiological changes we explored. Third, the environment's influence on compensatory evolution may depend on the specific cellular module perturbed and its genetic interactions with other modules that are significantly influenced by environmental conditions. For example, the actin cytoskeleton, which must rapidly respond to extracellular stimuli, is likely to be more directly influenced by environmental factors (Filteau et al, 2015) compared to the DNA replication machinery, which operates within the nucleus and is relatively insulated from such changes. Supporting this idea, a study examining mutants' fitness across diverse environments found that conditions such as different carbon sources or TOR inhibition, similar to those used in this study, primarily affected genes involved in vesicle trafficking, transcription, protein metabolism, and cell polarity. In contrast, genes associated with genome maintenance, as well as their epistatic interactions, were largely unaffected (Costanzo et al, 2021). Finally, the severity of the fitness defects caused by cellular perturbations may dictate the robustness of evolutionary adaptation. The stronger the initial perturbation, the more likely it is that adaptive mutations accessible through the genetic interaction network are limited in number and have large effects, making it less likely that the network is significantly influenced by environmental changes. A number of recent findings are relevant to these hypotheses: a recent study proposed how the global yeast genetic interaction network is largely robust to environmental perturbation (Costanzo et al, 2021). An increasing body of work has also proposed how the effect of any given mutation is largely dependent on the fitness of the strain in which it occurs (Couce and Tenaillon, 2015; Johnson et al, 2023). This phenomenon has been explained by invoking global epistasis, defined as the high-order interaction network of the mutation with

multiple genetic loci (Diaz-Colunga et al, 2023; Reddy and Desai, 2021; Kryazhimskiy et al, 2014), which has been recently shown to be robust to the environment (Ardell et al, 2024). Future experiments, however, will be needed to discriminate between the above-mentioned hypotheses in the context of evolutionary repair.

How generalizable are our conclusions about the reproducibility of evolutionary repair to DNA replication stress across other organisms, species, or replication challenges? While dedicated future studies are needed to fully address these important questions, several lines of evidence are encouraging. A recent report demonstrated that the identity of suppressor mutations of lethal alleles was conserved when introduced into highly divergent wild yeast isolates (Paltenghi and van Leeuwen, 2024). Similarly, earlier work showed that even ploidy, which significantly alters the target size for loss- and gain-of-function mutations, affected only the identity of the genes targeted by selection, while the broader cellular modules involved remained consistent (Fumasoni and Murray, 2021). Moreover, divergent organisms experiencing different types of DNA replication stress exhibit some of the adaptive responses described here. For example, the yeast genus *Hanseniaspora*, which lacks the Pol32 subunit of the replisome, has also been reported to have lost the DNA damage checkpoint (Steenwyk et al, 2019). Human Ewing sarcoma cells carrying the fusion oncogene *EWS-FLI1* frequently exhibit adaptive amplification of the cohesin subunit *RAD21* (Su et al, 2021). Together, these findings suggest that while the specific details of DNA replication perturbations and the genomic features of organisms may shape the precise targets of compensatory evolution, the overarching principles and cellular modules affected are broadly conserved.

Our findings demonstrate that while glucose availability significantly affects the physiology and adaptation speed of cells under replication stress, it does not alter the fundamental genetic mutations that drive fitness recovery and evolutionary repair. The consistency of genetic adaptations across different glucose conditions, and their fitness benefits across different environments, suggest that cells follow a conserved pathway to recover from replication stress. The nearly neutral effects on fitness of the core adaptive mutations in WT suggest that they are likely to persist even after the initial replication stress is resolved. These insights contribute to our broader understanding of the fundamental principles of evolutionary cell biology and hold significant implications for predicting evolutionary outcomes in critical areas such as cancer progression and antibiotic resistance. A deeper understanding of the robustness and predictability of adaptive responses could guide the development of therapeutic strategies targeting environment-independent mechanisms, making them more broadly effective.

## Methods

**Reagents and tools table**

| Reagent/resource | Reference or source | Identifier or catalog number |
| --- | --- | --- |
| **Experimental models** | | |
| *S. cerevisiae* W303 | Dana Branzei Lab | |
| **Recombinant DNA** | | |
| pFA6a-pr*ACT1*-yCerulean-HphMX4 | Our lab | |

| Reagent/resource | Reference or source | Identifier or catalog number |
|---|---|---|
| pESC-*URA-RNR1*(-2μ) | Chabes and Stillman, 2007 | |
| pESC-*URA-rnr1*-D57M(-2μ) | Chabes and Stillman, 2007 | |
| **Chemicals, enzymes, and other reagents** | | |
| D-(+)-Glucose | Sigma-Aldrich | G7021 |
| Bacto™ Yeast Extract | Thermo Scientific | 288620 |
| Bacto™ Peptone | Thermo Scientific | 211677 |
| Yeast Nitrogen Base Without Amino Acids | Sigma-Aldrich | Y0626 |
| Adenine hemisulfate salt | Sigma-Aldrich | A9126 |
| L-Tryptophan | Sigma-Aldrich | T0254 |
| Uracil | Fisher Scientific | 157301000 |
| L-Leucine | Fisher Scientific | BP385 |
| L-Histidine | Merck | H8000 |
| L-Arginine Hydrochloride | Fisher Scientific | BP372 |
| L-Isoleucine | Fisher Scientific | BP384 |
| L-Lysine Hydrochloride | Fisher Scientific | BP386 |
| L-Phenylalanine | Fisher Scientific | 130311000 |
| L-Tyrosine | Fisher Scientific | 140641000 |
| L-Aspartic Acid | Fisher Scientific | BP374 |
| L-Methionine | Fisher Scientific | BP388 |
| L-Threonine | Fisher Scientific | BP394 |
| L-Valine | Fisher Scientific | BP397 |
| L-Serine | Fisher Scientific | 132661000 |
| Yeast Nitrogen w/o AA and AS | Thermo Scientific | H26271-36 |
| Yeast Nitrogen w/o AA and Phosphate | Formedium | CYN6702 |
| Ammonium Sulphate | Sigma-Aldrich | A-6387 |
| Monobasic Potassium Phosphate | Sigma-Aldrich | P-5379 |
| Geneticin (G418) | Santa Cruz Biotechnology | sc-29065A |
| Phleomycin | InvivoGen | Ant-ph-1 |
| 96 deep-well plates | Starlab | E2896-2100 |
| 96-well shallow plates | Corning | 734-1554 |
| Breath-Easy® Membrane | Sigma-Aldrich | Z380059-1PAK |
| Adhesive Film for Culture Plates, Porous | VWR | 60941-086 |
| Adhesive Foil | VWR | 391-1282 |
| RNaseA Solution | Sigma-Aldrich | R6148 |
| Zymolase | Zymo Research | E1005 |
| Proteinase K | GRiSP | GE010.0100 |
| Sytox Green | Thermo Fisher | S7020 |
| Isoton II Solution | Beckman Coulter | 8546719 |
| AgeI-HF | New England Biolabs | 174R3552S |
| Glycerol | Bioworld | BW-40120816-2 |

| Reagent/resource | Reference or source | Identifier or catalog number |
|---|---|---|
| Illumina Tagment DNA Enzyme and Buffer Small Kit | Illumina | 20034197 |
| Nextera XT Index Kit v2 Set A | Illumina | FC-131-2001 |
| **Software** | | |
| Saccharomyces Genome Database (SGD) | https://www.yeastgenome.org | |
| BD FACSDiva™ | BD Biosciences | |
| Python v3.9 | www.python.org | |
| ChimeraX, UCSF | https://www.cgl.ucsf.edu/chimerax/ Meng et al, 2023 | |
| Agilent Gene5 | Agilent | |
| SAMtools | http://htslib.org/ Li et al, 2009 | |
| GATK | www.broadinstitute.org/gatk Van der Auwera and O'Connor, 2020 | |
| VarScan v2.3.8 | varscan.sourceforge.net Koboldt, D, 2009 | |
| Cytoscape v3.10.2 | https://cytoscape.org/ Shannon et al, 2003 | |
| STRING v12.0 | https://string-db.org Szklarczyk et al, 2023 | |
| **Other** | | |
| BioTek Epoch | Agilent | |
| Fortessa Flow Cytometer | BD Biosciences | |
| Coulter Counter, Multisizer 4e | Beckman Coulter | |
| Multi-Purpose Tube Rotator | Fisher Scientific | |
| Illumina NovaSeq | Illumina | |

## Strains

All strains were derivatives of a modified version (Rad5+) of *S. cerevisiae* strain W303 (leu2-3;112 trp1-1; can1-100; ura3-1; ade2-1; his3-11,15; RAD5+) kindly provided by Dana Branzei. All strains and respective genotypes used in this study are listed in Table EV1. The ancestors of WT and *ctf4Δ* strains were obtained by sporulating a *CTF4/ctf4Δ* heterozygous diploid. This was done to minimize the selection acting on the ancestor strains before the beginning of the experiment. Diploid strains were grown on YP (1% Yeast Extract, 2% Peptone) + 2% D-glucose (YPD), transferred to 2 mL sporulation media (YP + Potassium Acetate (KAc) 2% enriched with 0.01% of Adenine and Tryptophan (A + T)) and grown overnight (O/N) at 30 °C. Cells were then washed twice in sterile milli-Q water, resuspended in 2% KAc supplemented with A + T, and incubated for four days at 25 °C. Tetrads were resuspended in water containing zymolyase (Zymo Research, Irvine, CA, US, 0.025 U/μL), incubated at 37 °C for 1 m 45 s, and dissected on a YPD plate using a micromanipulator. Spores were allowed to grow into visible colonies and genotyped by the presence of genetic markers and PCR.

## Media and growth conditions

All experiments were conducted in standard rich medium (YP) unless otherwise noted. For glucose availability experiments, D-glucose concentrations were adjusted to 0.25%, 0.5%, 2%, or 8%, using a 50% (w/v) D-glucose stock solution in sterile Milli-Q water. YP was supplemented with 0.01% of A + T. Transformation experiments were carried out in YP medium supplemented with 2% D-glucose. All cultures were incubated at 30 °C. For macronutrient availability experiments, synthetic complete media were used. The composition of synthetic media was as follows: 0.67% Yeast Nitrogen Base (YNB), supplemented with 0.025 mg/mL histidine, uracil, tryptophan, and adenine; 0.05 mg/mL leucine; 0.768 mg/L arginine, isoleucine, lysine, phenylalanine, and tyrosine; 1.152 mg/L aspartic acid, 0.256 mg/L methionine, 1.536 mg/L threonine, 2.176 mg/L valine, and 2% D-glucose. For glucose-limitation, glucose was added to a final concentration of 0.25%. Phosphate and nitrogen availability were adjusted following the conditions described by (Boer et al, 2008). Phosphate and nitrogen restriction and abundance media was prepared with 0.17% YNB without potassium phosphate monobasic and 0.17% YNB without ammonium sulfate, respectively.

## Growth dynamics

Ancestral clones of WT and *ctf4Δ* strains were grown to saturation in YPD (2% glucose) and diluted 1:100 into 150 μL of YP medium containing different glucose concentrations, with six replicates per condition in a 96-well plate. The plate was covered with breathable membranes (Breath-Easy®, Sigma-Aldrich) and incubated at 30 °C for 48 h in a microplate reader (BioTek Epoch 2, Agilent). Optical density at 600 nm (OD600) was measured every 10 min after 10 s of orbital shaking (speed). Protocol was defined using Agilent Gene5 software (version 3.12). Growth data were analyzed in Python v3.9 and a script based on growth_curve_analysis.py (https://github.com/nwespe/OD_growth_finder) was adapted.

## Cell cycle analysis

Cell cycle analysis was performed as described by Fumasoni et al, 2015. Briefly, three independent cultures per genotype of exponentially growing cells (~$1 \times 10^7$ cells) were collected from each glucose condition (0.25%, 0.5%, 2%, or 8%) by centrifugation and incubated in 70% ethanol, then eluted in 250 mM Tris-HCl (pH 7.5) at 4 °C overnight. Cells were washed with 50 mM Tris-HCl (pH 7.5), resuspended in 50 mM Tris-HCl (pH 7.5) with 0.4 mg/mL RNaseA (Sigma-Aldrich), and incubated at 37 °C for 1 h. Protein digestion was carried out with 5 μL of Proteinase K (20 mg/mL, GRiSP) for 2 h at 55 °C. After centrifugation and washing with 50 mM Tris-HCl (pH 7.5), cells were diluted tenfold in 50 mM Tris-HCl (pH 7.8) containing 1 mM SYTOX Green (Thermo Fisher) and analyzed using a Fortessa flow cytometer (BD Bioscience). DNA content per cell was quantified via the FITC channel, acquiring 10,000 events per sample. Cell cycle profiles were analyzed and visualized in Python v3.9 using the FlowCytometryTools package (https://eyurtsev.github.io/FlowCytometryTools/). The time spent in G1 and G2 phases for each genotype was estimated by deriving the relative cell distribution across cell cycle phases from FACS data, and multiplying them by the doubling time estimated from

growth curves under each glucose condition as described in (Alsina et al, 2025). Mean values were calculated from three independent biological replicates. All experiments were performed at the Flow Cytometry Facility of the Gulbenkian Institute for Molecular Medicine.

## Cell size

Three independent replicates of ancestral WT and *ctf4Δ* strains were grown to saturation in YPD with varying glucose concentrations (0.25%, 0.5%, 2%, or 8%). Cell volume was measured using a Coulter Counter (Multisizer 4e, Beckman), which estimates particle sizes based on the Coulter principle: as a cell passes through an electrolyte-filled orifice, the impedance change is proportional to the cell volume. For measurements, 10 μL of cell culture was transferred into a cuvette containing 10 mL of conducting fluid (Isoton II solution), using a 50 μm aperture.

## Fitness assay

To assess relative fitness, strains were competed against a fluorescently labeled reference strain. The reference strain was generated by integrating the pFA6a-prACT1-yCerulean-HphMX4 plasmid, digested with AgeI, into the *ACT1* locus of the ancestral WT, enabling strong expression of the yCerulean fluorescent protein under the *ACT1* promoter. For ancestral strains, WT, *ctf4Δ*, and the reference strain were inoculated from frozen stocks into 5 mL YPD tubes and grown to saturation at 30 °C. For evolved strains, 96-well plates containing frozen populations were thawed, and 10 μL of each population was inoculated into deep 96-well plates containing 1 mL of culture medium with the appropriate glucose concentration/composition. Cells were grown for 24–48 h, depending on the passage. For the competition assay, test and reference strains were mixed in a 1 mL culture (in 96-well plates) at a ratio reflecting the expected fitness difference (e.g., 1:1 for near-equal fitness) and allowed to proliferate at 30 °C for 24 h. An initial sample (10 μL) was taken after 5 h (day 0) and strain ratios were immediately measured using a flow cytometer (HTS, Fortessa, BD Bioscience). Cultures were then propagated with 1:1000 dilutions into fresh medium every 24 h for another 2 days. Strain ratios were measured, and the number of generations was estimated at each passage.

Ratios *r* were calculated based on the number of fluorescent and non-fluorescent events detected by the flow cytometer:

$$r = \frac{NonFluorescent_{events}}{Fluorescent_{events}}$$

Generations between time points *g* were calculated based on total cell counts measured at time 0 and time 24 h:

$$g = \ln\left(\frac{events_{t24}}{events_{t0}}\right)$$

Linear regression was performed between the ($g$, $\log_e r$) points relative to every sample. Relative fitness was calculated as the slope of the resulting line. The mean relative fitness was calculated from measurements obtained from at least three independent biological replicates. The relative fitness of the ancestral WT strain was used to normalize fitness across conditions. For reconstruction of adaptive mutations, the fitness effect of each reconstructed

mutation in WT background was used to normalize fitness in *ctf4Δ* background. All measurements of delta (Δ) fitness represent the difference between the tested strain and *ctf4Δ* ancestor. The evolved lines fitness trajectories were fitted with both a power law and linear regression and R² was calculated. The power law formula, adapted from (Wiser et al, 2013), is expressed as:

$$\overline{w} = (bt)^a + w_0$$

Where $\overline{w}$ represents mean evolved fitness, $t$ represents generations, $w_0$ denotes ancestral fitness. Two parameters, $b$ and $a$, were estimated based on the fit, with $b$ serving as a proxy for the adaptation rate. Fit and estimation of b parameter was not possible for populations 1, 5, and 10 of evolved WT in 2% glucose. No blinding was performed; data collection and analysis relied on automated pipelines or objective quantitative measurements. All fitness experiments were performed at the Flow Cytometry Facility of the Gulbenkian Institute for Molecular Medicine.

## Experimental evolution

The sample size for this study was based on standard field practices. The 48 populations used for experimental evolution were derived from the same isogenic clone, initially inoculated in 5 mL of YPD. Clones were randomly assigned across conditions. Evolution was performed in a high-throughput system using deep 96-well plates, each well containing 1.5 mL of medium. Ten microliters of saturated isogenic culture were inoculated into each well. Twelve parallel populations of each genotype evolved under different glucose concentrations, following a fixed layout. Plates were incubated at 30 °C with 70% humidity using the Multi-Purpose Tube Rotator 5–80 rpm (Fisher Scientific). Plates were fixed to a modified 64-place drum tube carousel (Fisher Scientific) with a custom-made 3D-printed plate holder, and rotated at 20 rpm set to the second degree of inclination available. Plates were sealed with sterile breathable rayon film (Adhesive Film for Culture, VWR). Daily passages were conducted by diluting a defined number of cells, adjusted to maintain Ne within the same order of magnitude. Ne was estimated using the formula:

$$N_e \sim N_0 \times n$$

Where $N_0$ is the initial population size calculated based on the bottleneck and final population size and $n$ is the number of generations which can be calculated based using by calculating the log2 of the dilution factor (Lenski et al, 1991). A 12-channel pipette (10 μL) was used to perform dilutions, following a vertical order to avoid cross-contamination between parallel populations. The standard dilution for WT at 2% glucose was 1:1000, allowing ~10 generations per passage. Cell concentration and dilution rates were adjusted at passages 5, 15, 30, 60, 90, and 100. All populations were evolved for a total of 1000 generations. To prevent WT cross-contamination in *ctf4Δ* populations, G418 (200 μg/mL) was added to the medium every five passages. At every fifth passage, 100 μL of each evolving population was mixed with 100 μL of 30% (v/v) glycerol and stored at −80 °C in shallow 96-well plates sealed with Adhesive Foil (VWR), following the same layout as the evolution

plates for future analysis. Due to contamination, population 8 of WT evolved in 0.5% glucose was excluded from all analyses.

## Whole-genome sequencing

Genomic DNA library preparation was performed as in (Koschwanez et al, 2013) with an Illumina Nextera DNA Library Prep Kit. Libraries were then pooled and sequenced with an Illumina NovaSeq (150 bp paired-end reads). The SAMtools software (Li et al, 2009) was then used to sort and index the mapped reads into a BAM file. GATK (Van der Auwera and O'Connor, 2020) was used to realign local indels, and VarScan (Koboldt et al, 2009) was used to call variants. Mutations were found using a custom pipeline written in Python v3.9. The pipeline (github.com/koschwanez/mutantanalysis) compares variants between the reference strain, the ancestor strain, and the evolved strains. A variant that occurs between the ancestor and an evolved strain is labeled as a mutation if it either (1) causes a non-synonymous substitution in a coding region or (2) occurs in a regulatory region, defined as the 500 bp upstream and downstream of the coding region.

## Copy number variations

Whole-genome sequencing and read mapping were done as previously described. The read depths for every unique 100-bp region in the genome were then obtained by using the VarScan copy number tool. A custom pipeline written in Python v3.9 was used to visualize the genome-wide CNVs. First, the read depths of individual 100 bp windows were normalized to the genome-wide median read depth to control for differences in sequencing depths between samples. The coverage of the ancestor strains was then subtracted from the one of the evolved lines to reduce the noise in read depth visualization due to the repeated sequences across the genome. The resulting CNVs were smoothed across five 100 bp windows for a simpler visualization. Final CNVs were then plotted relative to their genomic coordinate at the center of the smoothed window. The custom pipeline used for the data analysis is available on GitHub. The total amount of segmental amplifications across the 48 populations was determined. To determine whether the 11 segmental amplifications within the region of *SCC2* were statistically significant, a binomial test was performed.

## Convergent evolution

This method was performed based on the analysis of (Fumasoni and Murray, 2020). Briefly, it relies on the assumption that those genes that have been mutated significantly more than expected by chance alone, represent cases of convergent evolution among independent lines. The mutations affecting those genes are therefore considered putatively adaptive. The same procedure was used independently on the mutations found in WT and *ctf4Δ* evolved lines:

We first calculated per-base mutation rates as the total number of mutations in coding regions occurring in a given background (*ctf4Δ* evolved or WT evolved), divided by the size of the coding yeast genome in bp (including 1000 bp per ORF to account for regulatory regions)

$$\lambda = \frac{SNPs + indels}{Coding\ Base\ Pairs}$$

If the mutations were distributed randomly in the genome at a rate $\lambda$, the probability of finding n mutations in a given gene of length $N$ is given by the Poisson distribution:

$$P_{(n\,mutations|gene\,of\,length\,N)} = \frac{(\lambda N)^n\, e^{-\lambda N}}{n!}$$

For each gene of length $N$, we then calculated the probability of finding $n$ mutations if these were occurring randomly.

$$P_{(\geq n\,mutations|gene\,of\,length\,N)} = \sum_{k=n}^{\infty} \frac{(\lambda N)^k\, e^{-\lambda N}}{k!} = 1 - \frac{\Gamma(n+1, \lambda N)}{n!}$$

(where $\Gamma$ is the upper incomplete gamma function) which gives us the $P$ value for the comparison of the observed mutations with the null, Poisson model. In order to decrease the number of false positives, we then performed multiple-comparison corrections. Benjamini–Hochberg correction ($\alpha = 0.05$) was used for both the WT and $ctf4\Delta$ mutation datasets. Source Data lists the mutations detected in evolved $ctf4\Delta$ clones, after filtering out those that occurred in genes that were significantly mutated in the WT populations. Genes significantly selected in these clones are shown in dark gray (after Benjamini–Hochberg correction with $\alpha = 0.05$). The custom pipeline used for the data analysis is available on GitHub.

## Mutational profile analysis

Mutational analysis was conducted using both the dataset of mutations in coding regions of $ctf4\Delta$ and the output from convergent evolution analyses. Mutation counts per population and genotype were analyzed to assess mutational profiles. Total and synonymous mutations were calculated separately for each genotype and glucose concentration. A Mann–Whitney $U$ test was applied to compare mutation counts between genotypes for both total and synonymous mutations. Mutation counts were also grouped by glucose concentration for each genotype, and the Kruskal–Wallis test was used to test the effect of glucose in mutation counts. To assess clonality in evolved populations, read fraction was used as a metric. Mutation (or read) fraction represents the percentage of sequencing reads containing a specific mutation, with 100% indicating the mutation is present in all sequenced genomes.

## GO enrichment analysis

To identify functions under selection that may be influenced by mutations across multiple genes, we performed GO enrichment analysis on mutations found to be positively selected in $ctf4\Delta$ evolved clones, using two complementary methods. First, we identified putatively adaptive GO terms using the same approach described above, but considering the GO terms, instead of genes, as the unit of our analysis. (see Source Data) Second, we analyzed the full list of mutated CDS genes in $ctf4\Delta$ evolved clones by constructing an interaction network in Cytoscape (Shannon et al, 2003), using the STRING database plug-in (Szklarczyk et al, 2023). This approach allowed us to identify genes that, while not individually flagged for adaptive mutations, may belong to functional modules under selection. Mutations in these genes could contribute to the overall phenotype. The interaction network was curated and visualized in Cytoscape. GO enrichment for the

RNA-seq data was performed using the GO Term Finder tool (biological process ontology) from the Saccharomyces Genome Database (https://www.yeastgenome.org/).

## Spot assay

Cultures were grown in YPD at 30 °C for 48 h and adjusted to a $1 \times 10^8$ cells per ml concentration. Five serial 1:10 dilutions were made in YPD, and one drop (approximately 3 μL) of each dilution was pin-spotted onto YPD plates. The plates were then incubated for 48 h at 30 °C.

## RNA-seq analysis

The RNA-seq dataset from (Warfield et al, 2022), available at the Gene Expression Omnibus under accession GEO: GSE190778, the signal per gene across samples was normalized using *Schizosaccharomyces pombe* reads mapped as spike-ins. Data counts were rounded and imported into the R programming language (v.3.6.3) for differential gene expression analysis and visualization using the DESeq2 package (v.1.31.7) (Love et al, 2014). We analyzed gene expression from three independent replicates of degron-mediated TID disruption (TID_3IAA) and corresponding controls (TID_DMSO), resulting in 4338 genes for downstream analysis. The DESeq2 pipeline was implemented using the DESeq function, which estimates size factors with estimateSizeFactors, dispersion with estimateDispersions, and performs binomial GLM fitting and Wald statistics using nbinomWaldTest. Pairwise comparisons between TID_3IAA and TID_DMSO were tested through contrasts using the results function. Genes with adjusted $P < 0.01$ and $|\log2FC| > 1$ were considered differentially expressed. (equivalent to a twofold change). Normalized gene expression counts were obtained with the counts function (using normalized = TRUE). Results from the analysis are available in Dataset EV1.

## Mediator complex structure visualization and amino acid substitution modeling

Molecular graphics and analysis performed with UCSF ChimeraX, developed by the Resource for Biocomputing, Visualization, and Informatics at the University of California, San Francisco, with support from National Institutes of Health R01-GM129325 and the Office of Cyber Infrastructure and Computational Biology, National Institute of Allergy and Infectious Diseases. (Meng et al, 2023). The publicly available structure (PDB: 7UIO) was uploaded to ChimeraX for visualization and color coding. The Rotamers Structure Editing tool was used to model amino acid substitutions, the Dunbrack rotamer library (Shapovalov and Dunbrack, 2011) was used to predict steric clashes with neighboring residues. We selected the rotamer with the highest prevalence (0.813) and minimal steric clashes to reduce the likelihood of introducing structural artifacts into the model.

# Data availability

The datasets and computer code used in this study are now available in the following databases: FASTAQ files: European Nucleotide Archive PRJEB87420. Scripts used for data analysis are

available at the GitHub repository (https://github.com/FumaLab/NatalinoFumasoni_2025).

The source data of this paper are collected in the following database record: biostudies:S-SCDT-10_1038-S44320-025-00127-z.

## Peer review information

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

## Acknowledgements

The authors thank Professor Andrei Chabes for providing the *RNR1* overexpression plasmids and Dr. Rowin Appanah for providing the *sen1-3* mutant strains. The authors thank Jorge Carneiro, Thomas LaBar, Isabel Gordo, Ana Garoña and Manuel Vasquez for critical reading of the manuscript; Adolfo Alsina for assistance in cell cycle data analysis; Victor Mello for assistance in protein structure analysis; All the members of the Genome Maintenance and Evolution lab for helpful discussions; Tiago Paixão and Hugo Lainé from the Advanced Data Analysis (ADA) facility of the Gulbenkian Institute for Molecular Medicine for their availability and help with RNA-seq data processing and statistical analysis; the Flow Cytometry and Genomics Facility of Gulbenkian Institute for Molecular Medicine for their technical support. MN acknowledges the support of Fundação para a Ciência e a Tecnologia (FCT) doctoral fellowship (UI/BD/152252/2021). MF acknowledges the support of the Horizon 2020 Marie Skłodowska-Curie Actions (101030203—MF) and an FCT fellowship (2023.09068.CEECIND, https://doi.org/10.54499/2023.09068.CEECIND/CP2854/CT0003). Work in the Genome Maintained and Evolution lab was supported by FCT (2022.07846.PTDC), EMBO (5349-2023), HFSP (RGEC28/2023, https://doi.org/10.52044/HFSP.RGEC282023.pc.gr.168580), the Gulbenkian Foundation (FCG) and the Gulbenkian Institute for Molecular Medicine Foundation (GIMM).

## Author contributions

**Mariana Natalino**: Conceptualization; Resources; Data curation; Software; Formal analysis; Validation; Investigation; Visualization; Methodology; Writing—original draft; Writing—review and editing. **Marco Fumasoni**: Conceptualization; Resources; Data curation; Software; Formal analysis; Supervision; Funding acquisition; Validation; Investigation; Visualization; Methodology; Writing—original draft; Project administration; Writing—review and editing.

Source data underlying figure panels in this paper may have individual authorship assigned. Where available, figure panel/source data authorship is listed in the following database record: biostudies:S-SCDT-10_1038-S44320-025-00127-z.

## Disclosure and competing interests statement

The authors declare no competing interests.

# Expanded View Figures

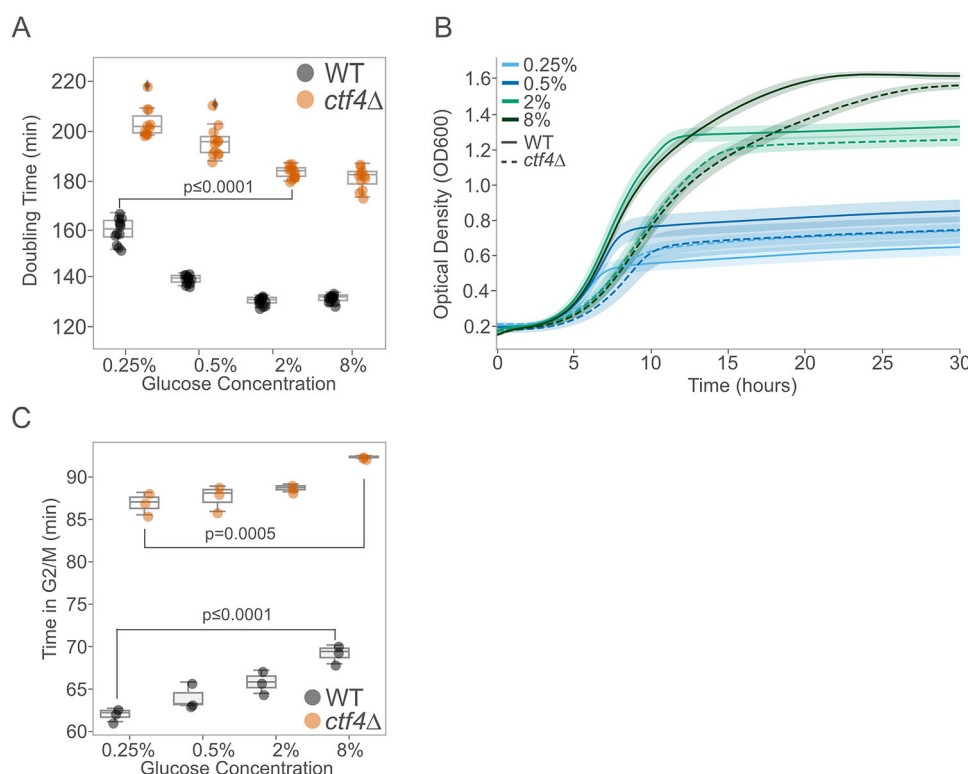

**Figure EV1. Glucose concentration impacts growth dynamics in the presence of DNA replication stress.**

(A) Population doubling time (min) of ancestral WT (black) and *ctf4Δ* mutant (orange) across different glucose concentrations ($n = 12$ biological replicates, Mann–Whitney *U* test with BH correction, *P* value $= 6.47 \times 10^{-5}$). (B) Growth curves of ancestral WT (solid line) and *ctf4Δ* (dashed line) over 30 h. Colors represent different glucose concentrations: light blue (0.25%), dark blue (0.5%), light green (2%), and dark green (8%). Bold lines indicate mean growth; shaded areas represent SD ($n = 12$ biological replicates). (C) Time spent (minutes) in G2/M phase for ancestral WT and *ctf4Δ*, across different glucose concentrations, estimated from DNA content and doubling times (see "Methods", $n = 3$ biological replicates, ANOVA Tukey's HSD) (WT 0.25% vs WT 8%, *P* value $= 2.10 \times 10^{-05}$). Box plots in (A, C) represent the median (center line), 25th and 75th percentiles (lower and upper bounds of the box), and whiskers extending to the smallest and largest values within 1.5× the IQR from the lower and upper quartiles, respectively. Detailed statistical analysis and underlying data for this figure are provided in Source Data. Source data are available online for this figure.

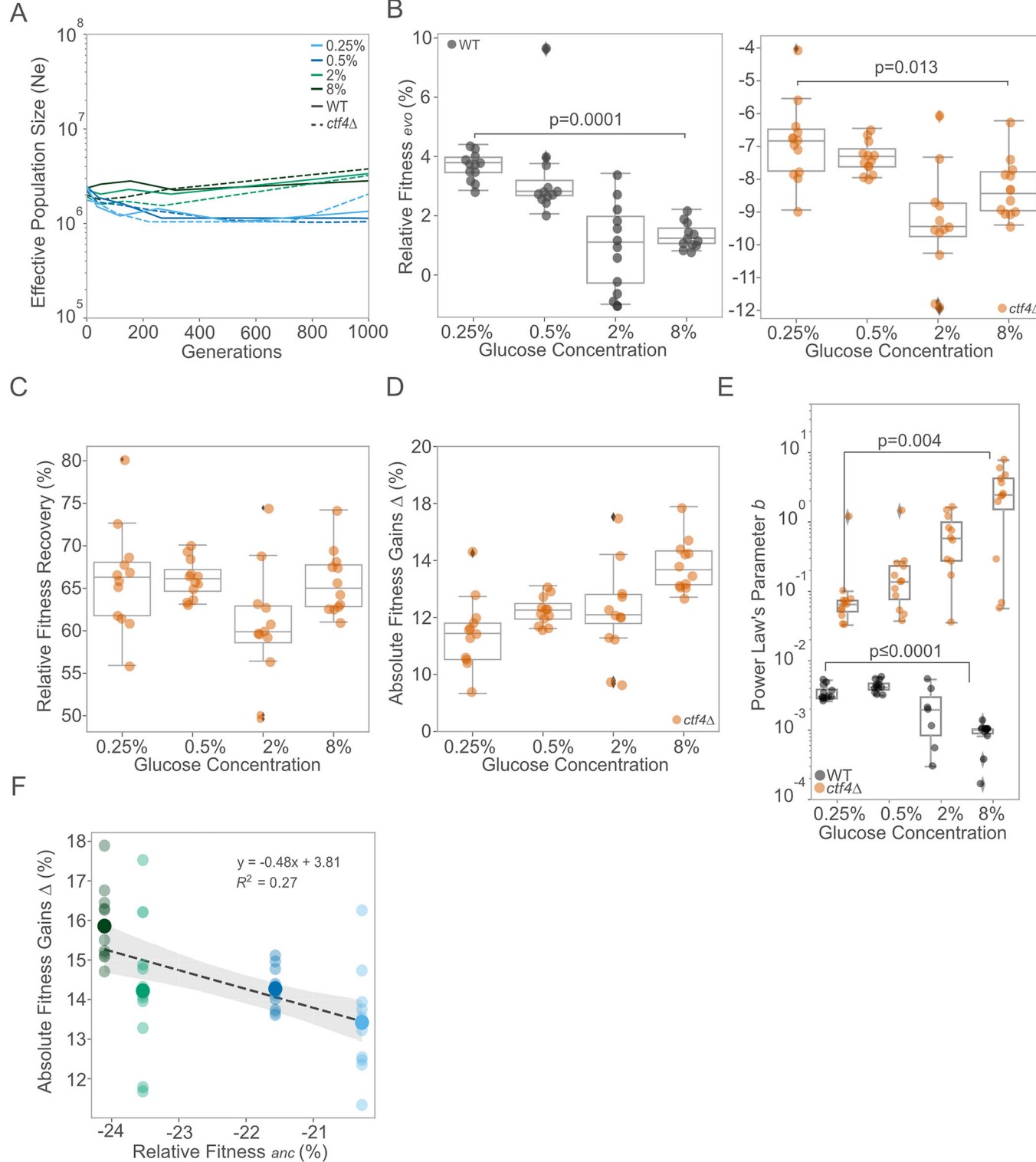

**Figure EV2.   Evolutionary dynamics under different glucose concentrations.**

(A) Estimated Ne across generations. Solid and dashed lines represent, respectively, adjusted Ne values for WT and *ctf4Δ* populations across generations. Colors represent different glucose concentrations: light blue (0.25%), dark blue (0.5%), light green (2%), and dark green (8%). (B) Relative fitness at generation 1000 for evolved WT (left panel, black) and *ctf4Δ* (right panel, orange) populations ($n = 3$ replicates per population, Mann–Whitney $U$ test with BH correction). (C) Relative fitness recovery at generation 1000 for evolved *ctf4Δ* (orange) populations. Percentage of fitness recovery was calculated by dividing the fitness gains (Δ) by the ancestral fitness ($n = 3$ replicates per population). (D) Absolute fitness gains (Δ) at generation 1000 for evolved *ctf4Δ* (orange) populations, per glucose concentration. Absolute fitness gains were calculated by subtracting ancestral relative fitness from evolved populations' relative fitness (Δ = evo% - anc%), both calculated as percentages relative to the same reference strain in the same glucose concentration. ($n = 3$ replicates per population). (E) Parameter $b$ from power law fit of fitness trajectories of populations across glucose concentrations ($n = 3$ replicates per population, Mann–Whitney with BH correction, $P$ value = $2.19 \times 10^{-4}$). (F) Correlation between the absolute fitness gains (Δ) during evolution and the fitness defect of ancestor strain in each glucose concentration. Colors represent different glucose concentrations: light blue (0.25%), dark blue (0.5%), light green (2%), and dark green (8%). Pairwise comparisons were performed using the Mann–Whitney test with Bonferroni correction. Box plots in (B–E) represent the median (center line), 25th and 75th percentiles (lower and upper bounds of the box), and whiskers extending to the smallest and largest values within 1.5× the IQR from the lower and upper quartiles, respectively. Detailed statistical analysis and underlying data for this figure are provided in Source Data. Source data are available online for this figure.

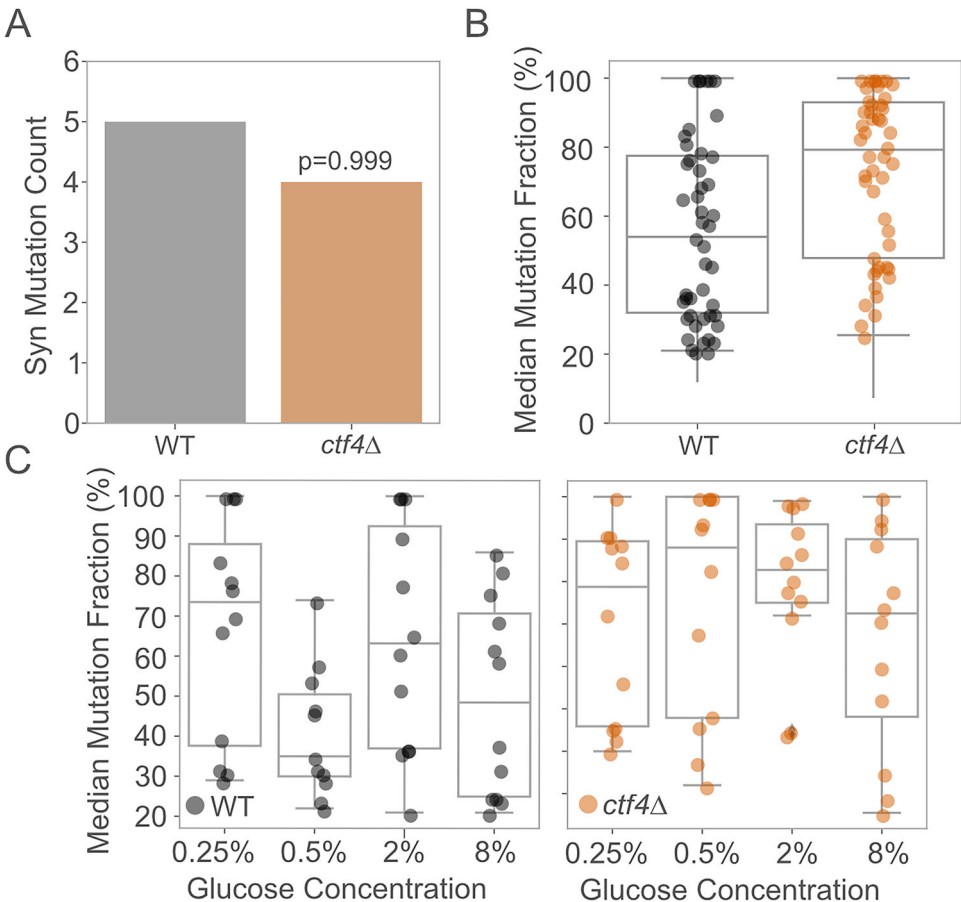

**Figure EV3. Mutational counts.**

(**A**) Total counts of synonymous (syn) mutations detected in evolved WT and *ctf4Δ* populations. Mann–Whitney *U* test was used to compare mutational counts. (**B**) Median mutation fraction (%) in CDS for evolved WT (black) and *ctf4Δ* (orange) populations at generation 1000 (*n* = 12 individual populations). Mann–Whitney *U* test was used to compare medians. (**C**) Median mutation fraction (%) of CDS mutations per glucose concentration, for WT (left) and *ctf4Δ* (right) at generation 1000. Statistical analysis was performed using the Mann–Whitney *U* test with BH correction. Box plots in (**B**, **C**) represent the median (center line), 25th and 75th percentiles (lower and upper bounds of the box), and whiskers extending to the smallest and largest values within 1.5× the IQR from the lower and upper quartiles, respectively. Detailed statistical analysis and underlying data for this figure are provided in Source Data. Source data are available online for this figure.

A

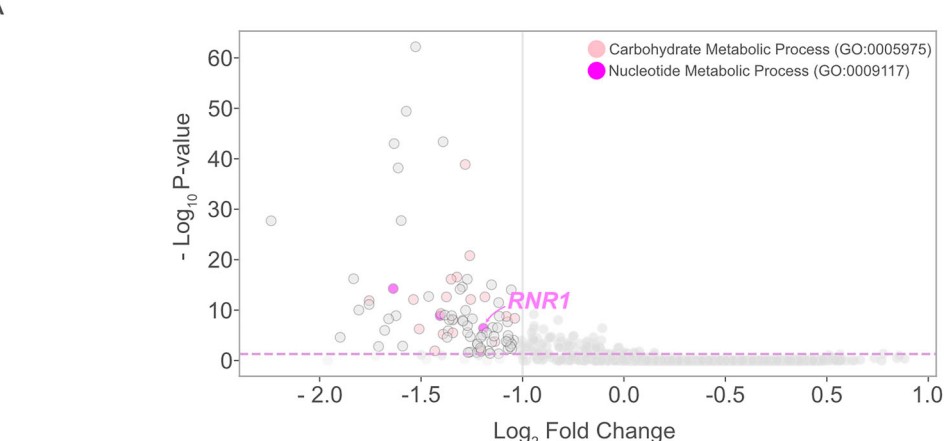

B

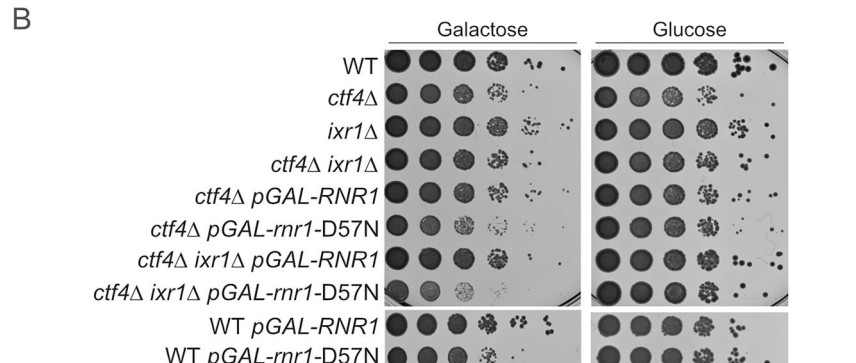

**Figure EV4.  Disruption of MED14 tail leads to reduced RNR1 expression.**

(A) Volcano plot of transcriptional changes after degron-mediated removal of Med14 C-terminal (Warfield et al, 2022). Dashed purple line indicates the significance threshold (P value ≤ 0.01). Differentially expressed genes were identified based on two criteria: (1) an adjusted P value < 0.01, ensuring statistical significance, and (2) an absolute log₂ fold change >1, corresponding to at least a twofold change in expression. Data analysis was performed using DESeq2. GO term enrichment analysis of downregulated genes highlights the carbohydrate (light pink) and nucleotide (magenta) metabolic processes. Detailed statistical analysis and underlying data for this figure are provided in Dataset EV1. (B) Overexpression of *RNR1* or its allele refractory to feedback inhibition (*rnr1*-D57N) under the Gal promoter in WT, *ctf4Δ* and *ctf4Δ ixr1Δ* backgrounds. Tenfold serial dilutions of the indicated strains were spotted onto media containing either galactose (inducing) or glucose (repressing) and incubated at 30 °C for 48 h. Overexpression of *rnr1*-D57N exacerbates the growth defects of both *ctf4Δ* and *ctf4Δ ixr1Δ*.

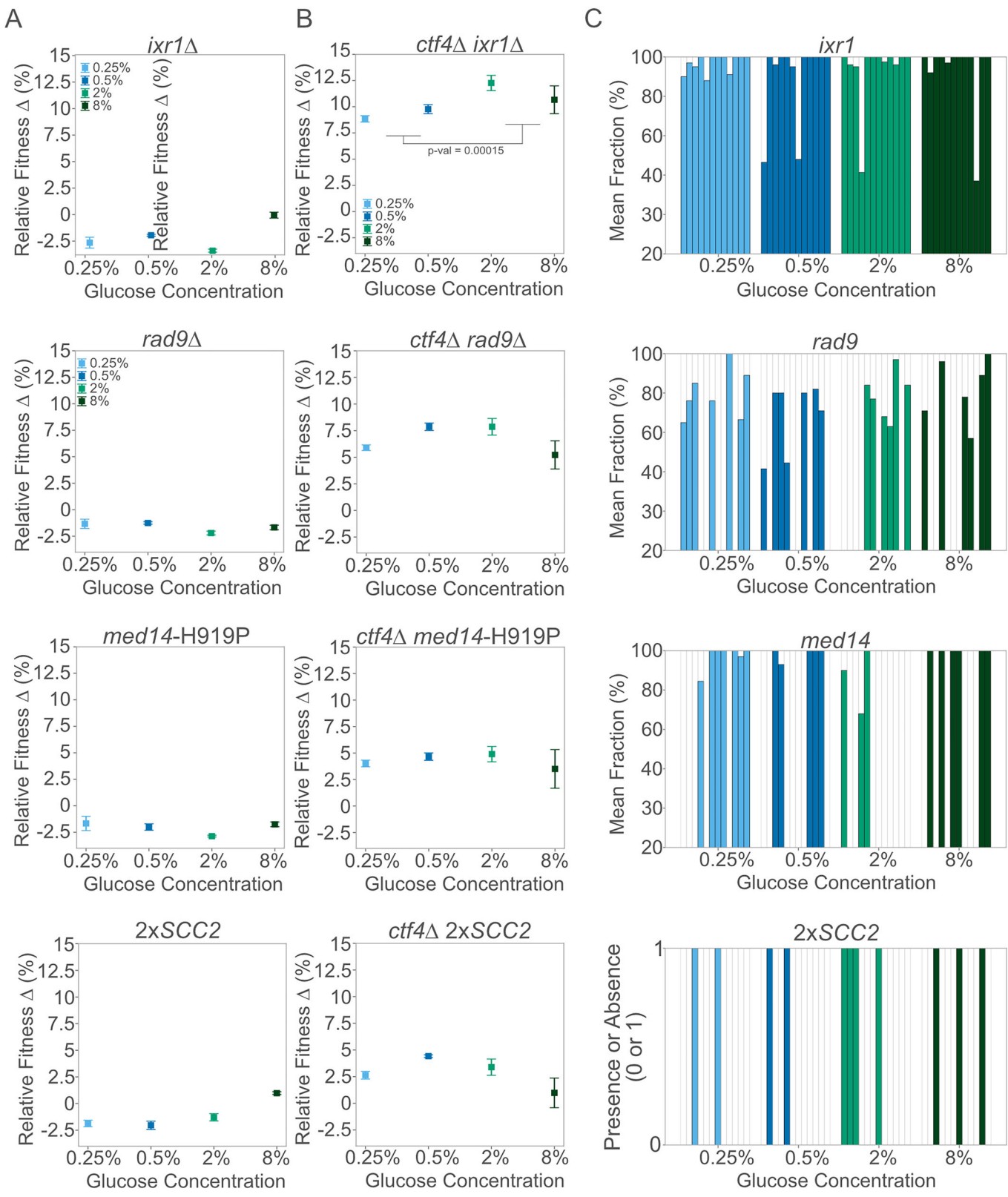

◄    **Figure EV5.  Fitness of reconstructed strains.**

(**A**) Mean relative fitness of reconstructed putative adaptive mutations in WT background. Error bars represent SD. Colors indicate glucose concentrations: light blue (0.25%), dark blue (0.5%), light green (2%), and dark green (8%) ($n = 4$). (**B**) Changes in mean relative fitness of ancestral *ctf4Δ* clones carrying reconstructed putative adaptive mutations ($\Delta = $ |anc %|-|reconstructed %|). Error bars represent standard deviation, with errors propagated from the two fitness measurements used to calculate fitness change ($\Delta$) ($n = 4$). (**C**) Frequencies of adaptive mutations across glucose concentrations at 1000 generations. Each bar represents the 12 parallel populations evolved in each glucose concentration, by order (1 to 12). Allele frequencies (mean fraction) in populations were derived from deep sequencing data of genomic DNA extracted from a population sample. Statistical analysis was performed using Mann–Whitney test to compare high and low glucose effect on fitness. Detailed statistical analysis and underlying data for this figure are provided in Source Data. Source data are available online for this figure.

