## [Peer Review File · Molecular Systems Biology]

Compensatory Evolution to DNA Replication Stress is Robust to Nutrient Availability

Mariana Natalino and Marco Fumasoni

Corresponding author(s): Marco Fumasoni (marco.fumasoni@gimm.pt)

Review Timeline:

Transfer from Review Commons:	19th Jan 25
Editorial Decision:	21st Jan 25
Revision Received:	31st Mar 25
Editorial Decision:	9th May 25
Revision Received:	13th May 25
Accepted:	5th Jun 25

Editor: Poonam Bheda

Transaction Report: This manuscript was transferred to Molecular Systems Biology following peer review at Review Commons.

Review #1

1. Evidence, reproducibility and clarity:

Evidence, reproducibility and clarity (Required)

This study investigates the compensatory evolutionary response of *Saccharomyces cerevisiae* to DNA replication stress, focusing on the influence of genotype-environment interactions (GXE). The authors used a range of experimental conditions with varying nutrient levels to assess evolutionary outcomes under replication stress. Their genomic analysis reveals that while glucose levels affect initial adaptation rates, the genetics of adaptation remain robust across all nutritional environments.

The research offers new insights into the adaptability of *S. cerevisiae*, emphasizing the role of the nutritional environment in evolutionary processes related to DNA replication stress. It identifies recurrent advantageous mutations under different macronutrient availabilities and uncovers a novel role for the RNA polymerase II mediator complex in adaptation to replication stress.

Overall, this well-designed study adds to the growing recognition of the complexity and robustness of evolutionary responses to environmental stressors. It provides strong evidence that compensatory evolution to replication stress is robust across varying nutritional conditions. It both challenges and reinforces previous findings regarding the resilience of the yeast genetic interaction network to environmental perturbations. The detailed analysis of specific compensatory mutations and their fitness impacts across different conditions offers valuable insights into adaptive dynamics over 1000 generations, contributing a clear empirical framework for understanding how replication-associated stress shapes evolutionary outcomes in diverse environments. Based on the analysis:

1. The conclusions are generally well-supported by the presented data. The evolution experiments and genomic analyses are robust and provide convincing evidence for the study's main claims. The authors took steps to eliminate bias, such as maintaining an adequate N_e , which, if not done, could have compromised their conclusions by affecting genetic drift and limiting the population's access to beneficial mutations.
2. The figures are well-designed and easy to understand.
3. The methodology is well-described and appears reproducible. The authors provide sufficient details on experimental procedures. Experimental replication is adequate, with multiple evolutionary lines.
4. They also made efforts to validate their observations, such as the validation of

mutations, the prediction of interactions in the Med14 structure, and its potential implication in gene regulation, as well as the analysis of the cumulative fitness benefit and the reconstruction of the quadruple mutant.

There are, however, a few results that would benefit from further clarification:

1. The experimental design is strong, offering a diverse range of conditions. However, the high glucose condition (8%) stands out as significantly different from the neutral 2% condition, both in range and margin, compared to the low glucose conditions (0.25-0.5%). While this mainly affects growth profiles and evolvability in the early generations, a brief explanation in the discussion would strengthen the conclusions. Specifically, addressing:

a) The rationale behind selecting these particular glucose concentrations.

b) How other glucose concentrations might influence the outcomes.

Providing this additional context would enhance the reader's understanding of the experimental setup and its potential implications, while also offering insights into the broader applicability of the findings and possible directions for future research.

2. In the discussion section, a more explicit comparison with similar studies in other model organisms would help contextualize the findings within the broader field of evolutionary biology. While the results appear robust, it would be beneficial to explore how they align with or contrast to previous studies on DNA damage, particularly in bacteria or highly complex eukaryotes.

****Minor comments:****

1. The presentation of data in the figures is clear and informative. However, some figure legends could benefit from more detailed explanations. For example, although the statistical tests used are mentioned in the methods section, it would be helpful to also include them in the figure legends, such as in legend 1acde, as well as in all other figures.

2. In terms of broader conclusions, here are a few suggestions, though they are, of course, optional:

a) The study could benefit from exploring the potential trade-offs of adaptive mutations in the hypothetical return to environments without replication stress, at least theoretically. This would provide a more comprehensive understanding of the evolutionary constraints.

b) A brief discussion of the potential limitations of using lab strains versus wild isolates of

S. cerevisiae would offer valuable context for the generalizability of the findings.

c) It would be valuable to present the differences in ploidy in the context of other studies, such as the nutrient-limitation hypothesis (e.g., 'The Evolutionary Advantage of Haploid Versus Diploid Microbes in Nutrient-Poor Environments' by Bessho, 2015), since, as previously demonstrated by the authors of this article that is being reviewed, ploidy may influence the evolutionary trajectories of DNA repair. Interrelating these three terms: nutrient-limitation, ploidy, and DNA repair could be an interesting avenue to explore in the discussion.

3. Specific details:

a) Line 116: To improve clarity, it would be beneficial to refer to the figure right after the statement: 'However, their relative fitness improved compared to the WT reference as the initial glucose levels (Figure X).'

b) Line 404: The statement about antibiotics and cancer progression is somewhat brief here; it might be helpful to provide more context on why this mechanism influences these processes (here or before).

c) Line 418: "were re-suspended in water containing zymolyase (Zymo Research, Irvine, CA, US, 0.025 $\mu/\mu\text{L}$), incubated at". Something is missing in the units.

d) Line 459: "and G2 phases for each genotype was estimated by deriving the the relative cell distribution". The article "the" is repeated.

e) Fig. 1a: The x-axis ticks appear misaligned, which makes it difficult to interpret the boxplots. For example, at 0.25, the tick is closer to the orange boxplot than to the black one. In contrast, at 2%, the tick seems well-centered."

f) Figure 3 could benefit from a general legend at the top regarding the colors, as finding it in 2c was not intuitively easy.

g) I didn't review the code on GitHub.

2. Significance:

Significance (Required)

The main strength of the study is that it shows robustness of compensatory evolution across varying nutrient conditions. The study adds to the growing body of literature on DNA replication stress and evolutionary adaptation by showing that compensatory evolution can occur regardless of nutrient availability. This fundamental finding challenges prior assumptions that nutrient conditions significantly alter evolutionary outcomes, contributing to a more nuanced understanding of how cells respond to stress. Furthermore, the discovery of the RNA polymerase II mediator complex's role in this process is particularly novel and opens new lines of investigation.

Advance in the field: The results advance our understanding of evolutionary biology, particularly in the context of DNA replication stress and compensatory evolution. The study demonstrates that evolutionary repair mechanisms are predictable, even under variable environmental conditions, which has key implications for evolutionary biology and therapeutic applications.

Audience:

This paper will be of interest to a specialized audience in evolutionary biology, genomics, and cell biology, particularly those interested in DNA replication stress and adaptive evolution. Researchers studying stress responses in model organisms, such as *S. cerevisiae*, will find the findings valuable, as will those working in applied fields where stress adaptation is a critical factor (e.g., industrial yeast fermentation, drug development, disease resistance, cancer research, or aging studies).

Expertise:

Evolutionary biology, genomic analysis, and cellular stress responses, with a particular focus on experimental evolution under DNA damage stress in *Saccharomyces cerevisiae*. Recently graduated and beginner reviewer.

3. How much time do you estimate the authors will need to complete the suggested revisions:

Estimated time to Complete Revisions (Required)

(Decision Recommendation)

Less than 1 month

4. Review Commons values the work of reviewers and encourages them to get credit for their work. Select 'Yes' below to register your reviewing activity at Web of Science Reviewer Recognition Service (formerly Publons); note that the content of your review will not be visible on Web of Science.

Yes

Review #2

1. Evidence, reproducibility and clarity:

Evidence, reproducibility and clarity (Required)

Review of "Compensatory evolution to DNA replication stress is robust to nutrient availability" from Natalino and Fumasoni.

The paper addresses the effect of sugar availability in shaping compensatory evolution. The first observation of the paper is that cell physiology changes by modulating glucose availability also in strains that come with defective DNA replication (ctf4-null previously studied by the authors). An intriguing result is that ctf4-null grows comparatively better in low concentrations of glucose. This is hypothesized to be a consequence of both the decrease in dNTPs in low glucose, which causes slow down of fork progression, and/or reduced fork collapse at rDNA locus. Hence, wild types and ctf4-null show an opposite trend: in the mutant, the lowest concentration of glucose is the least affected by the mutation; in wild type, the highest concentration is the least affected. Adaptation rate is inversely related with the initial fitness.

The effect on physiology and adaptation rate is a starting point for asking the key question: are evolutionary trajectories influenced by the growth conditions? The answer is negative: evolution experiments show the very same core of genetic changes at all sugar concentrations. The result is apparently at odds with previous publications, and the authors conclude that, in this particular setting, availability of carbon sources plays a minor role compared to impaired DNA replication. The different rates of adaptation in WT and mutant is rather explained by the initial fitness at the different glucose concentrations, which, as mentioned, is opposite in WT and ctf4-null mutants.

The paper also reports a new mutation in MED14, component of the transcription mediator complex, which rescues the lack of Ctf4 activity. The study is interesting and asks a relevant question. The experiments are well executed and convincing, but the paper can be strengthened by testing some of the hypotheses which are put forward.

****Main points****

1. The raw data for evolutionary dynamics (Figure S2C) are fitted with the power law suggested by Wisner and Lenski, and return different values of the parameter 'b'. The authors say that the result depends greatly on the initial conditions ("due to the varying initial fitness of *ctf4*Δ cells across different glucose environments, they display an opposite trend to WT"). Around the initial values, however, the curves are non-monotonic, especially for low glucose availability. Both for WT and *ctf4*-null there is an initial drop in fitness, after which fitness increases. If one would neglect this initial dynamics, the value of the parameter 'b' would likely be different. In general, one can question whether curves with this shape are best fitted by the power law proposed by Wisner and Lenski. For example, for the WT 0.25% glucose the linear fit gives a better R² (why do the authors show the linear fit anyway?). This impression is further reinforced by the observation that Wisner and Lenski fit dynamics that last 50,000 generations, here the curves last 1/50th of it. In conclusion, I would question whether the parameter 'b' is a solid measurement of 'rate of adaptation'. Also, normalizations makes it difficult to appreciate the result shown in Figure 2B.

I think the authors should look for a different way to show the different trend in adaptation dynamics for different glucose concentrations between wild types and mutants. For example, they could move Figure S2C in the main text to stress the result shown in Figure 2C, which already shows the difference between WT and mutant. This is especially true if what Figure 2C shows is $(\text{evo-anc})/\text{evo}$. This is not fully clear to me: in the legend it refers to the delta, in the label of the y-axis I read that this is a percentage.

To conclude: the data show a different trend between wild types and mutants, which is interesting. Fitting it with the power law seems to be neither required nor appropriate. I suggest the authors to show the WT vs mutant pattern differently.

2. In Figure S2C, the individual trajectories for WT at 2% glucose are strangely variable. In this case, plotting the average does not make too much sense. This result is strange, since this is the default condition, where cells are grown without any change of sugar concentration. Can the authors give any rationale? Are there other available results to

replace those published in Figure S2C?

3. The molecular explanation given for the rescue of *ctf4*-null proposes a very relevant role for dNTPs downregulation. Particularly, both for *Irx1* and *med14-H919P*, the authors propose that this happens via *Rnr1* downregulation.

At this stage, this is only a hypothesis. The molecular verification of the central role of *Rnr1* downregulation would make the conclusion much stronger. For example, a preliminary test would imply that duplicating *RNR1* in *ctf4*-null *irx1*-null and/or *ctf4*-null *med14-H919P* would revert the rescue. Any other experiment addressing this point would be useful to improve the paper.

4. The authors propose from Figure S4B that the rescue of *ixr1*-null is less evident at low sugar concentration since both conditions trigger a reduction of dNTPs. I think this is interesting, since it would provide a link between glucose concentration and evolutionary trajectories to adaptation, which is what the authors wanted to study.

In particular, one would predict that 0.25% glucose would see less *ixr1*-null than the other glucose conditions. I could not (was not able to) confute this hypothesis from the data shown in the paper. Likewise, for *med14-H919P*. If the authors have not tested it, it would be worth trying.

5. The combination of the four genetic adaptation (Fig 6B) would benefit from an experimental verification to show that the different solutions are not mutually exclusive. This is not obvious: if more than one solution acts by reducing dNTPs, maybe their combined effect is less strong than what measured theoretically. The authors could derive some clones at the end of the experiment and Sanger sequencing some of the four genes, to confirm the co-presence of some of them in the same cell.

****Minor points****

Figures

- S4B: in the legend it should be explained that it is compared to *ctf4D* .
- 2A: the color code is not fully clear to me: what does green and blue indicate? higher and lower than 2%?
- S3A: the authors should show the statistical difference between WT and *ctf4*-null, which is mentioned as non-existent in p.6

Text

- RNR1 is not really the gene with the highest score in Figure 5D, not even close: can you give a rationale for pin-pointing it (see also main point 3)?
- The med14-H919P mutation is observed in 22/48 wells. I guess the authors checked already: are some of these wells close to each other in the plate?
- Compensatory evolution of ctf4-null in 2% glucose is the experiment published by Fumasoni and Murray in eLife. In that paper, there is no trace of mutations in MED14. I think the authors should comment on this (different method for detecting putative compensatory mutations?).
- I may be mistaken, but Szamecz et al do not actually investigate whether different conditions result in different evolutionary trajectories (i.e., different genetics), and so their results may not be at odds with those presented here.

typos

p.18, line 564 preformed -> performed

p. 6 line 189 with a strongly skew -> with a strong skew ?

2. Significance:

Significance (Required)

This is a well-done paper that could be of interest for the community of evolutionary biologists, scientists working on metabolism and cell division. It addresses an interesting problem, how metabolism affects compensatory evolution. Among the strengths: experiments are well done, the results are novel, the cross-talk between metabolism and evolutionary repair is intriguing. Among the weaknesses, the fact that the molecular explanations for the observations are only hypothesized and not tested experimentally. This is where the authors could improve the manuscript.

3. How much time do you estimate the authors will need to complete the suggested revisions:

Estimated time to Complete Revisions (Required)

(Decision Recommendation)

Between 1 and 3 months

4. Review Commons values the work of reviewers and encourages them to get credit for their work. Select 'Yes' below to register your reviewing activity at Web of Science Reviewer Recognition Service (formerly Publons); note that the content of your review will not be visible on Web of Science.

Yes

Review #3

1. Evidence, reproducibility and clarity:

Evidence, reproducibility and clarity (Required)

This paper combines phenotypic and genomic data from an experimental evolution study in yeast to assess how repeatable evolution is in response to DNA replication stress. Importantly, the authors ask whether genotype by environment interactions influence repeatability of their evolved lines. To this end, the authors have constructed an elegant highly-replicated experiment in which two yeast genotypes (WT and CTF4 KO) were evolved under a variety of glucose levels for 1,000 generations. Recurrent mutations are found across many replicates, suggesting that repeatability is robust to GxE interactions. Of course, the authors correctly identify that these results are dependent on many particulars, as is always the case in biology, but provide a comprehensive discussion to accompany their results. I do not have any major comments to give, but simply some suggestions and points of clarification.

****Major comments:**** N/A

****Minor comments:****

L19: I found the definition for compensatory evolution/mutations to be somewhat vague in the introduction (and subsequently throughout the text). It's clear that this was written for a more medical/physiological audience, but without a more explicit explanation of compensatory evolution/mutations, it became difficult to properly weigh some claims/discussions made by the authors later on. Do you define compensatory mutations as those which completely recover WT function/fitness, or are simply of opposite effect to the altered genotype? Others define "compensatory evolution" as simply any epistatically

interacting amino acid substitutions (Ivankov et al, 2014). It would be nice to see more explicitly defined.

Along these lines, I would have liked to see a more direct comparison/discussion of the degree to which deletion lines recovered. I can see from Fig 2E and Fig S2B that fitness increased quite a bit; would it not be possible to include a figure on the degree of compensation (basically relative fitness of evolved deletion lines - relative fitness of ancestral deletion lines)?

L57: Another minor nitpick that just comes down to semantics. When discussing "96 parallel populations", it invokes a higher sense of replication than is actually present in the study. I would rephrase this to something along the lines of "12 replicate populations across 8 treatments under conditions of [...]".

L185-187: The wording here needs to be clarified. Be explicit in that are examine the ratio (or count) of synonymous to non-synonymous mutations here, otherwise the interpretations appears to be direct contradiction to the (as written) results. Only after viewing the supplemental figure was I able to figure out what exactly was meant here.

L349-350: The authors observe higher rates of adaptation in deletion lines than WT lines, and discuss this in adequate detail. Although not explicitly mentioned, this is consistent with a diminishing returns epistasis model (that could be beneficial to discuss, but is not necessary), which has been implicated in modulating the degree of repeatability observed along evolutionary trajectories (Wünsche et al. 2017). Although definitely not required for this already very nice manuscript, I think it would be very rewarding if the authors were to eventually analyze fine-scale dynamics of phenotypic and genomic adaptation to mine for these putative interactions and their influence on repeatability.

2. Significance:

Significance (Required)

It is clear to me that a great deal of time and care has been put into this study and the preparation of this manuscript. The science and analyses are appropriate to answer the questions at hand, and it bodes well that whenever I had a question pop up while reading, they were typically answered immediately after. I think that this manuscript will be broadly relevant to both biologists both evolutionary and clinical, and was written in a way to be accessible to both.

As someone with an expertise in repeatable evolution, I felt most excited by the observation of so many parallel substitutions at a single amino acid across deletion lines. As the authors rightfully point out in the results and discussion, it's likely that this degree of robustness is highly dependent on the particular mechanism of disruption that cells experience. The authors then go above and beyond to functionally validate the putative molecular mechanisms of (repeatable) adaptation in this system. While it may not always be possible to accomplish in non-model organisms, such multi-modal approaches will be crucial to advance the field of repeatable evolution.

3. How much time do you estimate the authors will need to complete the suggested revisions:

Estimated time to Complete Revisions (Required)

(Decision Recommendation)

Less than 1 month

Yes

Review #4

1. Evidence, reproducibility and clarity:

Evidence, reproducibility and clarity (Required)

The authors investigated the effects of DNA replication stress on adaptation in different nutrient availabilities by passaging wild-type and *ctf4Δ* *Saccharomyces cerevisiae* in media with varying levels of glucose over ~1000 generations. The *ctf4Δ* strain experiences increased DNA replication stress due to the deletion of a non-essential replication fork protein. The authors found differences in evolution between wild-type and *ctf4Δ* yeast, which held across different growth media. This study identified a compensatory single amino acid variant in Med14, a protein in the mediator complex of RNA polymerase II, that

was specifically selected in *ctf4Δ* strains. The authors conclude that while environmental nutrient availability has implications for cell fitness and physiology, adaptation is largely independent and instead dependent on genetic background. The data provide excellent support for the key aspects of the models, although some details are (to me) overstated.

****Major comments:****

- A *ctf4Δ* mutant strain was used to investigate the effects of replication stress. Why was this mutant chosen instead of other deletions that cause different types of replication stress? It's not clear from the study that the effects are generalizable to other forms of replication stress.
- The authors could be clearer that a (the?) cause of the *ctf4Δ* fitness defect is spurious upregulation of *RNR1*. I don't think it is mentioned until the Discussion, but it is highly relevant to Fig 4, and to the adaptations one would expect from *ctf4Δ*.
- In Figure 1E, there is a very large spread in the relative fitness at 2% and 8% glucose, but this was not commented on. Is this heteroscedasticity expected?
- The *med14-H919P* mutant was highly selected in *ctf4Δ* strains, independent of glucose availability. Is this variant found in any natural yeast strains (i.e., are there environments that select for this variant)? Also, if this variant is found in natural strains, does it co-occur with other mutations that could affect DNA replication?
- The statement on lines 271-273 is not particularly well-supported. The analysis of the Warfield data suggest that reduced expression of *RNR1* could be causal, but the data don't go as far as showing how the *med14* mutation is advantageous in *ctf4Δ*. Further experimentation would be necessary to support the possibilities that the authors discuss.
- The authors comment that the *med14-H919P* mutant could have implications for the stability of *Med14*, based on computational modelling. Verifying the stability of the *med14-H919P* in vivo would strengthen this discussion.
- In the discussion, the authors propose that the context of the perturbation may influence the robustness of adaptation. A more detailed explanation of this point (including a discussion of the findings of other similar studies investigating different conditions) would be helpful to further bolster this section.

****Minor comments:****

- Competitions were performed between *ctf4Δ* strains and a constructed strain with *yCerulean* integrated at *ACT1*. Is the fitness of the fluorescent strain comparable to the ancestral wild-type strain (i.e., in a competition between the ancestral WT and the fluorescent strain, does either have an advantage)?

- In Figure 3, the legends for panels B and C appear to be swapped. Discussion of Figure 3 on pages 6 and 7 appear to reference the wrong panels.
- In Figure 4A and B, having the same colour scale between both heatmaps is misleading, as the scales are different. Consider having the same scale across both heatmaps so that enrichments are visually comparable.
- In Figure 4C, having a legend in the figure for node size would be helpful to understand the actual number of populations with mutations in each gene.

2. Significance:

Significance (Required)

In this study, a high-throughput evolution experiment uncovered the effects of genetic background on the development of adaptive mutations. The authors were able to identify a single amino acid variant of Med14 (med14-H919P) that was positively selected in *ctf4Δ*. Furthermore, they demonstrated the causality of med14-H919P in conferring a fitness advantage in *ctf4Δ*. The novelty of this mechanistic finding opens future avenues of investigation regarding the interaction network of the mediator complex in conditions of DNA replication stress. A limitation of the study is that only one mechanism of replication stress was assessed (*ctf4Δ*). Other gene mutations that cause replication stress would be interesting to assess and would provide a more thorough investigation of the effects of DNA replication factors on evolvability.

This work will be of interest to researchers in the population genetics and genotype-by-environment fields, as it suggests the robustness of evolvability to environmental factors in the specific condition of DNA replication stress. As discussed by the authors, this finding differs from other works that have linked environmental conditions to adaptive evolution to different conditions, and is concordant with work that indicates the robustness of genetic interactions to environmental stresses. Furthermore, the identification of the highly-selected med14-H919P variant will be of interest to the DNA replication field. There is the potential for future work investigating the role of Med14 in mediating the response to DNA replication stress in both yeast and mammalian cell contexts, since the authors note that there are links between altered mediator complex regulation and cancers. Although I suspect that the very different regulation of RNR in mammalian cells makes it unlikely that the kind of upregulation of dNTP pools seen in *ctf4Δ* would be induced by replication stress in mammalian cells.

3. How much time do you estimate the authors will need to complete the suggested revisions:

Estimated time to Complete Revisions (Required)

(Decision Recommendation)

Less than 1 month

No

Revision Plan

Manuscript number: RC-2024-02762

Corresponding author(s): Marco, Fumasoni

1. General Statements

Dear Review Commons editorial team,

Thank you for coordinating the thorough and careful review of our manuscript. We are especially grateful to the four anonymous reviewers for recognizing the value of our work and for their constructive suggestions on how to improve it.

We are encouraged by the positive reception of our main conclusions on the robustness of adaptation to DNA replication stress and its relevance to multiple fields. All reviewers provided insightful comments, with reviewers #2 and #4 emphasizing that further experimental validation of the hypothesized role of reduced dNTPs in alleviating fitness during constitutive DNA replication stress would strengthen the paper. While the precise molecular mechanisms underlying this suppression are not the primary focus of this manuscript, we are eager to perform additional experiments based on the reviewers' suggestions.

Below, we present a detailed revision plan in the form of a point-by-point response to their comments. We have already implemented changes in response to most comments (highlighted in green). Additionally, to address the remaining requests, we propose further experiments and analyses to be conducted during the revision process (highlighted in blue).

The comments that have been partially addressed but still require revision are listed in Section 2, highlighted in the appropriate text color. If the reader prefers to review the comments in chronological order, they can refer to the attached document.

2. Description of the planned revisions

Insert here a point-by-point reply that explains what revisions, additional experimentations and analyses are planned to address the points raised by the referees.

REVIEWER #1

A brief discussion of the potential limitations of using lab strains versus wild isolates of *S. cerevisiae* would offer valuable context for the generalizability of the findings.

This is an excellent point. While addressing it fully would warrant a separate manuscript, we provide our comments here, along with similar observations raised by this and other reviewers, as follows:

Line 450: "How generalizable are our conclusions about the reproducibility of evolutionary repair to DNA replication stress across other organisms, species, or replication challenges? While dedicated future studies are needed to fully address these important questions, several lines of evidence are encouraging. A recent report demonstrated that the identity of suppressor mutations of lethal alleles was conserved when introduced into highly divergent wild yeast isolates (Paltenghi and van Leeuwen, 2024). Similarly, earlier work showed that even ploidy, which significantly alters the target size for loss- and gain-of-function mutations, affected only the identity of the genes targeted by selection, while the broader cellular modules involved remained consistent (Fumasoni and Murray, 2021). Moreover,

Revision Plan

divergent organisms experiencing different types of DNA replication stress exhibit some of the adaptive responses described here. For example, the yeast genus *Hanseniaspora*, which lacks the Pol32 subunit of the replisome, has also been reported to have lost the DNA damage checkpoint (Steenwyk et al., 2019). Human Ewing sarcoma cells carrying the fusion oncogene *EWS-FLII* frequently exhibit adaptive amplification of the cohesin subunit *RAD21* (Su et al., 2021). Together, these findings suggest that while the specific details of DNA replication perturbations and the genomic features of organisms may shape the precise targets of compensatory evolution, the overarching principles and cellular modules affected are broadly conserved.”

Furthermore, we plan to search a recently published database of variants found in natural isolates of *S. cerevisiae* to assess whether similar evolutionary processes to those described in this study may have occurred in wild strains.

REVIEWER #2

The molecular explanation given for the rescue of *ctf4*-null proposes a very relevant role for dNTPs downregulation. Particularly, both for *Irx1* and *med14-H919P*, the authors propose that this happens via *Rnr1* downregulation. At this stage, this is only a hypothesis. The molecular verification of the central role of *Rnr1* downregulation would make the conclusion much stronger. For example, a preliminary test would imply that duplicating *RNR1* in *ctf4*-null *irx1*-null and/or *ctf4*-null *med14-H919P* would revert the rescue. Any other experiment addressing this point would be useful to improve the paper.

We agree that the experiment suggested by the reviewer, or similar tests, would substantiate our hypotheses and strengthen the paper. Specifically, we plan to perturb dNTP production in both *ctf4Δ irx1Δ* and *ctf4Δ med14-H919P* mutants through genetic manipulation of known factors involved in dNTP synthesis. We will then compare the resulting fitness to the expectations based on our hypotheses: reduced fitness benefits of the double mutants upon increasing dNTP levels and/or increased fitness in *ctf4Δ* mutants by decreasing dNTP levels through alternative mechanisms.

The combination of the four genetic adaptation (Fig 6B) would benefit from an experimental verification to show that the different solutions are not mutually exclusive. This is not obvious: if more than one solution acts by reducing dNTPs, maybe their combined effect is less strong than what measured theoretically. The authors could derive some clones at the end of the experiment and Sanger sequencing some of the four genes, to confirm the co-presence of some of them in the same cell.

The co-occurrence of nearly every combination of the four core adaptive mutations we identified can be inferred from their relative frequencies, as revealed by deep whole-genome sequencing of the evolved populations (Fig. S4C). In these data, we observe populations carrying each pairwise combination of mutations at frequencies exceeding 50%, implying their coexistence. Moreover, many combinations of mutations approach or reach fixation. A particularly striking example is *ctf4Δ* Population 11, evolved in 8% glucose, where all core adaptive mutations are present at 100% frequency. These findings provide robust evidence that the different adaptive solutions are not mutually exclusive and can coexist within the same genetic background.

Nevertheless, we agree that experimentally verifying the compatibility and fitness of the four genetic adaptations described in Figure 6B (now Fig 6C) would further strengthen our conclusions. To this end, we plan to reconstruct all combinations of mutations observed at high frequency in the final evolved populations. We will then measure their fitness and compare it to that of the evolved populations, as well as to the theoretical expectations based on additivity currently presented in Figure 6C.

Revision Plan

RNR1 is not really the gene with the highest score in Figure 5D, not even close: can you give a rationale for pinpointing it (see also main point 3)?

The reviewer is correct. Perturbations of the mediator complex, which regulate the expression of most of RNA PolII transcripts, is expected to result in changes in the expression of a large set of genes. However, our focus on dNTPs and *RNR1* is based on the following rationale:

- 1) Gene Ontology Enrichment Analysis: The downregulated genes in our dataset are enriched for the 'nucleotide metabolism' term, which includes pathways critical for dNTP production and directly linked to DNA replication and repair.
- 2) Role of *RNR1*: Among the downregulated genes, *RNR1* stands out as it encodes the major subunit of ribonucleotide reductase, the rate-limiting enzyme in dNTP synthesis. This enzyme is essential for DNA replication, and cells experiencing constitutive DNA replication stress, as in our system, are particularly sensitive to changes in dNTP levels.

To make this rationale more explicit to the reader, we are adding the following sentence in the discussion:

Line 404: "Nucleotide metabolism, particularly ribonucleotide reductase, is essential for dNTP production. Given the role of dNTPs in regulating DNA replication and repair, the advantage of *med14-H919P* mutants in the *ctf4Δ* background may stem from reduced dNTP levels caused by the perturbed TID domain."

In addition, following the reviewers' suggestions, we are conducting additional experiments to investigate the role of *med14-H919P* mutants in enhancing fitness under conditions of constitutive DNA replication stress (See response to reviewer #4). We anticipate that the final revised manuscript will offer further insights into the role of dNTPs or present alternative explanations for the observed phenomena.

REVIEWER #3: N/A

REVIEWER #4

The *med14-H919P* mutant was highly selected in *ctf4Δ* strains, independent of glucose availability. Is this variant found in any natural yeast strains (i.e., are there environments that select for this variant)? Also, if this variant is found in natural strains, does it co-occur with other mutations that could affect DNA replication?

We agree that this is an intriguing question. To address it, we plan to explore existing databases of variants identified in *S. cerevisiae* natural isolates. Specifically, we will investigate whether the *med14-H919P* mutation is present in these strains, identify any potential environmental factors that may select for it, and assess whether it co-occurs with other mutations that could influence DNA replication processes.

The statement on lines 271-273 is not particularly well-supported. The analysis of the Warfield data suggest that reduced expression of RNR1 could be causal, but the data don't go as far as showing how the *med14* mutation is advantageous in *ctf4Δ*. Further experimentation would be necessary to support the possibilities that the authors discuss.

The sentence the reviewer refers to is: "Overall, these results show how an amino acid substitution in the Med14 subunit of the mediator complex, putatively affecting transcription, is strongly selected, and advantageous, in the

Revision Plan

presence of constitutive DNA replication stress.” We are unsure which aspect of the statement is seen as unsupported. The mutation's strong selection in *ctf4Δ* is demonstrated in Figures 5A, 6A, and S4C, while its advantageous nature is supported by Figures 5B and S4B. Regarding the mechanism, we have been cautious with our phrasing, describing its effect on transcription as “putative” (Line 272) and suggesting that our observations “are compatible with” reduced dNTP availability in *med14-H919P* cells due to *RNR1* downregulation (Line 361).

The main focus of this study is to explore how nutrient availability influences evolutionary dynamics and compensatory adaptation in cells lacking Ctf4. We believe the identification of a novel selected allele (Fig. 5A) and confirmation of its benefit across glucose conditions (Fig. 5B) serves as an excellent complement to the primary conclusions (present in the title). We invite the reviewer to consider that the molecular basis of such a phenotype is not mentioned in our abstract, as we believe that its precise characterization would require a dedicated study on Med14.

Nonetheless, we are encouraged by the reviewer’s interest in this newly identified compensatory mutant (also noted by Reviewer #2), and we are eager to perform further experiments to better understand the biological processes affected by this mutation. We plan to extend our work as follows:

Based on known phenotypes associated with perturbations of Med14, we propose the following novel hypotheses regarding the mechanism by which *med14-H919P* alleviates *ctf4Δ* defects:

- 1) Decreased replication-transcription conflicts: Conflicts between the transcription machinery and replication forks are known to cause fragile sites, leading to increased chromosome breaks and genomic instability (Garcia-Muse and Aguilera, 2016). A general reduction in PolIII transcription during replication, resulting from perturbations of the mediator complex, could reduce these conflicts and mitigate the fitness defects observed in *ctf4Δ* cells.
- 2) Increased cohesin loading: We have demonstrated that amplification of the cohesin loader *SCC2* is beneficial in the absence of Ctf4. Recent findings (Mattingly et al., 2022) indicate that the mediator complex recruits *SCC2* to PolIII-transcribed genes. The *med14-H919P* mutation may enhance the fitness of *ctf4Δ* cells by facilitating cohesin loading during DNA replication.
- 3) Decreased dNTP levels: As discussed in the manuscript, perturbations of Med14 subunits in the mediator complex reduce the expression of genes, including those associated with nucleotide metabolism. Notably, these include *RNR1*, the major subunit of ribonucleotide reductase. The *med14-H919P* mutation could benefit the *ctf4Δ* background by counteracting the reported spurious increase in dNTPs, which affects replication fork speed (Poli et al., 2012).

We plan to distinguish between these hypotheses using the following approaches. First, the proposed mechanisms underlying Hypotheses 1 and 3 suggest that *med14-H919P* is a loss-of-function mutation, while Hypothesis 2 implies a gain-of-function effect. Testing the impact of a heterozygous *med14-H919P* allele in a homozygous *ctf4Δ* strain will allow us to differentiate between these two categories of mechanisms. Additionally, we aim to investigate the molecular process affected by the *med14-H919P* allele by analyzing its genetic interactions with genes involved in replication-transcription conflicts, cohesin loading, and dNTP production (See also response to reviewer #2).

We believe that the results of these experiments will provide further insights on the mechanism of suppression exerted by *med14-H919P* in the presence of constitutive DNA replication stress, without diverting the reader from the main message of the paper.

Revision Plan

In the discussion, the authors propose that the context of the perturbation may influence the robustness of adaptation. A more detailed explanation of this point (including a discussion of the findings of other similar studies investigating different conditions) would be helpful to further bolster this section.

We are now supporting this concept more explicitly by commenting on other studies as follows:

Line 429: “Third, the environment’s influence on compensatory evolution may depend on the specific cellular module perturbed and its genetic interactions with other modules that are significantly influenced by environmental conditions. For example, the actin cytoskeleton, which must rapidly respond to extracellular stimuli, is likely to be more directly influenced by environmental factors (Filateau et al., 2015) compared to the DNA replication machinery, which operates within the nucleus and is relatively insulated from such changes. Supporting this idea, a study examining mutants’ fitness across diverse environments found that conditions such as different carbon sources or TOR inhibition, similar to those used in this study, primarily affected genes involved in vesicle trafficking, transcription, protein metabolism, and cell polarity. In contrast, genes associated with genome maintenance, as well as their epistatic interactions, were largely unaffected (Costanzo et al., 2021)”.

In addition, to further substantiate this hypothesis, we plan to re-analyze published datasets on fitness and epistatic interactions among genes in various environments, testing whether specific cellular modules are more prone to changes following shifts in nutrient conditions.

3. Description of the revisions that have already been incorporated in the transferred manuscript

Please insert a point-by-point reply describing the revisions that were already carried out and included in the transferred manuscript. If no revisions have been carried out yet, please leave this section empty.

REVIEWER #1

The experimental design is strong, offering a diverse range of conditions. However, the high glucose condition (8%) stands out as significantly different from the neutral 2% condition, both in range and margin, compared to the low glucose conditions (0.25-0.5%). While this mainly affects growth profiles and evolvability in the early generations, a brief explanation in the discussion would strengthen the conclusions. Specifically, addressing:

- a) The rationale behind selecting these particular glucose concentrations.
- b) How other glucose concentrations might influence the outcomes. Providing this additional context would enhance the reader's understanding of the experimental setup and its potential implications, while also offering insights into the broader applicability of the findings and possible directions for future research.

We thank the reviewer for pointing out the need to clarify the rationale behind the glucose concentrations used in our study, an aspect we agree should have been better explained. In response, we have added the following text detailing the chosen conditions and their established effects on cellular metabolism.

Line 67: “Glucose is the most abundant monosaccharide in nature, and represents the preferred source of energy for most cells.”

Revision Plan

Line 110: "...we grew WT and *ctf4Δ* cells in varying glucose concentrations to induce distinct physiological states. Low glucose levels (0.25% and 0.5%) induce caloric restriction and ultimately glucose starvation (Lin et al 2000, Smith et al. 2009). These conditions elicit increased respiration (Lin et al., 2002), sirtuins expression (Guarente, 2013), autophagy (Bagherniya et al. 2018), DNA repair (Heydari et al., 2007), and reduced recombination at the ribosomal DNA locus (Riesen and Morgan, 2009) ultimately extending lifespan in several organisms (Kapahi et al., 2016). In contrast, standard laboratory conditions typically use 2% glucose, promoting a rapid proliferation environment to which strains have been adapted since laboratory domestication (Lindergren, 1949). Finally, elevated glucose concentrations (such as 8%) result in higher ethanol production (Lin et al., 2012) and reactive oxygen species (ROS) levels (Maslanka et al., 2017).

2) In the discussion section, a more explicit comparison with similar studies in other model organisms would help contextualize the findings within the broader field of evolutionary biology. While the results appear robust, it would be beneficial to explore how they align with or contrast to previous studies on DNA damage, particularly in bacteria or highly complex eukaryotes.

We appreciate this suggestion to better contextualize our findings within the broader literature, as it provides an opportunity to highlight the unique aspects of our work. While many studies have explored how environmental factors shape fitness landscapes and influence evolutionary strategies, to our knowledge, only a few have addressed this in the context of compensatory evolution, where cells must recover fitness lost due to intracellular perturbations. To address this point, we have added a discussion of additional examples involving other model organisms, highlighting their difference with the question asked in this work.

Line 34: "Genotype-by-environment (GxE) interactions are well-documented. For example, several studies on *E. coli* have demonstrated how different environments influence fitness and epistatic interactions among adaptive mutations in the Lenski Long-Term Evolution Experiment (Ostrowski et al., 2005, 2008; Flynn et al., 2012; Hall et al., 2019). Adaptive mutations in viral genomes similarly exhibit variable fitness effects across different hosts (Lalic and Elena, 2012; Cervera, 2016). Furthermore, interactions between mutations in the *Plasmodium falciparum* dihydrofolate reductase gene have been shown to predict distinct patterns of resistance to antimalarial drugs (Ogbunugafor et al., 2016). However, the role of environmental factors in shaping evolution within the context of compensatory adaptation, when fitness defects primarily arise from intracellular perturbations, remains much less explored."

However, if the reviewer have particular additional studies in mind, we welcome further suggestions to include in the final manuscript.

Minor comments:

1) The presentation of data in the figures is clear and informative. However, some figure legends could benefit from more detailed explanations. For example, although the statistical tests used are mentioned in the methods section, it would be helpful to also include them in the figure legends, such as in legend 1acde, as well as in all other figures.

We are now reporting the statistical test used for each comparison also in figure legends.

2) In terms of broader conclusions, here are a few suggestions, though they are, of course, optional:

- a) The study could benefit from exploring the potential trade-offs of adaptive mutations in the hypothetical return to environments without replication stress, at least theoretically. This would provide a more comprehensive understanding of the evolutionary constraints.

Revision Plan

We thank the reviewer for the suggestion, we had performed the measurements but did not comment on them explicitly. We are now commenting on them as follows:

Line 310: "In the WT background, all mutations were nearly neutral, with only minimal deleterious or advantageous effects on fitness depending on glucose concentrations (Fig S4A)."

Line 468: "The nearly neutral effects on fitness of the core adaptive mutations in WT suggest that they are likely to persist even after the initial replication stress is resolved."

- c) It would be valuable to present the differences in ploidy in the context of other studies, such as the nutrient-limitation hypothesis (e.g., 'The Evolutionary Advantage of Haploid Versus Diploid Microbes in Nutrient-Poor Environments' by Bessho, 2015), since, as previously demonstrated by the authors of this article that is being reviewed, ploidy may influence the evolutionary trajectories of DNA repair.
- d) Interrelating these three terms: nutrient-limitation, ploidy, and DNA repair could be an interesting avenue to explore in the discussion.

In response to comments c and d, we have now commented on the intersection between ploidy and other types of DNA perturbation in the paragraph starting in line 491 (see response above)

3) Specific details:

- a) Line 116: To improve clarity, it would be beneficial to refer to the figure right after the statement: 'However, their relative fitness improved compared to the WT reference as the initial glucose levels (Figure X).'
- b) Line 404: The statement about antibiotics and cancer progression is somewhat brief here; it might be helpful to provide more context on why this mechanism influences these processes (here or before).
- c) Line 418: "were re-suspended in water containing zymolyase (Zymo Research, Irvine, CA, US, 0.025 μM), incubated at". Something is missing in the units.
- d) Line 459: "and G2 phases for each genotype was estimated by deriving the the relative cell distribution". The article "the" is repeated.
- e) Fig. 1a: The x-axis ticks appear misaligned, which makes it difficult to interpret the boxplots. For example, at 0.25, the tick is closer to the orange boxplot than to the black one. In contrast, at 2%, the tick seems well-centered."
- f) Figure 3 could benefit from a general legend at the top regarding the colors, as finding it in 2c was not intuitively easy.

The typos and suggestions raised in points 3a-f have now been corrected in the manuscript.

REVIEWER #2

The raw data for evolutionary dynamics (Figure S2C) are fitted with the power law suggested by Wisner and Lenski, and return different values of the parameter 'b'. The authors say that the result depends greatly on the initial conditions

Revision Plan

("due to the varying initial fitness of *ctf4Δ* cells across different glucose environments, they display an opposite trend to WT"). Around the initial values, however, the curves are non-monotonic, especially for low glucose availability. Both for WT and *ctf4*-null there is an initial drop in fitness, after which fitness increases. If one would neglect this initial dynamics, the value of the parameter 'b' would likely be different.

The non-monotonic trend in fitness highlighted by the reviewer is likely due to technical factors: Fitness at Generation 0 was measured with high precision in a low-throughput manner early in the project. In contrast, fitness from Generation 100 to 1000 was measured later in the study in a high-throughput fashion, necessitated by the large number of competitions conducted (96 wells × 4 time points × 6 replicates = 2304 assays). This difference in methodologies may have introduced a slight offset when the datasets were combined at Generation 100. Following the reviewer's suggestion, we have excluded the data point at Generation 100 responsible for this non-monotonic behavior and re-fitted the curves. While this adjustment has caused minor changes in the parameter 'b', the qualitative trends, particularly the opposing trends between WT and *ctf4Δ* as glucose increases, remain consistent (Figure_rev_only 1). To ensure transparency, we have retained all recorded fitness values in the original figure for reference.

In general, one can question whether curves with this shape are best fitted by the power law proposed by Wisner and Lenski. For example, for the WT 0.25% glucose the linear fit gives a better R2 (why do the authors show the linear fit anyway?). This impression is further reinforced by the observation that Wisner and Lenski fit dynamics that last 50,000 generations, here the curves last 1/50th of it. In conclusion, I would question whether the parameter 'b' is a solid measurement of 'rate of adaptation'. Also, normalizations makes it difficult to appreciate the result shown in Figure 2B. I think the authors should look for a different way to show the different trend in adaptation dynamics for different glucose concentrations between wild types and mutants. For example, they could move Figure S2C in the main text to stress the result shown in Figure 2C, which already shows the difference between WT and mutant. This is especially true if what Figure 2C shows is (evo-anc)/evo. This is not fully clear to me: in the legend it refers to the delta, in the label of the y-axis I read that this is a percentage.

We thank the reviewer for prompting us to clarify our methods for reporting fitness changes over time. The fitness values are reported, throughout the paper, as a percentage change relative to the reference WT strain. The gain in fitness during evolution (reported as Δ) represents the difference between the evolved strain (evo%) and the ancestral strain (anc%), calculated as $\Delta = \text{evo}\% - \text{anc}\%$. This represents the absolute gain, rather than the relative gain. This value is still reported as a percentage as it's the same scale and unit as the two values being subtracted. We have included additional details to clarify this aspect in the figure legend.

“(C) Absolute fitness gains (Δ) at generation 1000 for evolved WT (upper panel, black) and *ctf4Δ* (lower panel, orange) populations. Box plots show median, IQR, and whiskers extending to 1.5×IQR, with individual data points beyond whiskers considered outliers. Absolute fitness gains were calculated by subtracting the ancestral relative fitness from the relative fitness of the evolved ($\Delta = \text{evo}\% - \text{anc}\%$), both calculated as percentages relative to the same reference strain in the same glucose concentration.”

To conclude: the data show a different trend between wild types and mutants, which is interesting. Fitting it with the power law seems to be neither required nor appropriate. I suggest the authors to show the WT vs mutant pattern differently.

We followed the reviewer's suggestion and moved Figure S2C, which depicts the detailed fitness trajectories over time, into the main manuscript as Figure 2D. We agree that presenting these trajectories alongside the absolute fitness gains (now in Figure S2C) provides a more intuitive and effective depiction of the evolutionary dynamics of WT and *ctf4Δ* strains without relying solely on the power-law fit. Additionally, we quantified the *mean* adaptation rate,

Revision Plan

calculated as the absolute fitness gain (Δ) divided by the total number of generations (now Figure 2B). While no individual method definitively captures the adaptation rates across the experiment, these complementary analyses consistently highlight the same trends noted by the reviewer. We have re-written the main text as follows:

Line 171: “By generation 1000, both WT and *ctf4Δ* evolved lines achieved, on average, slightly higher fitness in low glucose compared to high glucose conditions (Fig S2B). However, due to the varying initial fitness of *ctf4Δ* cells across different glucose environments, they recovered the same extent of the original defect (Fig S2C). *ctf4Δ* lines displayed an opposite trend to WT, with increasing absolute fitness throughout the experiment as glucose concentration rose (Fig S2B vs S2D). The different absolute fitness gains over the same number of generations highlight distinct mean adaptation rates (Fig 2B). These differences are evident when examining the evolutionary dynamics of the evolved lines over time (Fig 2C). Additionally, we approximated the fitness trajectories using the power law function (Fig 2C, dashed purple lines), previously proposed to describe long-term evolutionary dynamics in constant environments (Wiser et al., 2013). The parameter *b* in this formula determines the curve's steepness, and can be used to quantify the global adaptation rate over generations (Fig S2E). Collectively, these analyses demonstrate that, unlike WT cells, *ctf4Δ* lines adapt faster in the presence of high glucose. This evidence aligns with the declining adaptability observed in other studies (Moore et al., 2000; Kryazhimskiy et al., 2014; Couce & Tenaillon, 2015), where low-fitness strains consistently adapt faster than their more fit counterparts (Fig S2F).”

Overall, these results demonstrate that cells can recover from fitness defects caused by constitutive DNA replication stress regardless of the glucose environment. However, adaptation rates under DNA replication stress exhibit opposing trends compared to WT cells, with faster adaptation yielding greater fitness gains in higher glucose conditions.”

2- In Figure S2C, the individual trajectories for WT at 2% glucose are strangely variable. In this case, plotting the average does not make too much sense. This result is strange, since this is the default condition, where cells are grown without any change of sugar concentration. Can the authors give any rationale? Are there other available results to replace those published in Figure S2C?

We agree with the reviewer that the individual trajectories for WT at 2% glucose are intriguing. However, we do not find these results necessarily “strange” as they could be explained by the following rationale: WT cells have been cultivated in 2% glucose since the 1950s, likely fixing most beneficial mutations for this condition. When many isogenic strains are evolved in parallel, (a) some lines show no improvement due to the scarcity of available beneficial mutations, (b) others exhibit slight decreases in fitness due to genetic drift fixing deleterious mutations, and (c) a few lines discover rare beneficial mutations, leading to fitness increases. In contrast, other conditions represent “newer” environments with larger mutational target sizes, resulting in more consistent outcomes.

Prompted by the reviewer’s comment, we look for other studies reporting detailed fitness measurements of evolved WT strains in standard laboratory media. We downloaded and plotted the fitness data from Johnson et al. 2021, where authors studied the evolution of WT strains over 10,000 generations. Interestingly, we see that in the early phase of the evolution (generations 500-1400) evolved lines show similar levels of variability in fitness as the one reported in our study (Figure_rev_only 2). Of note is that in Johnson et al. 2021 most of the adaptive mutations alleviate the toxicity of the *ade2-1* allele. In our WT strain the gene was preemptively restored, further reducing the target size for adaptation in YPD.

We believe it is important to report these measurements and decided to leave the original data, with the appropriate quantifications of variability, in Figure 2.

Revision Plan

The authors propose from Figure S4B that the rescue of *ixr1*-null is less evident at low sugar concentration since both conditions trigger a reduction of dNTPs. I think this is interesting, since it would provide a link between glucose concentration and evolutionary trajectories to adaptation, which is what the authors wanted to study. In particular, one would predict that 0.25% glucose would see less *ixr1*-null than the other glucose conditions. I could not (was not able to) confute this hypothesis from the data shown in the paper. Likewise, for med14-H919P. If the authors have not tested it, it would be worth trying.

We had reported the appearance and frequency of all ‘core adaptive mutations’ (Figure S6C) but did not explicitly test the likelihood of their appearance under different glucose conditions. Following the reviewer’s suggestion, we have now performed χ^2 tests (on the presence or absence of mutations) and ANOVA tests (on their mean frequency) to determine whether any mutation is particularly enriched or depleted in a given glucose environment. At first glance, the results do not support the hypothesis proposed by the reviewer. However, we note that although *ixr1* mutants are less beneficial in low glucose than in high glucose, they still confer an 8% fitness advantage, which is likely sufficient to drive clones to fixation. We believe the reviewer’s reasoning is correct but is potentially masked by the still elevated fitness advantage of *ixr1* in low glucose.

To better convey the results of this analysis, we have included a visual representation of the presence and frequency of the mutations in Figure 6A, and the results of the χ^2 and ANOVA tests in Supplementary File 5. We also comment on the analysis as follows:

Line 314: “Similarly, we did not detect differences in the frequency of occurrence (χ^2 tests) or average fractions (ANOVA test) achieved by the mutations in the populations evolved under different glucose environments (Fig 6A, Fig S4C and Supplementary File 5. The presence of all mutations in the final evolved lines correlated with their fitness benefits, suggesting how their selection in all glucose conditions was mostly dictated by their relative fitness benefits, rather than the environment (Fig 6A).”

Minor points

Figures

- S4B: in the legend it should be explained that it is compared to *ctf4D*

We now report how the values were obtained in the figure legend:

($D = |\text{anc\%}| - |\text{reconstructed\%}|$)

-2A: the color code is not fully clear to me: what does green and blue indicate? higher and lower than 2%?

We apologise for not having included an explicit description of the color code in Figure 2A. Throughout the paper blue refers to glucose starvation (light blue for 0,25%, dark blue for 0,5%), while green refers to glucose abundance (light blue for 2%, dark blue for 8%). We now include a detailed description of the color code when it first appears (Fig 1B) and make sure it is properly reported in all figure legends.

- S3A: the authors should show the statistical difference between WT and *ctf4*-null, which is mentioned as non-existent in p.6

The p value is now represented in Fig S3A

- The med14-H919P mutation is observed in 22/48 wells. I guess the authors checked already: are some of these wells close to each other in the plate?

Revision Plan

Correct. We took significant precautions in our experimental design to prevent cross-contamination, as outlined in the Materials and Methods section. Specifically, rows of *ctf4Δ* samples were alternated with rows of WT samples. Daily dilutions were then performed row by row using a 12 channels pipette. This approach ensured that any potential carry-over of cells would result in them being placed in wells containing a different genotype, where they would be eliminated by the consistent use of genotype-specific drugs.

As a result of these measures, we do not observe any distinct pattern of core genetic adaptation corresponding to the plate layout (Figure_rev_only 3). The only exception are mutations in *IXRI*, which appear in all *ctf4Δ* strains (albeit with different alleles, see supplementary File 3). Moreover, we reasoned that if a highly fit strain had invaded other wells, all the pre-existing mutations from its lineage would have been detected in those wells. However, apart from the recurrent *ixr1* and *rad9* mutations, which are also strongly adaptive, we find no evidence of shared mutations in wells carrying the *med14-H919P* allele (Figure_rev_only 4).

- Compensatory evolution of *ctf4*-null in 2% glucose is the experiment published by Fumasoni and Murray in eLife. In that paper, there is no trace of mutations in *MED14*. I think the authors should comment on this (different method for detecting putative compensatory mutations?).

We also noticed the absence of *MED14* mutations in the eLife study by Fumasoni and Murray and find this discrepancy intriguing. One possible explanation lies in methodological differences. Our current study employed an improved version of the mutational analysis pipeline. However, we have not yet reanalyzed the original data from the previous study to determine whether *MED14* mutations were present but undetected.

Interestingly, in the current study, we observed that in 2% glucose, *MED14* mutations arose in only 3 out of 12 populations, a frequency lower than in other glucose conditions (Figure S6C). Assuming a similar frequency occurred in the 8 populations evolved in 2% glucose by Fumasoni and Murray (2020), one would expect only 2 populations to carry the mutation. This number falls below the threshold required for our algorithm to detect statistically significant parallelism.

Additionally, two significant experimental differences may also contribute to the observed discrepancy. First, the culture volumes and vessels differed: 10 mL cultures in tubes were used previously, whereas 1.5 mL cultures in 96-well plates were used in the current study.

- I may be mistaken, but Szamecz et al do not actually investigate whether different conditions result in different evolutionary trajectories (i.e., different genetics), and so their results may not be at odds with those presented here.

The reviewer is correct that Szamecz et al. do not explicitly test whether different conditions result in different evolutionary trajectories. However, in the section titled “Compensatory Evolution Generates Diverse Growth Phenotypes across Environments,” they examine how lines evolved in 2% YPD perform across various environments. They report how in roughly 50% of the cases tested, evolved lines showed either no improvement or even some lower fitness than the ancestor (Figure 5A).

While this could be explained by the accumulation of detrimental non-adaptive mutations in specific contexts, it likely implies that the adaptive strategies compensating for the original mutation in one environment do not confer similar benefits in other environments. This observation contrasts with our findings in Figure 6D, where we demonstrate that the main adaptive strategies provide a consistent benefit across diverse environments, including those with glucose, nitrogen, or phosphate abundance or starvation.

Revision Plan

We have now modified the introduction, results and discussion to avoid misleading interpretations:

Line 42: “Szamecz and colleagues examined the evolutionary trajectories of 180 haploid yeast gene deletions over 400 generations (Szamecz et al., 2014). They found that, while fitness recovery occurred in the environment where evolution took place, the evolved lines often showed no improvement over their ancestors in other environments. This suggests that compensatory mutations beneficial in one environment often fail to restore fitness in others.”

Line 327: “A previous study in yeast showed how evolved lines which compensate for detrimental defects of gene deletions in standard laboratory conditions often failed to show fitness benefits compared to their ancestor when tested in other environments (Szamecz et al., 2014). We thus investigated the extent to which the core genetic adaptation to DNA replication stress was beneficial under alternative nutrient conditions.”

Line 422: “What could explain the discrepancies between our results, and previous studies on evolutionary repair highlighting the role of the environment in shaping evolutionary trajectories (Filteau et al., 2015), and the heterogeneous behavior of evolved lines in various environments (Szamecz et al., 2014)?”

typos

p.18, line 564 preformed -> performed

p. 6 line 189 with a strongly skew -> with a strong skew ?

Typos are now corrected in the main text

REVIEWER #3

Major comments: N/A

Minor comments:

L19: I found the definition for compensatory evolution/mutations to be somewhat vague in the introduction (and subsequently throughout the text). It's clear that this was written for a more medical/physiological audience, but without a more explicit explanation of compensatory evolution/mutations, it became difficult to properly weigh some claims/discussions made by the authors later on. Do you define compensatory mutations as those which completely recover WT function/fitness, or are simply of opposite effect to the altered genotype? Others define "compensatory evolution" as simply any epistatically interacting amino acid substitutions (Ivankov et al, 2014). It would be nice to see more explicitly defined.

We thank the reviewer for highlighting the need for a precise definition of compensatory evolution and compensatory mutations. We recognize that the literature encompasses multiple definitions, including the one cited by the reviewer, which emphasizes compensatory mutations within the context of structural biology. This particular definition, prevalent in molecular evolution, was introduced by Kimura (Kimura, 1985) and is frequently used to explain the co-occurrence of amino acid mutations within a protein. These mutations offset each other's defects, restoring or maintaining protein function. Here, however, we are using an older and broader definition of compensatory mutation, first introduced by Wright (Wright, 1964, 1977, 1982) and frequently used in evolutionary genomics (e.g., Moore et al., 2000; Szamecz et al., 2014; Rajon and Mazel, 2013; Eckart et al., 2024). This definition includes any mutation in the rest of the genome that compensates (fully or partially) for another mutation's detrimental effects on fitness.

Revision Plan

We have now included this definition in the introduction:

Line 19: “Compensatory evolution is a process by which cells mitigate the negative fitness effects of persistent perturbations in cellular processes across generations. This adaptation occurs through spontaneously arising compensatory mutations anywhere in the genome (Wright, 1964, 1977, 1982) that partially or fully alleviate the negative fitness effects of perturbations (Moore et al., 2000). The successive accumulation of compensatory mutations over evolutionary timescales progressively repair the cellular defects, ultimately restoring fitness.”

Line 361: “Our findings demonstrate that while glucose availability significantly affects the physiology and adaptation speed of cells under replication stress, it does not alter the fundamental genome-wide compensatory mutations that drive fitness recovery and evolutionary repair.”

Along these lines, I would have liked to see a more direct comparison/discussion of the degree to which deletion lines recovered. I can see from Fig 2E and Fig S2B that fitness increased quite a bit; would it not be possible to include a figure on the degree of compensation (basically relative fitness of evolved deletion lines - relative fitness of ancestral deletion lines)?

If the reviewer is suggesting calculating the difference between the evolved and ancestor fitness, the data is already in Figure S2B and S2D, defined as ‘Absolute fitness gains Δ ’ and calculated as $\Delta = \text{evo\%} - \text{anc\%}$.

If instead is suggesting to plot the fitness of evolved deletion lines (Y axis) against the relative fitness of ancestral deletion lines (X axis), we have now produced the plot is Figure S2F.

To better understand the extent of the fitness recovery in *Ctf4* strains, we have also calculated and plotted the ‘relative fitness gain’ calculated as $|\text{evo\%}| / |\text{anc\%}| * 100$ (Figure S2C)

We are now commenting on these comparisons in the following paragraph:

Line 171: “By generation 1000, both WT and *ctf4* Δ evolved lines achieved, on average, slightly higher fitness in low glucose compared to high glucose conditions (Fig S2B). However, due to the varying initial fitness of *ctf4* Δ cells across different glucose environments, they recovered the same extent of the original defect (Fig S2C), displaying an opposite trend to WT, with increasing absolute fitness throughout the experiment as glucose concentration rose (Fig S2B vs S2D). The different absolute fitness gains over the same number of generations highlight distinct mean adaptation rates (Fig 2B). These differences are evident when examining the evolutionary dynamics of the evolved lines over time (Fig 2C). Additionally, we approximated the fitness trajectories using the power law function (Fig 2C, dashed purple lines), previously proposed to describe long-term evolutionary dynamics in constant environments (Wiser et al., 2013). The parameter *b* in this formula determines the curve's steepness, and can be used to quantify the global fitness change over generations (Fig S2E). Collectively, these analyses demonstrate that, unlike WT cells, *ctf4* Δ lines adapt faster in the presence of high glucose. This evidence aligns with the declining adaptability observed in other studies (Moore et al., 2000; Kryazhimskiy et al., 2014; Couce & Tenaillon, 2015), where low-fitness strains consistently adapt faster than their more fit counterparts (Fig S2F).”

L57: Another minor nitpick that just comes down to semantics. When discussing “96 parallel populations”, it invokes a higher sense of replication than is actually present in the study. I would rephrase this to something along the lines of “12 replicate populations across 8 treatments under conditions of [...]”.

We changed the sentence as follows:

Revision Plan

Line 66: “We evolved 96 parallel populations of budding yeast, organized into 12 replicate lines, across four conditions of glucose availability (from starvation to abundance) with or without replication stress.”

L185-187: The wording here needs to be clarified. Be explicit in that are examine the ratio (or count) of synonymous to non-synonymous mutations here, otherwise the interpretations appears to be direct contradiction to the (as written) results. Only after viewing the supplemental figure was I able to figure out what exactly was meant here.

We changed the sentence as follows:

Line 212: “We found no significant differences in the numbers of synonymous mutations detected in evolved populations in WT and *ctf4Δ* populations (Fig. S3A). These results support the hypothesis that replication stress in *ctf4Δ* lines favors the retention of beneficial mutations, rather than simply increasing the overall mutation rate.”

L349-350: The authors observe higher rates of adaptation in deletion lines than WT lines, and discuss this in adequate detail. Although not explicitly mentioned, this is consistent with a diminishing returns epistasis model (that could be beneficial to discuss, but is not necessary), which has been implicated in modulating the degree of repeatability observed along evolutionary trajectories (Wünsche et al. 2017). Although definitely not required for this already very nice manuscript, I think it would be very rewarding if the authors were to eventually analyze fine-scale dynamics of phenotypic and genomic adaptation to mine for these putative interactions and their influence on repeatability.

We agree with the reviewer on how our results align with a model of diminishing returns epistasis. This pattern is apparent not only between *ctf4Δ* and WT lines but also among *ctf4Δ* lines evolved in different glucose conditions. This phenomenon likely arises from the interaction of various adaptive mutations, which we aim to explore further in a dedicated manuscript. However, until we do so, we prefer to refer generally to a pattern of declining adaptability. To explicit this trend we have now included Fig S2F and commented on it in the manuscript:

Line 181: “This evidence aligns with the declining adaptability observed in other studies (Moore et al., 2000; Kryazhimskiy et al., 2014; Couce & Tenaillon, 2015), where low-fitness strains consistently adapt faster than their more fit counterparts (Fig S2F).”

Line 388: "Our results are consistent with declining adaptability, as evidenced by the reduced rates of adaptation observed both between *ctf4Δ* and WT lines and among *ctf4Δ* lines evolved in different glucose conditions (Fig S2F)"

REVIEWER #4

A *ctf4Δ* mutant strain was used to investigate the effects of replication stress. Why was this mutant chosen instead of other deletions that cause different types of replication stress?

We appreciate the opportunity to clarify our rationale for choosing the *ctf4Δ* mutant. The following are the main reasons why we believe *ctf4Δ* strains represent an ideal tool to study a global perturbation of the DNA replication program over evolutionary timescales:

- 1) General replication stress: The absence of Ctf4 perturbs replication fork progression, leading to a spectrum of replication stress-related phenotypes, including DNA damage sensitivity, single-stranded DNA gaps, reversed forks (Abe et al., 2018; Fumasoni et al., 2015), checkpoint activation (Poli et al., 2012), cell cycle delays (Miles and Formosa, 1992), increased recombination (Alvaro et al., 2007), and chromosome instability

Revision Plan

(Kouprina et al., 1992). This broad disruption makes it an excellent model for observing global perturbations in replication processes. In contrast, other mutants typically affect specific enzymatic (e.g., *POL32* and *RRM3*) or signaling (e.g., *MRC1*) functions, making them better suited to address specific questions.

- 2) Constitutive stress: Unlike drug-induced stress (e.g., Hydroxyurea; Krakoff et al., 1968) or conditional depletion systems (e.g., *GALI-POLE*; Zhang et al., 2022), which cells can easily circumvent through single mutations, *ctf4Δ* enforces persistent replication stress. Its deletion cannot be complemented by a single mutation, ensuring a robust and consistent stress environment for evolutionary studies.

We have now modified the main text to convey these advantages in a concise form:

Line 91: “In the absence of Ctf4, cells exhibit multiple defects commonly associated with DNA replication stress, such as single-stranded DNA gaps and altered replication forks (Fumasoni et al., 2015), leading to basal cell cycle checkpoint activation (Poli et al., 2012). These defects result in severe and persistent growth impairments, cell cycle delays, elevated nucleotides pools and chromosome instability (Miles and Formosa, 1992; Kouprina et al., 1992; Poli et al., 2012), making *ctf4Δ* mutants an ideal model for studying the cellular consequences of general and constitutive replication stress over evolutionary time.”

- The authors could be clearer that a (the?) cause of the *ctf4Δ* fitness defect is spurious upregulation of *RNR1*. I don't think it is mentioned until the Discussion, but it is highly relevant to Fig 4, and to the adaptations one would expect from *ctf4Δ*.

We thank the reviewer for the opportunity to clarify this aspect. We do not think that the fitness defects of *ctf4Δ* cells stem solely from the spurious upregulation of *RNR1*. However, we believe that a major aspect of the evolutionary adaptation is aimed at decreasing dNTP levels, potentially through different mechanisms. We are now mentioning increased dNTPs as major phenotype of *ctf4Δ* and commenting on the hypothesis more clearly in the discussion.

Line 93: “These defects result in severe and persistent growth impairments, cell cycle delays, elevated nucleotides pools and chromosome instability (Miles and Formosa, 1992; Kouprina et al., 1992; Poli et al., 2012)”

Line 409: “This condition will, in turn, be detrimental when proliferation rates are high (as in WT in high glucose) but beneficial under constitutive DNA replication stress (*ctf4Δ*), where cells experience spurious upregulation of dNTP production (Poli et al., 2012; Davidson et al., 2012).

- In Figure 1E, there is a very large spread in the relative fitness at 2% and 8% glucose, but this was not commented on. Is this heteroscedasticity expected?

The observed heteroscedasticity is expected. Our competition assays tend to exhibit increased variability when a strain approaches very low fitness levels. Specifically, as one strain nears extinction by the third day of competition, its abundance is estimated based on a much smaller number of events in the flow cytometer. Furthermore, we noticed a small number of reference cells carrying *pACT1-yCerulean* not showing strong fluorescence in 8% glucose. The nature of this effect is uncertain, and possibly linked to metabolism-linked changes in the cytoplasm. The combination of these two phenomena amplifies the impact of noise inherent to the methodology, leading to increased variability across replicates.

Nonetheless, the overall decreasing fitness trend across glucose conditions, combined with the statistical significance observed between high and low glucose levels, collectively convey a robust phenotype

Revision Plan

Minor comments:

- Competitions were performed between *ctf4Δ* strains and a constructed strain with yCerulean integrated at ACT1. Is the fitness of the fluorescent strain comparable to the ancestral wild-type strain (i.e., in a competition between the ancestral WT and the fluorescent strain, does either have an advantage)?

We noticed a slight disadvantage of the reference strain compare to WT, likely due to the costs of the extra fluorescence reporter. However, the disadvantage is minimal, ranging from -0.5 to -2.5 depending on the glucose environment (raw measurements are reported supplementary file 1, sheet 5). To take this into account, all fitness reported in figures are normalized for the WT value measured in the same environment line 613: “Relative fitness of the ancestral WT strain was used to normalize fitness across conditions.”

- In Figure 3, the legends for panels B and C appear to be swapped. Discussion of Figure 3 on pages 6 and 7 appear to reference the wrong panels.

We are unsure about this typo. Main text and figure legend seem to refer to the appropriate panels, 3B for mutation fractions and 3C for mutation counts. Perhaps the organization of the panels with B being under A instead of on its right confounds the reader?

- In Figure 4A and B, having the same colour scale between both heatmaps is misleading, as the scales are different. Consider having the same scale across both heatmaps so that enrichments are visually comparable.

Following the reviewer’s suggestion we have have chosen a uniform heatmap to visually represent GO terms enrichment in WT and *ctf4Δ* genetic backgrounds.

- In Figure 4C, having a legend in the figure for node size would be helpful to understand the actual number of populations with mutations in each gene.

A legend for node size has now being added next to Figure 4C.

4. Description of analyses that authors prefer not to carry out

Please include a point-by-point response explaining why some of the requested data or additional analyses might not be necessary or cannot be provided within the scope of a revision. This can be due to time or resource limitations or in case of disagreement about the necessity of such additional data given the scope of the study. Please leave empty if not applicable.

REVIEWER #1: N/A

REVIEWER #2: N/A

REVIEWER #3: N/A

REVIEWER #4

It's not clear from the study that the effects are generalizable to other forms of replication stress.

As with any method to induce DNA replication stress (including commonly used drugs like HU) each approach inevitably affects replication in a specific manner. Testing the broader applicability of our conclusions would require

Revision Plan

evolving additional strains with different replisome perturbations. For instance, mutations in *ELG1* and *CTF18* (affecting the alternative Replication Factor C), *POL30* (affecting the sliding clamp PCNA), *POL32* (affecting Pole), *RRM3* (protective helicase) and (*MRC1* (coordinating leading strand activities and signalling to the checkpoint) would have to be taken into account. Furthermore, specific mutant alleles of Ctf4 that disrupt interactions with particular binding partners (Such as *ctf4-4E* and *ctf4-3E*, perturbing the interaction with the CMG helicase and accessory factors respectively) will be highly informative on which specific aspects of the replication stress generated by the lack of Ctf4 each adaptive mutation alleviate.

However, accommodating such extensive variability would inflate the sample size to an extent that will become unfeasible within the experimental design focused on capturing parallel evolution over a nutrient gradient (the primary focus of this study). We agree that this is an important question and intend to address it comprehensively in a dedicated future study.

- The authors comment that the *med14-H919P* mutant could have implications for the stability of Med14, based on computational modelling. Verifying the stability of the *med14-H919P* in vivo would strengthen this discussion.

We believe that in vivo and in vitro structural studies investigating the effect of this mutation on the stability and function of the Mediator complex are beyond the scope of this manuscript. These investigations would be more appropriately addressed in future, dedicated studies focused on these specific aspects.

21st Jan 2025

Manuscript Number: MSB-2025-12869-T

Title: Compensatory Evolution to DNA Replication Stress is Robust to Nutrient Availability

Author: Mariana Natalino

Marco Fumasoni

Dear Dr. Fumasoni,

Thank you again for submitting your work to Molecular Systems Biology. We have now discussed your manuscript, the reviews from Review Commons, and your revision plan to address the remaining reviewer comments. We are happy to let you know that with the addition of the proposed revisions, we would send your paper back to the original reviewers who reviewed your paper at Review Commons for re-review at Molecular Systems Biology (given resubmission in a reasonable timeframe).

When submitting your revised manuscript, please carefully review the instructions that follow below. We perform an initial quality control of all revised manuscripts before re-review; failure to include requested items may delay the evaluation of your revision.

We require:

1) A .docx formatted version of the manuscript text (including legends for main figures, EV figures and tables). Please make sure that the changes are highlighted to be clearly visible. Alternatively you may choose to submit your manuscript as a LaTeX file.

4) A .docx formatted letter INCLUDING the reviewers' reports and your detailed point-by-point responses to their comments. As part of the EMBO Press transparent editorial process, the point-by-point response is part of the Peer Review File (PRF), which will be published alongside your paper.

5) A complete author checklist, which you can download from our author guidelines (<https://www.embopress.org/page/journal/17574684/authorguide#submissionofrevisions>). Please insert information in the checklist that is also reflected in the manuscript. The completed author checklist will also be part of the PRF.

6) Please note that all corresponding authors are required to supply an ORCID ID for their name upon submission of a revised manuscript.

7) It is mandatory to include a 'Data Availability' section after the Materials and Methods. Before submitting your revision, primary datasets produced in this study need to be deposited in an appropriate public database, and the accession numbers and database listed under 'Data Availability'. Please remember to provide a reviewer password if the datasets are not yet public (see <https://www.embopress.org/page/journal/17574684/authorguide#dataavailability>).

In case you have no data that requires deposition in a public database, please state so in this section as follows: "This study includes no data deposited in external repositories". Note that the Data Availability Section is restricted to new primary data that are part of this study.

8) All Materials and Methods need to be described in the main text using our 'Structured Methods' format, which is required for all research articles. According to this format, the Methods section includes a Reagents and Tools Table (listing key reagents, experimental models, software and relevant equipment and including their sources and relevant identifiers) followed by a Methods and Protocols section describing the methods using a step-by-step protocol format. The aim is to facilitate adoption of the methodologies across labs. Please upload the Reagents and Tools table as a separate document when submitting your revised manuscript. More information on how to adhere to this format as well as a downloadable template (.docx) for the Reagents and Tools Table can be found in our author guidelines:

<https://www.embopress.org/page/journal/17444292/authorguide#structuredmethods>

9) For data quantification: please specify the name of the statistical test used to generate error bars and p-values, the number (n) of independent experiments (specify technical or biological replicates) underlying each data point and the test used to calculate p-values in each figure legend. The figure legends should contain a basic description of n, p-values and the test applied. Graphs must include a description of the bars and the error bars (s.d., s.e.m.). Please provide exact p-values (in either the figure or figure legend).

10) Our journal encourages inclusion of *data citations in the reference list* to directly cite datasets that were re-used and obtained from public databases. Data citations in the article text are distinct from normal bibliographical citations and should directly link to the database records from which the data can be accessed. In the main text, data citations are formatted as follows: "Data ref: Smith et al, 2001" or "Data ref: NCBI Sequence Read Archive PRJNA342805, 2017". In the Reference list, data citations must be labeled with "[DATASET]". A data reference must provide the database name, accession number/identifiers and a resolvable link to the landing page from which the data can be accessed at the end of the reference. Further instructions are available at .

11) We replaced Supplementary Information with Expanded View (EV) Figures and Tables that are collapsible/expandable online. EV Figures should be cited as 'Figure EV1, Figure EV2' etc... in the text and their respective legends should be included in the main text after the legends of regular figures.

- Additional Tables/Datasets should be labeled and referred to as Table EV1, Dataset EV1, etc. Legends should be provided in a separate tab in case of .xls files. Alternatively, the legend can be supplied as a separate text file (README) and zipped together with the Table/Dataset file.

<https://www.embopress.org/page/journal/17574684/authorguide#expandedview>

12) Author contributions: CRedit has replaced the traditional author contributions section because it offers a systematic machine-readable author contributions format that allows for more effective research assessment. Please remove the Authors Contributions from the manuscript and use the free text boxes beneath each contributing author's name in our system to add specific details on the author's contribution. More information is available in our guide to authors.

13) Disclosure statement and competing interests: We updated our journal's competing interests policy in January 2022 and request authors to consider both actual and perceived competing interests. Please review the policy <https://www.embopress.org/competing-interests> and update your competing interests if necessary.

14) Every published paper now includes a 'Synopsis' to further enhance discoverability. Synopses are displayed on the journal webpage and are freely accessible to all readers. They include a short stand first (maximum of 300 characters, including space) as well as 2-5 one-sentences bullet points that summarizes the paper. Please write the bullet points to summarize the key NEW findings. They should be designed to be complementary to the abstract - i.e. not repeat the same text. We encourage inclusion of key acronyms and quantitative information (maximum of 30 words / bullet point). Please use the passive voice. Please attach these in a separate file or send them by email, we will incorporate them accordingly.

Please note that these would be the final versions and changes during proofing are usually not allowed.

15) As part of the EMBO Publications transparent editorial process initiative (see our policy here:

https://www.embopress.org/transparent-process#Review_Process), Molecular Systems Biology will publish online a Peer Review File (PRF) to accompany accepted manuscripts.

In the event of acceptance, this file will be published in conjunction with your paper and will include the anonymous referee reports, your point-by-point response and all pertinent correspondence relating to the manuscript. Let us know whether you agree with the publication of the PRF and as here, if you want to remove or not any figures from it prior to publication.

Please note that the Author checklist will be published at the end of the PRF.

Molecular Systems Biology has a "scooping protection" policy, whereby similar findings that are published by others during review or revision are not a criterion for rejection. Should you decide to submit a revised version, I do ask that you get in touch after three months if you have not completed it, to update us on the status.

If you have any questions, please do not hesitate to ask. I look forward to receiving your revised manuscript.

Yours sincerely,

Poonam Bheda

Poonam Bheda, PhD
Scientific Editor
Molecular Systems Biology

Rev_Com_number: RC-2024-02762

New_manu_number: MSB-2025-12869-T

Corr_author: Fumasoni

Title: Compensatory Evolution to DNA Replication Stress is Robust to Nutrient Availability

Dear Molecular Systems Biology editorial team,

We thank Review Commons for coordinating the thorough and careful review of our manuscript. We are especially grateful to the four anonymous reviewers for recognizing the value of our work and for their constructive suggestions on how to improve it.

We are encouraged by the positive reception of our main conclusions on the robustness of adaptation to DNA replication stress and its relevance to multiple fields. All reviewers provided insightful comments, with reviewers #2 and #4 emphasizing that further experimental validation of the hypothesized role of reduced dNTPs in alleviating fitness during constitutive DNA replication stress would strengthen the paper. While the precise molecular mechanisms underlying this suppression are not the primary focus of this manuscript, we were eager to perform additional experiments based on the reviewers' suggestions.

In fact, the proposed experiments made us reconsider our hypothesis on the role of *med14-H919P* in alleviating DNA replication stress, which we now attribute to more nuanced pleiotropic effects on transcription and cohesion. We believe these revisions, along with the new data and refined analyses, have strengthened our study and its contributions to the field.

Reviewer #1

This study investigates the compensatory evolutionary response of *Saccharomyces cerevisiae* to DNA replication stress, focusing on the influence of genotype-environment interactions (GXE). The authors used a range of experimental conditions with varying nutrient levels to assess evolutionary outcomes under replication stress. Their genomic analysis reveals that while glucose levels affect initial adaptation rates, the genetics of adaptation remain robust across all nutritional environments. The research offers new insights into the adaptability of *S. cerevisiae*, emphasizing the role of the nutritional environment in evolutionary processes related to DNA replication stress. It identifies recurrent advantageous mutations under different macronutrient availabilities and uncovers a novel role for the RNA polymerase II mediator complex in adaptation to replication stress. Overall, this well-designed study adds to the growing recognition of the complexity and robustness of evolutionary responses to environmental stressors. It provides strong evidence that compensatory evolution to replication stress is robust across varying nutritional conditions. It both challenges and reinforces previous findings regarding the resilience of the yeast genetic interaction network to environmental perturbations. The detailed analysis of specific compensatory mutations and their fitness impacts across different conditions offers valuable insights into adaptive dynamics over 1000 generations, contributing a clear empirical framework for understanding how replication-associated stress shapes evolutionary outcomes in diverse environments.

Based on the analysis:

- 1) The conclusions are generally well-supported by the presented data. The evolution experiments and genomic analyses are robust and provide convincing evidence for the study's main claims. The authors took steps to eliminate bias, such as maintaining an adequate N_e , which, if not done, could have compromised their conclusions by affecting genetic drift and limiting the population's access to beneficial mutations.
- 2) The figures are well-designed and easy to understand.
- 3) The methodology is well-described and appears reproducible. The authors provide sufficient details on experimental procedures. Experimental replication is adequate, with multiple evolutionary lines.

4) They also made efforts to validate their observations, such as the validation of mutations, the prediction of interactions in the Med14 structure, and its potential implication in gene regulation, as well as the analysis of the cumulative fitness benefit and the reconstruction of the quadruple mutant.

There are, however, a few results that would benefit from further clarification:

1) The experimental design is strong, offering a diverse range of conditions. However, the high glucose condition (8%) stands out as significantly different from the neutral 2% condition, both in range and margin, compared to the low glucose conditions (0.25-0.5%). While this mainly affects growth profiles and evolvability in the early generations, a brief explanation in the discussion would strengthen the conclusions. Specifically, addressing:

- a) The rationale behind selecting these particular glucose concentrations.
- b) How other glucose concentrations might influence the outcomes. Providing this additional context would enhance the reader's understanding of the experimental setup and its potential implications, while also offering insights into the broader applicability of the findings and possible directions for future research.

We thank the reviewer for pointing out the need to clarify the rationale behind the glucose concentrations used in our study, an aspect we agree should have been better explained. In response, we have added the following text detailing the chosen conditions and their established effects on cellular metabolism.

Line 82: “Glucose is the most abundant monosaccharide in nature, and represents the preferred source of energy for most cells.”

Line 125: “...we grew WT and *ctf4Δ* cells in varying glucose concentrations to induce distinct physiological states. Low glucose levels (0.25% and 0.5%) induce caloric restriction and ultimately glucose starvation (Lin et al, 2000; Smith et al, 2009). These conditions elicit increased respiration (Lin et al, 2002), sirtuins expression (Guarente, 2013), autophagy (Bagherniya et al, 2018), DNA repair (Heydari et al, 2007), and reduced recombination at the ribosomal DNA locus (Riesen & Morgan, 2009) ultimately extending lifespan in several organisms (Kapahi et al, 2017). In contrast, standard laboratory conditions typically use 2% glucose, promoting a rapid proliferation environment to which strains have been adapted since laboratory domestication (Lindgren, 1949). Finally, elevated glucose concentrations (such as 8%) result in higher ethanol production (Lin et al, 2012) and reactive oxygen species (ROS) levels (Maslanka et al, 2017).

2) In the discussion section, a more explicit comparison with similar studies in other model organisms would help contextualize the findings within the broader field of evolutionary biology. While the results appear robust, it would be beneficial to explore how they align with or contrast to previous studies on DNA damage, particularly in bacteria or highly complex eukaryotes.

We appreciate this suggestion to better contextualize our findings within the broader literature, as it provides an opportunity to highlight the unique aspects of our work. While many studies have explored how environmental factors shape fitness landscapes and influence evolutionary strategies, to our knowledge, only a few have addressed this in the context of compensatory evolution, where cells must recover fitness lost due to intracellular perturbations. To address this point, we have added a discussion of additional examples involving other model organisms, highlighting their difference with the question asked in this work.

Line 50: “Genotype-by-environment (GxE) interactions are well-documented. For example, several studies on *E. coli* have demonstrated how different environments influence fitness and epistatic interactions among adaptive mutations in the Lenski Long-Term Evolution Experiment (Ostrowski et al., 2005, 2008; Flynn et al., 2012; Hall et al., 2019). Adaptive mutations in viral genomes similarly exhibit variable fitness effects across different hosts (Lalic and Elena, 2012; Cervera, 2016). Furthermore, interactions between mutations in the *Plasmodium falciparum* dihydrofolate reductase gene have been shown to predict distinct patterns of resistance to antimalarial drugs (Ogbunugafor et al., 2016). However, the role of environmental factors in shaping evolution within the context of compensatory adaptation, when fitness defects primarily arise from intracellular perturbations, remains much less explored.”

However, if the reviewer have particular additional studies in mind, we welcome further suggestions to include in the final manuscript.

Minor comments:

1) The presentation of data in the figures is clear and informative. However, some figure legends could benefit from more detailed explanations. For example, although the statistical tests used are mentioned in the methods section, it would be helpful to also include them in the figure legends, such as in legend 1acde, as well as in all other figures.

We are now reporting the statistical test used for each comparison also in figure legends.

2) In terms of broader conclusions, here are a few suggestions, though they are, of course, optional:

- a) The study could benefit from exploring the potential trade-offs of adaptive mutations in the hypothetical return to environments without replication stress, at least theoretically. This would provide a more comprehensive understanding of the evolutionary constraints.

We thank the reviewer for the suggestion, we had performed the measurements but did not comment on them explicitly. We are now commenting on them as follows:

Line 340: “In the WT background, all mutations were nearly neutral, with only minimal deleterious or advantageous effects on fitness depending on glucose concentrations (Fig EV5A).”

Line 500: “The nearly neutral effects on fitness of the core adaptive mutations in WT suggest that they are likely to persist even after the initial replication stress is resolved.”

- b) A brief discussion of the potential limitations of using lab strains versus wild isolates of *S. cerevisiae* would offer valuable context for the generalizability of the findings.

This is an excellent point. While addressing it fully would warrant a separate manuscript, we provide our comments here, along with similar observations raised by this and other reviewers, as follows:

Line 482: “How generalizable are our conclusions about the reproducibility of evolutionary repair to DNA replication stress across other organisms, species, or replication challenges? While dedicated future studies are needed to fully address these important questions, several lines of evidence are encouraging. A recent report demonstrated that the identity of suppressor mutations of lethal alleles was conserved when introduced into highly divergent wild yeast isolates (Paltenghi and van Leeuwen, 2024). Similarly, earlier work showed that even ploidy, which significantly alters the target size for loss- and gain-of-function mutations, affected only the identity of the

genes targeted by selection, while the broader cellular modules involved remained consistent (Fumasoni and Murray, 2021). Moreover, divergent organisms experiencing different types of DNA replication stress exhibit some of the adaptive responses described here. For example, the yeast genus *Hanseniaspora*, which lacks the Pol32 subunit of the replisome, has also been reported to have lost the DNA damage checkpoint (Steenwyk et al., 2019). Human Ewing sarcoma cells carrying the fusion oncogene *EWS-FLI1* frequently exhibit adaptive amplification of the cohesin subunit *RAD21* (Su et al., 2021). Together, these findings suggest that while the specific details of DNA replication perturbations and the genomic features of organisms may shape the precise targets of compensatory evolution, the overarching principles and cellular modules affected are broadly conserved.”

- c) It would be valuable to present the differences in ploidy in the context of other studies, such as the nutrient-limitation hypothesis (e.g., 'The Evolutionary Advantage of Haploid Versus Diploid Microbes in Nutrient-Poor Environments' by Bessho, 2015), since, as previously demonstrated by the authors of this article that is being reviewed, ploidy may influence the evolutionary trajectories of DNA repair.
- d) Interrelating these three terms: nutrient-limitation, ploidy, and DNA repair could be an interesting avenue to explore in the discussion.

In response to comments c and d, we have now commented on the intersection between ploidy and other types of DNA perturbation in the paragraph starting in line 482 (see response above)

3) Specific details:

- a) Line 116: To improve clarity, it would be beneficial to refer to the figure right after the statement: 'However, their relative fitness improved compared to the WT reference as the initial glucose levels (Figure X).'
- b) Line 404: The statement about antibiotics and cancer progression is somewhat brief here; it might be helpful to provide more context on why this mechanism influences these processes (here or before).
- c) Line 418: "were re-suspended in water containing zymolyase (Zymo Research, Irvine, CA, US, 0.025 μL), incubated at". Something is missing in the units.
- d) Line 459: "and G2 phases for each genotype was estimated by deriving the the relative cell distribution". The article "the" is repeated.
- e) Fig 1a: The x-axis ticks appear misaligned, which makes it difficult to interpret the boxplots. For example, at 0.25, the tick is closer to the orange boxplot than to the black one. In contrast, at 2%, the tick seems well-centered."
- f) Figure 3 could benefit from a general legend at the top regarding the colors, as finding it in 2c was not intuitively easy.

The typos and suggestions raised in points 3a-f have now been corrected in the manuscript.

- g) I didn't review the code on GitHub.

Reviewer #1 (Significance (Required)):

The main strength of the study is that it shows robustness of compensatory evolution across varying nutrient conditions. The study adds to the growing body of literature on DNA replication stress and evolutionary adaptation by showing that compensatory evolution can occur regardless of nutrient availability. This fundamental finding

challenges prior assumptions that nutrient conditions significantly alter evolutionary outcomes, contributing to a more nuanced understanding of how cells respond to stress. Furthermore, the discovery of the RNA polymerase II mediator complex's role in this process is particularly novel and opens new lines of investigation.

Advance in the field: The results advance our understanding of evolutionary biology, particularly in the context of DNA replication stress and compensatory evolution. The study demonstrates that evolutionary repair mechanisms are predictable, even under variable environmental conditions, which has key implications for evolutionary biology and therapeutic applications.

Audience:

This paper will be of interest to a specialized audience in evolutionary biology, genomics, and cell biology, particularly those interested in DNA replication stress and adaptive evolution. Researchers studying stress responses in model organisms, such as *S. cerevisiae*, will find the findings valuable, as will those working in applied fields where stress adaptation is a critical factor (e.g., industrial yeast fermentation, drug development, disease resistance, cancer research, or aging studies).

Expertise:

Evolutionary biology, genomic analysis, and cellular stress responses, with a particular focus on experimental evolution under DNA damage stress in *Saccharomyces cerevisiae*. Recently graduated and beginner reviewer.

Reviewer #2

The paper addresses the effect of sugar availability in shaping compensatory evolution. The first observation of the paper is that cell physiology changes by modulating glucose availability also in strains that come with defective DNA replication (*ctf4*-null previously studied by the authors). An intriguing result is that *ctf4*-null grows comparatively better in low concentrations of glucose. This is hypothesized to be a consequence of both the decrease in dNTPs in low glucose, which causes slow down of fork progression, and/or reduced fork collapse at rDNA locus. Hence, wild types and *ctf4*-null show an opposite trend: in the mutant, the lowest concentration of glucose is the least affected by the mutation; in wild type, the highest concentration is the least affected. Adaptation rate is inversely related with the initial fitness. The effect on physiology and adaptation rate is a starting point for asking the key question: are evolutionary trajectories influenced by the growth conditions? The answer is negative: evolution experiments show the very same core of genetic changes at all sugar concentrations. The result is apparently at odds with previous publications, and the authors conclude that, in this particular setting, availability of carbon sources plays a minor role compared to impaired DNA replication. The different rates of adaptation in WT and mutant is rather explained by the initial fitness at the different glucose concentrations, which, as mentioned, is opposite in WT and *ctf4*-null mutants. The paper also reports a new mutation in *MED14*, component of the transcription mediator complex, which rescues the lack of Ctf4 activity. The study is interesting and asks a relevant question. The experiments are well executed and convincing, but the paper can be strengthened by testing some of the hypotheses which are put forward.

Main points

1- The raw data for evolutionary dynamics (Figure S2C) are fitted with the power law suggested by Wisner and Lenski, and return different values of the parameter 'b'. The authors say that the result depends greatly on the initial conditions ("due to the varying initial fitness of *ctf4*Δ cells across different glucose environments, they display an opposite trend to WT"). Around the initial values, however, the curves are non-monotonic, especially for low glucose availability. Both for WT and *ctf4*-null there is an initial drop in fitness, after which fitness increases. If one would neglect this initial dynamics, the value of the parameter 'b' would likely be different.

The non-monotonic trend in fitness highlighted by the reviewer is likely due to technical factors: Fitness at Generation 0 was measured with high precision in a low-throughput manner early in the project. In contrast, fitness from Generation 100 to 1000 was measured later in the study in a high-throughput fashion, necessitated by the large number of competitions conducted (96 wells \times 4 time points \times 6 replicates = 2304 assays). This difference in methodologies may have introduced a slight offset when the datasets were combined at Generation 100. Following the reviewer's suggestion, we have excluded the data point at Generation 100 responsible for this non-monotonic behavior and re-fitted the curves. While this adjustment has caused minor changes in the parameter 'b', the qualitative trends, particularly the opposing trends between WT and *ctf4Δ* as glucose increases, remain consistent (Figure_rev_only 1). To ensure transparency, we have retained all recorded fitness values in the original figure for reference.

In general, one can question whether curves with this shape are best fitted by the power law proposed by Wisner and Lenski. For example, for the WT 0.25% glucose the linear fit gives a better R² (why do the authors show the linear fit anyway?). This impression is further reinforced by the observation that Wisner and Lenski fit dynamics that last 50,000 generations, here the curves last 1/50th of it. In conclusion, I would question whether the parameter 'b' is a solid measurement of 'rate of adaptation'. Also, normalizations makes it difficult to appreciate the result shown in Figure 2B. I think the authors should look for a different way to show the different trend in adaptation dynamics for different glucose concentrations between wild types and mutants. For example, they could move Figure S2C in the main text to stress the result shown in Figure 2C, which already shows the difference between WT and mutant. This is especially true if what Figure 2C shows is (evo-anc)/evo. This is not fully clear to me: in the legend it refers to the delta, in the label of the y-axis I read that this is a percentage.

We thank the reviewer for prompting us to clarify our methods for reporting fitness changes over time. The fitness values are reported, throughout the paper, as a percentage change relative to the reference WT strain. The gain in fitness during evolution (reported as Δ) represents the difference between the evolved strain (evo%) and the ancestral strain (anc%), calculated as $\Delta = \text{evo}\% - \text{anc}\%$. This represents the absolute gain, rather than the relative gain. This value is still reported as a percentage as it's the same scale and unit as the two values being subtracted. We have included additional details to clarify this aspect in the figure legend of now figEV2C.

“(D) Absolute fitness gains (Δ) at generation 1000 for evolved WT (upper panel, black) and *ctf4Δ* (lower panel, orange) populations. Box plots show median, IQR, and whiskers extending to 1.5 \times IQR, with individual data points beyond whiskers considered outliers. Absolute fitness gains were calculated by subtracting the ancestral relative fitness from the relative fitness of the evolved ($\Delta = \text{evo}\% - \text{anc}\%$), both calculated as percentages relative to the same reference strain in the same glucose concentration.”

To conclude: the data show a different trend between wild types and mutants, which is interesting. Fitting it with the power law seems to be neither required nor appropriate. I suggest the authors to show the WT vs mutant pattern differently.

We followed the reviewer's suggestion and moved Figure S2C, which depicts the detailed fitness trajectories over time, into the main manuscript as Figure 2D. We agree that presenting these trajectories alongside the absolute fitness gains (now in Figure S2C) provides a more intuitive and effective depiction of the evolutionary dynamics of WT and *ctf4Δ* strains without relying solely on the power-law fit. Additionally, we quantified the *mean* adaptation rate, calculated as the absolute fitness gain (Δ) divided by the total number of generations (now Figure 2B). While no individual method definitively captures the adaptation rates across the experiment, these complementary analyses consistently highlight the same trends noted by the reviewer. We have re-written the main text as follows:

Line 185: “By generation 1000, both WT and *ctf4Δ* evolved lines achieved, on average, slightly higher fitness in low glucose compared to high glucose conditions (Fig EV2B). However, due to the varying initial fitness of *ctf4Δ* cells across different glucose environments, the proportion of the fitness defect recovered remained constant across conditions (Fig EV2C). *ctf4Δ* lines thus displayed an opposite trend to WT, with increasing absolute fitness throughout the experiment as glucose concentration rose (Fig EV2B vs S2D). The different absolute fitness gains over the same number of generations highlight distinct mean adaptation rates (Fig 2B). These differences are evident when examining the evolutionary dynamics of the evolved lines over time (Fig 2C). Additionally, we approximated the fitness trajectories using the power law function (Fig 2C, dashed purple lines), previously proposed to describe long-term evolutionary dynamics in constant environments (Wiser et al., 2013). The parameter *b* in this formula determines the curve's steepness, and can be used to quantify the global adaptation rate over generations (Fig EV2E). Collectively, these analyses demonstrate that, unlike WT cells, *ctf4Δ* lines adapt faster in the presence of high glucose. This evidence aligns with the declining adaptability observed in other studies (Moore et al., 2000; Kryazhimskiy et al., 2014; Couce & Tenaillon, 2015), where low-fitness strains consistently adapt faster than their more fit counterparts (Fig EV2F).”

Overall, these results demonstrate that cells can recover from fitness defects caused by constitutive DNA replication stress regardless of the glucose environment. However, adaptation rates under DNA replication stress exhibit opposing trends compared to WT cells, with faster adaptation yielding greater fitness gains in higher glucose conditions.”

2- In Figure S2C, the individual trajectories for WT at 2% glucose are strangely variable. In this case, plotting the average does not make too much sense. This result is strange, since this is the default condition, where cells are grown without any change of sugar concentration. Can the authors give any rationale? Are there other available results to replace those published in Figure S2C?

We agree with the reviewer that the individual trajectories for WT at 2% glucose are intriguing. However, we do not find these results necessarily “strange” as they could be explained by the following rationale: WT cells have been cultivated in 2% glucose since the 1950s, likely fixing most beneficial mutations for this condition. When many isogenic strains are evolved in parallel, (a) some lines show no improvement due to the scarcity of available beneficial mutations, (b) others exhibit slight decreases in fitness due to genetic drift fixing deleterious mutations, and (c) a few lines discover rare beneficial mutations, leading to fitness increases. In contrast, other conditions represent “newer” environments with larger mutational target sizes, resulting in more consistent outcomes.

Prompted by the reviewer’s comment, we look for other studies reporting detailed fitness measurements of evolved WT strains in standard laboratory media. We downloaded and plotted the fitness data from Johnson et al. 2021, where authors studied the evolution of WT strains over 10,000 generations. Interestingly, we see that in the early phase of the evolution (generations 500-1400) evolved lines show similar levels of variability in fitness as the one reported in our study (Figure_rev_only 2). Of note is that in Johnson et al. 2021 most of the adaptive mutations alleviate the toxicity of the *ade2-1* allele. In our strains the gene was preemptively restored, further reducing the target size for adaptation in YPD.

We believe it is important to report these measurements and decided to leave the original data, with the appropriate quantifications of variability, in Figure 2.

3- The molecular explanation given for the rescue of *ctf4*-null proposes a very relevant role for dNTPs downregulation. Particularly, both for *Irx1* and *med14-H919P*, the authors propose that this happens via *Rnr1* downregulation. At this stage, this is only a hypothesis. The molecular verification of the central role of *Rnr1* downregulation would make the conclusion much stronger. For example, a preliminary test would imply that

duplicating *RNR1* in *ctf4*-null *ixr1*-null and/or *ctf4*-null *med14-H919P* would revert the rescue. Any other experiment addressing this point would be useful to improve the paper.

Following the reviewer's suggestion, we conducted additional experiments to investigate the role of dNTP levels in *ctf4Δ* cells. Overexpression of the mutant allele *rnr1-D657N*, refractory to feedback inhibition, led to a reduction in fitness in *ctf4Δ* cells (Fig EV4B). Similarly, previous work by Fumasoni and Murray showed that while overexpression of the Rnr1 inhibitor *SML1* was beneficial in *ctf4Δ* cells, its repression was detrimental (Fumasoni and Murray, 2020, Figure 4—figure supplement 4, panel B). Furthermore, the fitness benefits associated with *IXR1* deletion, previously reported to reduce dNTP pools, were abolished by *rnr1-D657N* overexpression (Fig EV4B).

Together, these results support the idea that reduced dNTP levels mitigate the deleterious effects of *CTF4* loss and that adaptive mutations in *IXR1* contribute to this compensation.

We have now revised the following text in the manuscript:

Line 408: “To substantiate this hypothesis, we show that overexpression of an *RNR1* allele refractory to feedback inhibition (*rnr1-D57N*) (Chabes et al, 2003; Chabes & Stillman, 2007) reduces fitness in *ctf4Δ* cells and abolishes the fitness benefits of *IXR1* deletion in a *ctf4Δ* background (Fig EV4B).”

Regarding the role of *med14-H919P*, our experiments performed in response to the reviewers do not support anymore its involvement in reducing dNTP levels. Accordingly, we have revised our interpretation in the manuscript (see a more detailed discussion in response to a comment below).

4- The authors propose from Figure S4B that the rescue of *ixr1*-null is less evident at low sugar concentration since both conditions trigger a reduction of dNTPs. I think this is interesting, since it would provide a link between glucose concentration and evolutionary trajectories to adaptation, which is what the authors wanted to study. In particular, one would predict that 0.25% glucose would see less *ixr1*-null than the other glucose conditions. I could not (was not able to) confute this hypothesis from the data shown in the paper. Likewise, for *med14-H919P*. If the authors have not tested it, it would be worth trying.

We had reported the appearance and frequency of all ‘core adaptive mutations’ (Figure S6C) but did not explicitly test the likelihood of their appearance under different glucose conditions. Following the reviewer's suggestion, we have now performed χ^2 tests (on the presence or absence of mutations) and ANOVA tests (on their mean frequency) to determine whether any mutation is particularly enriched or depleted in a given glucose environment. At first glance, the results do not support the hypothesis proposed by the reviewer. However, we note that although *ixr1* mutants are less beneficial in low glucose than in high glucose, they still confer an 8% fitness advantage, which is likely sufficient to drive clones to fixation. We believe the reviewer's reasoning is correct but is potentially masked by the still elevated fitness advantage of *ixr1* in low glucose.

To better convey the results of this analysis, we have included a visual representation of the presence and frequency of the mutations in Figure 6A, and the results of the χ^2 and ANOVA tests in Source Data. We also comment on the analysis as follows:

Line 314: “Similarly, we did not detect differences in the frequency of occurrence (χ^2 tests) or average fractions (ANOVA test) achieved by the mutations in the populations evolved under different glucose environments (Fig 6A, Fig EV4C and Source Data. The presence of all mutations in the final evolved lines correlated with their fitness benefits, suggesting how their selection in all glucose conditions was mostly dictated by their relative fitness benefits, rather than the environment (Fig 6B).”

5- The combination of the four genetic adaptation (Fig 6B) would benefit from an experimental verification to show that the different solutions are not mutually exclusive. This is not obvious: if more than one solution acts by reducing dNTPs, maybe their combined effect is less strong than what measured theoretically. The authors could derive some clones at the end of the experiment and Sanger sequencing some of the four genes, to confirm the co-presence of some of them in the same cell.

The co-occurrence of nearly every combination of the four core adaptive mutations we identified can be inferred from their relative frequencies, as revealed by deep whole-genome sequencing of the evolved populations (Fig EV4C). In these data, we observe populations carrying each pairwise combination of mutations at frequencies exceeding 50%, implying their coexistence. Moreover, many combinations of mutations approach or reach fixation. A particularly striking example is *ctf4Δ* Population 11, evolved in 8% glucose, where all core adaptive mutations are present at 100% frequency. These findings provide robust evidence that the different adaptive solutions are not mutually exclusive and can coexist within the same genetic background.

Following the reviewer's suggestion, we generated a strain carrying the four core adaptive mutations in a *ctf4Δ* background. In Figure 6C, we present the fitness of this strain (referred to as "core reconstructed," marked by a yellow star) across all glucose conditions. Notably, in every condition, its fitness is comparable to, or higher, than that of the final evolved populations (One-side t-test). For example, in 8% glucose, the observed fitness of the core reconstructed strain is ~ -6% while the predicted value / computed for a population with all four mutations was ~ -5%, indicating that the four core adaptive mutations are nearly additive.

We have commented on these experiments in the main text:

Line 351: "To address this question, we engineered a quintuple mutant carrying all alleles that mimic the core adaptive mutations in a *ctf4Δ* background. Across glucose concentrations, the fitness of this reconstructed strain closely approximated that of the final evolved populations. We also computed the cumulative fitness benefit of the core adaptive mutations present in each evolved population (Fig 6C). For instance, in glucose 8%, population 11, where all four core mutations had fixed, exhibited an observed and computed fitness of ~ -8% and ~ -5%, respectively. Under the same condition, the core reconstructed strain showed a fitness of ~ -6%. Overall, these results highlight the robustness of core genetic adaptations to DNA replication stress. The four recurrently selected mutations provide significant fitness benefits across conditions, and their nearly additive effects largely account for the fitness gains observed in populations evolved over 1,000 generations."

Minor points

Figures

- S4B: in the legend it should be explained that it is compared to *ctf4Δ*

We now report how the values were obtained in the figure legend:

(D = |anc%| - |reconstructed%|)

-2A: the color code is not fully clear to me: what does green and blue indicate? higher and lower than 2%?

We apologize for not having included an explicit description of the color code in Figure 2A. Throughout the paper, blue refers to glucose starvation (light blue for 0,25%, dark blue for 0,5%), while green refers to glucose abundance (light blue for 2%, dark blue for 8%). We now include a detailed description of the color code when it first appears (Fig 1B) and make sure it is properly reported in all figure legends.

- S3A: the authors should show the statistical difference between WT and *ctf4*-null, which is mentioned as non-existent in p.6

The p value is now represented in Fig EV3A

Text

- RNR1 is not really the gene with the highest score in Figure 5D, not even close: can you give a rationale for pin-pointing it (see also main point 3)?

The reviewer is correct. Perturbations of the mediator complex, which regulate the expression of most of RNA Pol II transcripts, is expected to result in changes in the expression of a large set of genes. However, our focus on dNTPs and *RNR1* was based on the following rationale:

- 1) Gene Ontology Enrichment Analysis: The downregulated genes in our dataset are enriched for the 'nucleotide metabolism' term, which includes pathways critical for dNTP production and directly linked to DNA replication and repair.
- 2) Role of *RNR1*: Among the downregulated genes, *RNR1* stands out as it encodes the major subunit of ribonucleotide reductase, the rate-limiting enzyme in dNTP synthesis. This enzyme is essential for DNA replication, and cells experiencing constitutive DNA replication stress, as in our system, are particularly sensitive to changes in dNTP levels.

However, in response to the reviewers' comments, we now present the decrease in dNTP levels solely as one of three non-mutually exclusive hypotheses, which we reject based on additional experiments:

Line 313: “ The mediator complex was also recently shown to recruit the cohesin loader Scc2 to transcribed genes through an interaction with the Med14 subunit (Mattingly *et al*, 2022). Based on this, we propose three non-mutually exclusive hypotheses to account for the beneficial roles of *med14*-H919P in alleviating DNA replication stress: i) By alleviating transcription genome-wide *med14*-H919P could lower the frequency of collisions of the transcription machinery with replication forks. ii) Alternatively, *med14*-H919P could slow and stabilize replication forks by lowering the dNTP pools through a decreased RNR1 expression. iii) Finally, *med14*-H919P could enhance the recruitment of Scc2, whose excess is beneficial in *ctf4* Δ cells (Fumasoni & Murray, 2020). To distinguish between these hypotheses, we examined the genetic interactions of *med14*-H919P with mutant alleles of *DUN1*, involved in dNTPs regulation (Zhao & Rothstein, 2002), Ribonuclease H1 and H2 (*RNH1* and *RNH201*, respectively), and *SEN1*, implicated in RNA metabolism (Appanah *et al*, 2020; Aguilera & García-Muse, 2012), and *CHL1*, required for cohesion establishment during DNA replication (Skibbens, 2004). Interestingly, *med14*-H919P showed slightly positive genetic interaction with all the alleles (Fig 5D and E). Positive epistasis with both *rnh1* Δ *rnh201* Δ and *sen1*-3, which exacerbate replication-transcription conflicts by failing to process R-loops, supports a role of *med14*-H919P in alleviating these conflicts by lowering transcription. The positive genetic interaction with *dun1* Δ which leads to a decrease in dNTP pools, argues instead against a role for *med14*-H919P in further lowering dNTPs. Finally, the positive interaction manifested with *chl1* Δ is compatible with a role of *med14*-H919P in facilitating cohesion establishment. Overall, these results show how an amino acid substitution in the Med14 subunit of the mediator complex, putatively affecting transcription, is strongly selected, and advantageous, in the presence of constitutive DNA replication stress.”

Line 433: “Based on existing literature, and our analysis of the transcriptome dataset available for the degen-mediated depletion of the Med14 TID we tested three non-mutually exclusive hypotheses for the beneficial effect of *med14*-H919P: (i) reducing replication-transcription conflicts, (ii) lowering dNTP pools, and (iii) facilitating cohesion establishment. Interestingly, *med14*-H919P exhibited a slightly positive genetic interaction with mutant alleles implicated in all three processes (Fig 5D, E). Since hypothesis ii predicts a negative interaction with *dun1* Δ , which also reduces dNTP levels, our findings allow us to reject the possibility that the fitness benefits of *med14*-

H919P arise from altered deoxyribonucleotide metabolism. Instead, our results support hypotheses i and iii, which predict the alleviation of defects caused by defective R-loop metabolism (*rnh1Δ rnh201Δ* and *sen1-3*) and premature sister chromatid separation (*chl1Δ*).

Based on these results, we speculate that the fitness benefits of *med14-H919P* in alleviating DNA replication stress stem from its combined pleiotropic effects, including reduced replication-transcription conflicts and enhanced cohesion establishment. However, future mechanistic studies will be necessary to elucidate the molecular details underlying these effects.”

- The *med14-H919P* mutation is observed in 22/48 wells. I guess the authors checked already: are some of these wells close to each other in the plate?

Correct. We took significant precautions in our experimental design to prevent cross-contamination, as outlined in the Materials and Methods section. Specifically, rows of *ctf4Δ* samples were alternated with rows of WT samples. Daily dilutions were then performed row by row using a 12 channels pipette. This approach ensured that any potential carry-over of cells would result in them being placed in wells containing a different genotype, where they would be eliminated by the consistent use of genotype-specific drugs.

As a result of these measures, we do not observe any distinct pattern of core genetic adaptation corresponding to the plate layout (Figure_rev_only 3). The only exceptions are mutations in *IXR1*, which appear in all *ctf4Δ* strains (albeit with different alleles, see Source Data). Moreover, we reasoned that if a highly fit strain had invaded other wells, all the pre-existing mutations from its lineage would have been detected in those wells. However, apart from the recurrent *ixr1* and *rad9* mutations, which are also strongly adaptive, we find no evidence of shared mutations in wells carrying the *med14-H919P* allele (Figure_rev_only 4).

- Compensatory evolution of *ctf4*-null in 2% glucose is the experiment published by Fumasoni and Murray in eLife. In that paper, there is no trace of mutations in *MED14*. I think the authors should comment on this (different method for detecting putative compensatory mutations?).

We also noticed the absence of *MED14* mutations in the eLife study by Fumasoni and Murray and find this discrepancy intriguing. One possible explanation lies in methodological differences. Our current study employed an improved version of the mutational analysis pipeline. However, we have not yet reanalyzed the original data from the previous study to determine whether *MED14* mutations were present but undetected.

Additionally, two significant experimental differences may also contribute to the observed discrepancy. First, the culture volumes and vessels differed: 10 mL cultures in tubes were used previously, whereas 1.5 mL cultures in 96-well plates were used in the current study.

- I may be mistaken, but Szamecz et al do not actually investigate whether different conditions result in different evolutionary trajectories (i.e., different genetics), and so their results may not be at odds with those presented here.

The reviewer is correct that Szamecz et al. do not explicitly test whether different conditions result in different evolutionary trajectories. However, in the section titled “Compensatory Evolution Generates Diverse Growth Phenotypes across Environments,” they examine how lines evolved in 2% YPD perform across various environments. They report how in roughly 50% of the cases tested, evolved lines showed either no improvement or even some lower fitness than the ancestor (Figure 5A).

While this could be explained by the accumulation of detrimental non-adaptive mutations in specific contexts, it likely implies that the adaptive strategies compensating for the original mutation in one environment do not confer similar benefits in other environments. This observation contrasts with our findings in Figure 6D, where we demonstrate that the main adaptive strategies provide a consistent benefit across diverse environments, including those with glucose, nitrogen, or phosphate abundance or starvation.

We have now modified the introduction, results and discussion to avoid misleading interpretations:

Line 58: “Szamecz and colleagues examined the evolutionary trajectories of 180 haploid yeast gene deletions over 400 generations (Szamecz et al., 2014). They found that, while fitness recovery occurred in the environment where evolution took place, the evolved lines often showed no improvement over their ancestors in other environments. This suggests that compensatory mutations beneficial in one environment often fail to restore fitness in others.”

Line 360: “A previous study in yeast showed how evolved lines that compensate for detrimental defects of gene deletions in standard laboratory conditions often failed to show fitness benefits compared to their ancestor when tested in other environments (Szamecz et al., 2014). We thus investigated the extent to which the core genetic adaptation to DNA replication stress was beneficial under alternative nutrient conditions.”

Line 455: “What could explain the discrepancies between our results, and previous studies on evolutionary repair highlighting the role of the environment in shaping evolutionary trajectories (Filteau et al., 2015), and the heterogeneous behavior of evolved lines in various environments (Szamecz et al., 2014)?”

typos

p.18, line 564 preformed -> performed

p. 6 line 189 with a strongly skew -> with a strong skew ?

Typos are now corrected in the main text

Reviewer #2 (Significance (Required)):

This is a well-done paper that could be of interest for the community of evolutionary biologists, scientists working on metabolism and cell division. It addresses an interesting problem, how metabolism affects compensatory evolution. Among the strengths: experiments are well done, the results are novel, the cross-talk between metabolism and evolutionary repair is intriguing. Among the weaknesses, the fact that the molecular explanations for the observations are only hypothesized and not tested experimentally. This is where the authors could improve the manuscript.

Reviewer #3

This paper combines phenotypic and genomic data from an experimental evolution study in yeast to assess how repeatable evolution is in response to DNA replication stress. Importantly, the authors ask whether genotype by environment interactions influence repeatability of their evolved lines. To this end, the authors have constructed an elegant highly-replicated experiment in which two yeast genotypes (WT and CTF4 KO) were evolved under a variety of glucose levels for 1,000 generations. Recurrent mutations are found across many replicates, suggesting that repeatability is robust to GxE interactions. Of course, the authors correctly identify that these results are dependent on many particulars, as is always the case in biology, but provide a comprehensive discussion to accompany their results. I do not have any major comments to give, but simply some suggestions and points of clarification.

Major comments: N/A

Minor comments:

L19: I found the definition for compensatory evolution/mutations to be somewhat vague in the introduction (and subsequently throughout the text). It's clear that this was written for a more medical/physiological audience, but without a more explicit explanation of compensatory evolution/mutations, it became difficult to properly weigh some claims/discussions made by the authors later on. Do you define compensatory mutations as those which completely recover WT function/fitness, or are simply of opposite effect to the altered genotype? Others define "compensatory evolution" as simply any epistatically interacting amino acid substitutions (Ivankov et al, 2014). It would be nice to see more explicitly defined.

We thank the reviewer for highlighting the need for a precise definition of compensatory evolution and compensatory mutations. We recognize that the literature encompasses multiple definitions, including the one cited by the reviewer, which emphasizes compensatory mutations within the context of structural biology. This particular definition, prevalent in molecular evolution, was introduced by Kimura (Kimura, 1985) and is frequently used to explain the co-occurrence of amino acid mutations within a protein. These mutations offset each other's defects, restoring or maintaining protein function. Here, however, we are using an older and broader definition of compensatory mutation, first introduced by Wright (Wright, 1964, 1977, 1982) and frequently used in evolutionary genomics (e.g., Moore et al., 2000; Szamecz et al., 2014; Rajon and Mazel, 2013; Eckart et al., 2024). This definition includes any mutation in the rest of the genome that compensates (fully or partially) for another mutation's detrimental effects on fitness.

We have now included this definition in the introduction:

Line 35: "Compensatory evolution is a process by which cells mitigate the negative fitness effects of persistent perturbations in cellular processes across generations. This adaptation occurs through spontaneously arising compensatory mutations anywhere in the genome (Wright, 1964, 1977, 1982) that partially or fully alleviate the negative fitness effects of perturbations (Moore et al., 2000). The successive accumulation of compensatory mutations over evolutionary timescales progressively repair the cellular defects, ultimately restoring fitness."

Line 497: "Our findings demonstrate that while glucose availability significantly affects the physiology and adaptation speed of cells under replication stress, it does not alter the fundamental genome-wide compensatory mutations that drive fitness recovery and evolutionary repair."

Along these lines, I would have liked to see a more direct comparison/discussion of the degree to which deletion lines recovered. I can see from Fig 2E and Fig EV2B that fitness increased quite a bit; would it not be possible to include a figure on the degree of compensation (basically relative fitness of evolved deletion lines - relative fitness of ancestral deletion lines)?

If the reviewer is suggesting calculating the difference between the evolved and ancestor fitness, the data is already in Figure EV2B and EV2D, defined as 'Absolute fitness gains Δ ' and calculated as $\Delta = \text{evo\%} - \text{anc\%}$.

If instead is suggesting to plot the fitness of evolved deletion lines (Y axis) against the relative fitness of ancestral deletion lines (X axis), we have now produced the plot is Figure EV2F.

To better understand the extent of the fitness recovery in Ctf4 strains, we have also calculated and plotted the 'relative fitness gain' calculated as $|\text{evo\%}| / |\text{anc\%}| * 100$ (Figure EVC)

We are now commenting on these comparisons in the following paragraph:

Line 185: “By generation 1000, both WT and *ctf4Δ* evolved lines achieved, on average, slightly higher fitness in low glucose compared to high glucose conditions (Fig EV2B). However, due to the varying initial fitness of *ctf4Δ* cells across different glucose environments, they recovered the same extent of the original defect (Fig EV2C), displaying an opposite trend to WT, with increasing absolute fitness throughout the experiment as glucose concentration rose (Fig EV2B vs S2D). The different absolute fitness gains over the same number of generations highlight distinct mean adaptation rates (Fig 2B). These differences are evident when examining the evolutionary dynamics of the evolved lines over time (Fig 2C). Additionally, we approximated the fitness trajectories using the power law function (Fig 2C, dashed purple lines), previously proposed to describe long-term evolutionary dynamics in constant environments (Wiser et al., 2013). The parameter *b* in this formula determines the curve's steepness, and can be used to quantify the global fitness change over generations (Fig EV2E). Collectively, these analyses demonstrate that, unlike WT cells, *ctf4Δ* lines adapt faster in the presence of high glucose. This evidence aligns with the declining adaptability observed in other studies (Moore et al., 2000; Kryazhimskiy et al., 2014; Couce & Tenaillon, 2015), where low-fitness strains consistently adapt faster than their more fit counterparts (Fig EV2F).”

L57: Another minor nitpick that just comes down to semantics. When discussing "96 parallel populations", it invokes a higher sense of replication than is actually present in the study. I would rephrase this to something along the lines of "12 replicate populations across 8 treatments under conditions of [...]".

We changed the sentence as follows:

Line 81: “We evolved 96 parallel populations of budding yeast, organized into 12 replicate lines, across four conditions of glucose availability (from starvation to abundance) with or without replication stress.”

L185-187: The wording here needs to be clarified. Be explicit in that are examine the ratio (or count) of synonymous to non-synonymous mutations here, otherwise the interpretations appears to be direct contradiction to the (as written) results. Only after viewing the supplemental figure was I able to figure out what exactly was meant here.

We changed the sentence as follows:

Line 226: “We found no significant differences in the numbers of synonymous mutations detected in evolved populations in WT and *ctf4Δ* populations (Fig EV3A). These results support the hypothesis that replication stress in *ctf4Δ* lines favors the retention of beneficial mutations, rather than simply increasing the overall mutation rate.”

L349-350: The authors observe higher rates of adaptation in deletion lines than WT lines, and discuss this in adequate detail. Although not explicitly mentioned, this is consistent with a diminishing returns epistasis model (that could be beneficial to discuss, but is not necessary), which has been implicated in modulating the degree of repeatability observed along evolutionary trajectories (Wünsche et al. 2017). Although definitely not required for this already very nice manuscript, I think it would be very rewarding if the authors were to eventually analyze fine-scale dynamics of phenotypic and genomic adaptation to mine for these putative interactions and their influence on repeatability.

We agree with the reviewer on how our results align with a model of diminishing returns epistasis. This pattern is apparent not only between *ctf4Δ* and WT lines but also among *ctf4Δ* lines evolved in different glucose conditions. This phenomenon likely arises from the interaction of various adaptive mutations, which we aim to explore further in a dedicated manuscript. However, until we do so, we prefer to refer generally to a pattern of declining adaptability. To explicit this trend we have now included Fig EV2F and commented on it in the manuscript:

Line 195: “This evidence aligns with the declining adaptability observed in other studies (Moore et al., 2000; Kryazhimskiy et al., 2014; Couce & Tenaillon, 2015), where low-fitness strains consistently adapt faster than their more fit counterparts (Fig EV2F).”

Line 420: "Our results are consistent with declining adaptability, as evidenced by the reduced rates of adaptation observed both between *ctf4Δ* and WT lines and among *ctf4Δ* lines evolved in different glucose conditions (Fig EV2F)"

Reviewer #3 (Significance (Required)):

It is clear to me that a great deal of time and care has been put into this study and the preparation of this manuscript. The science and analyses are appropriate to answer the questions at hand, and it bodes well that whenever I had a question pop up while reading, they were typically answered immediately after. I think that this manuscript will be broadly relevant to both biologists both evolutionary and clinical, and was written in a way to be accessible to both.

As someone with an expertise in repeatable evolution, I felt most excited by the observation of so many parallel substitutions at a single amino acid across deletion lines. As the authors rightfully point out in the results and discussion, it's likely that this degree of robustness is highly dependent on the particular mechanism of disruption that cells experience. The authors then go above and beyond to functionally validate the putative molecular mechanisms of (repeatable) adaptation in this system. While it may not always be possible to accomplish in non-model organisms, such multi-modal approaches will be crucial to advance the field of repeatable evolution.

Reviewer #4

The authors investigated the effects of DNA replication stress on adaptation in different nutrient availabilities by passaging wild-type and *ctf4Δ* *Saccharomyces cerevisiae* in media with varying levels of glucose over ~1000 generations. The *ctf4Δ* strain experiences increased DNA replication stress due to the deletion of a non-essential replication fork protein. The authors found differences in evolution between wild-type and *ctf4Δ* yeast, which held across different growth media. This study identified a compensatory single amino acid variant in Med14, a protein in the mediator complex of RNA polymerase II, that was specifically selected in *ctf4Δ* strains. The authors conclude that while environmental nutrient availability has implications for cell fitness and physiology, adaptation is largely independent and instead dependent on genetic background. The data provide excellent support for the key aspects of the models, although some details are (to me) overstated.

Major comments:

- A *ctf4Δ* mutant strain was used to investigate the effects of replication stress. Why was this mutant chosen instead of other deletions that cause different types of replication stress?

We appreciate the opportunity to clarify our rationale for choosing the *ctf4Δ* mutant. The following are the main reasons why we believe *ctf4Δ* strains represent an ideal tool to study a global perturbation of the DNA replication program over evolutionary timescales:

- 1) General replication stress: The absence of Ctf4 perturbs replication fork progression, leading to a spectrum of replication stress-related phenotypes, including DNA damage sensitivity, single-stranded DNA gaps, reversed forks (Abe et al., 2018; Fumasoni et al., 2015), checkpoint activation (Poli et al., 2012), cell cycle delays (Miles and Formosa, 1992), increased recombination (Alvaro et al., 2007), and chromosome instability (Kouprina et al., 1992). This broad disruption makes it an excellent model for observing global perturbations in replication processes. In contrast, other mutants typically affect specific enzymatic (e.g.,

POL32 and *RRM3*) or signaling (e.g., *MRC1*) functions, making them better suited to address specific questions.

- 2) Constitutive stress: Unlike drug-induced stress (e.g., Hydroxyurea; Krakoff et al., 1968) or conditional depletion systems (e.g., *GALI-POLε*; Zhang et al., 2022), which cells can easily circumvent through single mutations, *ctf4Δ* enforces persistent replication stress. Its deletion cannot be complemented by a single mutation, ensuring a robust and consistent stress environment for evolutionary studies.

We have now modified the main text to convey these advantages in a concise form:

Line 106: “In the absence of Ctf4, cells exhibit multiple defects commonly associated with DNA replication stress, such as single-stranded DNA gaps and altered replication forks (Fumasoni et al., 2015), leading to basal cell cycle checkpoint activation (Poli et al., 2012). These defects result in severe and persistent growth impairments, cell cycle delays, elevated nucleotide pools and chromosome instability (Miles and Formosa, 1992; Kouprina et al., 1992; Poli et al., 2012), making *ctf4Δ* mutants an ideal model for studying the cellular consequences of general and constitutive replication stress over evolutionary time.”

It's not clear from the study that the effects are generalizable to other forms of replication stress.

As with any method to induce DNA replication stress (including commonly used drugs like HU) each approach inevitably affects replication in a specific manner. Testing the broader applicability of our conclusions would require evolving additional strains with different replisome perturbations. For instance, mutations in *ELG1* and *CTF18* (affecting the alternative Replication Factor C), *POL30* (affecting the sliding clamp PCNA), *POL32* (affecting Polε), *RRM3* (protective helicase) and (*MRC1* (coordinating leading strand activities and signaling to the checkpoint) would have to be considered. Furthermore, specific mutant alleles of Ctf4 that disrupt interactions with particular binding partners (Such as *ctf4-4E* and *ctf4-3E*, perturbing the interaction with the CMG helicase and accessory factors respectively) will be highly informative on which specific aspects of the replication stress generated by the lack of Ctf4 each adaptive mutation alleviate.

However, accommodating such extensive variability would inflate the sample size to an extent that will become unfeasible within the experimental design focused on capturing parallel evolution over a nutrient gradient (the primary focus of this study). We agree that this is an important question and intend to address it comprehensively in a dedicated future study.

- The authors could be clearer that a (the?) cause of the *ctf4Δ* fitness defect is spurious upregulation of *RNR1*. I don't think it is mentioned until the Discussion, but it is highly relevant to Fig 4, and to the adaptations one would expect from *ctf4Δ*.

We thank the reviewer for the opportunity to clarify this aspect. We do not think that the fitness defects of *ctf4Δ* cells stem solely from the spurious upregulation of *RNR1*. However, we believe that a major aspect of the evolutionary adaptation is aimed at decreasing dNTP levels. We are now mentioning increased dNTPs as a major phenotype of *ctf4Δ* and commenting on the hypothesis more clearly in the discussion.

Line 108: “These defects result in severe and persistent growth impairments, cell cycle delays, elevated nucleotide pools and chromosome instability (Miles and Formosa, 1992; Kouprina et al., 1992; Poli et al., 2012)”

- In Figure 1E, there is a very large spread in the relative fitness at 2% and 8% glucose, but this was not commented on. Is this heteroscedasticity expected?

The observed heteroscedasticity can be explained as follows: Our competition assays tend to exhibit increased variability when a strain approaches very low fitness levels. Specifically, as one strain nears extinction by the third day of competition, its abundance is estimated based on a much smaller number of events in the flow cytometer. Furthermore, we noticed a small number of reference cells carrying *pACT1-yCerulean* not showing strong fluorescence in 8% glucose. The nature of this effect is uncertain, and possibly linked to metabolism-linked changes in the cytoplasm. The combination of these two phenomena amplifies the impact of noise inherent to the methodology, leading to increased variability across replicates.

Nonetheless, the overall decreasing fitness trend across glucose conditions, combined with the statistical significance observed between high and low glucose levels, collectively convey a robust phenotype

- The *med14-H919P* mutant was highly selected in *ctf4Δ* strains, independent of glucose availability. Is this variant found in any natural yeast strains (i.e., are there environments that select for this variant)? Also, if this variant is found in natural strains, does it co-occur with other mutations that could affect DNA replication?

To address this point, we analyzed a recently published dataset reporting genetic variation among 3,034 natural and domesticated *S. cerevisiae* isolates collected in multiple studies (Loegler et al., bioRxiv, 2024). While we observed substantial variability across several regions of *MED14*, including much of the Tail Interaction Domain (TID), we did not identify any variants affecting the specific residue we found mutated (site 2756, Figure_rev_only 5).

At this stage, we are uncertain about including this analysis in the manuscript. We appreciate the reviewer's suggestion, as this is indeed an interesting point. However, addressing it thoroughly would require predicting the functional impact of variants within the TID domain and assessing their co-occurrence with putative loss-of-function alleles in DNA replication factors. We believe that such an analysis, along with broader investigations into the co-occurrence of genetic variants affecting DNA replication, would be best suited for a dedicated computational study.

- The statement on lines 271-273 is not particularly well-supported. The analysis of the Warfield data suggest that reduced expression of *RNR1* could be causal, but the data don't go as far as showing how the *med14* mutation is advantageous in *ctf4Δ*. Further experimentation would be necessary to support the possibilities that the authors discuss.

Based on the lines, we believe the sentence the reviewer refers to is: “Overall, these results show how an amino acid substitution in the *Med14* subunit of the mediator complex, putatively affecting transcription, is strongly selected, and advantageous, in the presence of constitutive DNA replication stress.” We are unsure which aspect of the statement is seen as unsupported. The mutation's strong selection in *ctf4Δ* is demonstrated in Figures 5A, 6A, and S4C, its effect on transcription is cautiously defined as ‘putative’, while its advantageous nature is supported by Figures 5B and S4B.

Nonetheless, we were encouraged by reviewers' #2 and #4 interest in this newly identified compensatory mutant. We thus decided to investigate alternative hypotheses and performed further experiments to better understand the biological processes affected by *med14-H919P*.

Line 313: “ The mediator complex was also recently shown to recruit the cohesin loader *Scs2* to transcribed genes through an interaction with the *Med14* subunit (Mattingly *et al*, 2022). Based on this, we propose three non-mutually exclusive hypotheses to account for the beneficial roles of *med14-H919P* in alleviating DNA replication stress: i) By alleviating transcription genome-wide *med14-H919P* could lower the frequency of collisions of the transcription machinery with replication forks. ii) Alternatively, *med14-H919P* could slow and stabilize replication

forks by lowering the dNTP pools through a decreased RNR1 expression. iii) Finally, *med14*-H919P could enhance the recruitment of Scc2, whose excess is beneficial in *ctf4* Δ cells (Fumasoni & Murray, 2020). To distinguish between these hypotheses, we examined the genetic interactions of *med14*-H919P with mutant alleles of *DUN1*, involved in dNTPs regulation (Zhao & Rothstein, 2002), Ribonuclease H1 and H2 (*RNH1* and *RNH201*, respectively), and *SEN1*, implicated in RNA metabolism (Appanah *et al*, 2020; Aguilera & García-Muse, 2012), and *CHL1*, required for cohesion establishment during DNA replication (Skibbens, 2004). Interestingly, *med14*-H919P showed slightly positive genetic interaction with all the alleles (Fig 5D and E). Positive epistasis with both *rnh1* Δ *rnh201* Δ and *sen1-3*, which exacerbate replication-transcription conflicts by failing to process R-loops, supports a role of *med14*-H919P in alleviating these conflicts by lowering transcription. The positive genetic interaction with *dun1* Δ which leads to a decrease in dNTP pools, argues instead against a role for *med14*-H919P in further lowering dNTPs. Finally, the positive interaction manifested with *chl1* Δ is compatible with a role of *med14*-H919P in facilitating cohesion establishment. Overall, these results show how an amino acid substitution in the Med14 subunit of the mediator complex, putatively affecting transcription, is strongly selected, and advantageous, in the presence of constitutive DNA replication stress.”

Line 433: “Based on existing literature, and our analysis of the transcriptome dataset for the degron-mediated depletion of the Med14 TID (Fig EV4A) we tested three non-mutually exclusive hypotheses for the beneficial effect of *med14*-H919P: (i) reducing replication-transcription conflicts, (ii) lowering dNTP pools, and (iii) facilitating cohesion establishment. Interestingly, *med14*-H919P exhibited a slightly positive genetic interaction with mutant alleles implicated in all three processes (Fig 5D, E). Since hypothesis II predicts a negative interaction with *dun1* Δ , which also reduces dNTP levels, our findings allow us to reject the possibility that the fitness benefits of *med14*-H919P arise from altered deoxyribonucleotide metabolism. Instead, our results support hypotheses I and III, which predict the alleviation of defects caused by defective R-loop metabolism (*rnh01* Δ *rnh201* Δ and *sen1-3*) and premature sister chromatid separation (*chl1* Δ).

Based on these results, we speculate that the fitness benefits of *med14*-H919P in alleviating DNA replication stress stem from its combined pleiotropic effects, including reduced replication-transcription conflicts and enhanced cohesion establishment. However, future mechanistic studies will be necessary to elucidate the molecular details underlying these effects.”

We are grateful to the reviewer for prompting us to test our previous hypothesis, which we found to be incorrect. We believe that the new data, along with our insights into the processes affected by *med14*-H919P, provide a valuable complement to the identification of this novel selected allele (Fig 5A) and the confirmation of its fitness benefits across glucose conditions (Fig 5B). We hope the reviewer appreciates that further molecular characterization of *med14*-H919P falls beyond the scope of this study, which focuses on how nutrient availability influences evolutionary dynamics and compensatory adaptation in *ctf4* Δ cells.

- The authors comment that the *med14*-H919P mutant could have implications for the stability of Med14, based on computational modelling. Verifying the stability of the *med14*-H919P in vivo would strengthen this discussion.

We believe that in vivo and in vitro structural studies investigating the effect of this mutation on the stability and function of the Mediator complex are beyond the scope of this manuscript. These investigations would be more appropriately addressed in future, dedicated studies focused on these specific aspects.

- In the discussion, the authors propose that the context of the perturbation may influence the robustness of adaptation. A more detailed explanation of this point (including a discussion of the findings of other similar studies investigating different conditions) would be helpful to further bolster this section.

We are now supporting this concept more explicitly by commenting on other studies as follows:

Line 462: “Third, the environment’s influence on compensatory evolution may depend on the specific cellular module perturbed and its genetic interactions with other modules that are significantly influenced by environmental conditions. For example, the actin cytoskeleton, which must rapidly respond to extracellular stimuli, is likely to be more directly influenced by environmental factors (Filteau et al., 2015) compared to the DNA replication machinery, which operates within the nucleus and is relatively insulated from such changes. Supporting this idea, a study examining mutants’ fitness across diverse environments found that conditions such as TOR inhibition, or level of carbon sources similar to those used in this study, primarily affected genes involved in vesicle trafficking, transcription, protein metabolism, and cell polarity. In contrast, genes associated with genome maintenance, as well as their epistatic interactions, were largely unaffected (Costanzo et al., 2021)”.

Minor comments:

- Competitions were performed between *ctf4Δ* strains and a constructed strain with yCerulean integrated at ACT1. Is the fitness of the fluorescent strain comparable to the ancestral wild-type strain (i.e., in a competition between the ancestral WT and the fluorescent strain, does either have an advantage)?

We noticed a slight disadvantage of the reference strain compared to WT, likely due to the costs of the extra fluorescence reporter. However, the disadvantage is minimal, ranging from -0.5 to -2.5 depending on the glucose environment (raw measurements are reported in Source Data (Fig1), sheet 5. To take this into account, all fitness reported in figures are normalized for the WT value measured in the same environment, line 596: “Relative fitness of the ancestral WT strain was used to normalize fitness across conditions.”

- In Figure 3, the legends for panels B and C appear to be swapped. Discussion of Figure 3 on pages 6 and 7 appear to reference the wrong panels.

We are unsure about this typo. Main text and figure legend seem to refer to the appropriate panels, 3B for mutation fractions and 3C for mutation counts.

- In Figure 4A and B, having the same colour scale between both heatmaps is misleading, as the scales are different. Consider having the same scale across both heatmaps so that enrichments are visually comparable.

Following the reviewer’s suggestion we have chosen a uniform heatmap to visually represent GO terms enrichment in WT and *ctf4Δ* genetic backgrounds.

- In Figure 4C, having a legend in the figure for node size would be helpful to understand the actual number of populations with mutations in each gene.

A legend for node size has now been added next to Figure 4C.

Reviewer #4 (Significance (Required)):

In this study, a high-throughput evolution experiment uncovered the effects of genetic background on the development of adaptive mutations. The authors were able to identify a single amino acid variant of Med14 (med14-H919P) that was positively selected in *ctf4Δ*. Furthermore, they demonstrated the causality of med14-H919P in conferring a fitness advantage in *ctf4Δ*. The novelty of this mechanistic finding opens future avenues of investigation regarding the interaction network of the mediator complex in conditions of DNA replication stress. A

limitation of the study is that only one mechanism of replication stress was assessed (*ctf4Δ*). Other gene mutations that cause replication stress would be interesting to assess and would provide a more thorough investigation of the effects of DNA replication factors on evolvability.

This work will be of interest to researchers in the population genetics and genotype-by-environment fields, as it suggests the robustness of evolvability to environmental factors in the specific condition of DNA replication stress. As discussed by the authors, this finding differs from other works that have linked environmental conditions to adaptive evolution to different conditions, and is concordant with work that indicates the robustness of genetic interactions to environmental stresses. Furthermore, the identification of the highly-selected *med14-H919P* variant will be of interest to the DNA replication field. There is the potential for future work investigating the role of Med14 in mediating the response to DNA replication stress in both yeast and mammalian cell contexts, since the authors note that there are links between altered mediator complex regulation and cancers. Although I suspect that the very different regulation of RNR in mammalian cells makes it unlikely that the kind of upregulation of dNTP pools seen in *ctf4Δ* would be induced by replication stress in mammalian cells.

9th May 2025

Manuscript Number: MSB-2025-12869R

Title: Compensatory Evolution to DNA Replication Stress is Robust to Nutrient Availability

Author: Mariana Natalino

Marco Fumasoni

Dear Dr. Fumasoni,

Thank you for the submission of your revised manuscript to Molecular Systems Biology. We have now received the enclosed reports from the referees that were asked to re-assess it. As you will see the reviewers are now globally supportive and I am pleased to inform you that we will be able to accept your manuscript pending the following final amendments:

- 1) Please clearly indicate the corresponding author on the main manuscript title page along with a contact email address.
- 2) Please format the Data availability section according to the example below (i.e. please provide a specific URL for PRJEB87420 datasets and please provide a more specific link to the scripts for analysis available on your Github page, as currently the link leads to the general homepage):
"The datasets and computer code produced in this study are available in the following databases:
- Chip-Seq data: Gene Expression Omnibus GSE46748 (<https://www.ncbi.nlm.nih.gov/geo/query/acc.cgi?acc=GSE46748>)
- Modeling computer scripts: GitHub (<https://github.com/SysBioChalmers/GECKO/releases/tag/v1.0>)
- [data type]: [full name of the resource] [accession number/identifier] ([doi or URL or identifiers.org/DATABASE:ACCESSION])"
- 3) Please include a "Disclosure and competing interests statement". We updated our journal's competing interests policy in January 2022 and request authors to consider both actual and perceived competing interests. Please review the policy <https://www.embopress.org/competing-interests> and update your competing interests if necessary.
- 4) Our journal encourages inclusion of *data citations in the reference list* to directly cite datasets that were re-used and obtained from public databases. Data citations in the article text are distinct from normal bibliographical citations and should directly link to the database records from which the data can be accessed. In the main text, data citations are formatted as follows: "Data ref: Smith et al, 2001" or "Data ref: NCBI Sequence Read Archive PRJNA342805, 2017". In the Reference list, data citations must be labeled with "[DATASET]". A data reference must provide the database name, accession number/identifiers and a resolvable link to the landing page from which the data can be accessed at the end of the reference. Further instructions are available at .
- 5) In the Methods, please take care of the following:
- The Materials and Methods section should be renamed to "Methods".
- 6) Please place individual sections of the manuscript in the following order: Title page - Abstract & Keywords - Introduction - Results - Discussion - Methods - Data Availability - Acknowledgements - Disclosure and Competing Interests Statement - References - Figure Legends - Expanded View Figure Legends.
- 7) For the figures and figure legends, please take care of the following:
- Please remove all figures from main manuscript file and leave only main figure legends followed by the expanded view figure legends, both placed after the references.
- Please note that the exact p values are not provided in the legends of figures 1A, 2B, 3A, EV1 A, C; EV2 E.
- Please note that the box plots need to be defined in terms of minima, maxima, centre, bounds of box and whiskers, and percentile in the legends of figures 1C, D; 3C, EV1 C, EV2 C-E; EV3 C
- Please note that information related to n is missing in the legends of figures 3A, C; EV3 B; EV5 A, B.
- Although 'n' is provided, please describe the nature of entity for 'n' in the legends of figures 1A, C, D, E; 2B, C; 5B, D; 6C, D; EV1 A, C; EV2 B-E.
- 8) Expanded view dataset: There is one dataset uploaded as part of the submission as Dataset EV4 (source file and callout actually read Dataset EV5). Since there does not appear to be Datasets EV1-EV3, this should be updated to Dataset EV1. Please be sure to update the callout for this dataset in the main manuscript and the name of the file in our system.
- 9) Appendix file: As you have included only the expanded view figures in the Appendix, the Appendix file is actually not needed (since expanded view figures are linked as individual high quality files). The only remaining Appendix file is Appendix Table S1, but this was uploaded separately as Excel file. In this case, since an Appendix file is not really necessary, we would suggest changing this to an Expanded View table as Table EV1.
- 10) Source Data: Please ensure that a completed Source Data checklist is uploaded as a Related Manuscript File (this file should have been sent to you previously in a separate email). Source Data should be organized as a single source data file (zipped) per figure for main figures (all EV and/or Appendix figure Source Data can be included in a single folder), with the panels clearly visible in the folder structure instead of a single excel file for all Source Data. e.g. all the Source data files for figure 1 need to be saved in a single folder and this needs to be zipped and then uploaded as "SD figure 1.zip" file. Currently the Source Data files are not clearly labeled and e.g. Figure 1 and EV1 should not be in the same folder.
- 11) As part of the EMBO Publications transparent editorial process initiative (see our policy here: https://www.embopress.org/transparent-process#Review_Process), Molecular Systems Biology will publish online a Peer Review File (PRF) to accompany accepted manuscripts. This file will be published in conjunction with your paper and will include

the anonymous referee reports, your point-by-point response and all pertinent correspondence relating to the manuscript. Let us know whether you agree with the publication of the PRF and as here, if you want to remove or not any figures from it prior to publication. Please note that the Authors checklist will be published at the end of the PRF.

12) After your paper is published, we may promote it on social media. If you have any handles or hashtags for Bluesky you would like included, please let us know.

13) Please provide a point-by-point letter INCLUDING my comments as well as the reviewer's reports and your detailed responses (as Word file).

I look forward to reading a new revised version of your manuscript as soon as possible.

Yours sincerely,

Poonam Bheda, PhD
Scientific Editor
Molecular Systems Biology

Reviewer #1:

All my previous concerns have been addressed. The manuscript looks great.

Reviewer #2:

The authors have addressed carefully most of my concerns. In particular, they produced new experimental evidence for addressing the role of the compensatory mutation in MED14 (Figure 5); they have tested a mutant that express all the 'core genetic adaptation genes' (Figure 6); and have tested the interplay between Rnr1 and ixr1 compensatory mutation (Figure EV4). The manuscript is much improved. I have only some minor points:

- Figure EV4 shows that the compensatory effects of the inactivation of IXR1 is lost when a form of RNR1 refractory to feedback inhibition (rnr1-D57N) is overexpressed. The data also show that the simple overexpression of RNR1 does not affect the rescue (why is that?). These other data are shown but not discussed. Moreover, this Figure is only cited in the Discussion (i.e., does not appear in between Figure EV4A and EV5A). I think it could be discussed a bit more thoroughly in the right spot.

- line 70, not clear to me what 'For example' refers to (as an example of what?).

- lines 190-193 have become quite difficult to follow, I would suggest to explain better observation and conclusion (why the proportion of fitness effect recovered should remain constant in consequence of the varying initial fitness?).

- line 350, 'with the exception of IXR1 deletion', but then in the figure we do not see the statistical significance.

- Figure EV5A-B: maybe use the same scale on the two plots, to better appreciate the difference between WT and ctf4-null?

- line 372, 'a frequent combination' somehow sounds strange to me (a combination of frequent adaptive mutations?).

I also had a look at the comments of referee 4, and here are my thoughts:

I think the authors have convincingly addressed the concerns.

One major concern dealt with the lack of generality of the effect (i.e., whether other forms of replication stress may obtain the same effect). Here the authors argue,

convincingly, that a thorough analysis of other mutants that cause replication stress would be material for several papers.

A second major point was the mechanism by which the med14 mutations is advantageous. This was addressed with new experiments shown in Figure 5. The results do not explain the full molecular mechanism, but they suffice to exclude the hypothesis presented originally and to focus on the pleiotropic effects of the mutation.

The other major points raised by the reviewer lead to a study on the presence in wild strain of the med14 mutation (specific figure 5 for the reviewer not included in the paper); to specify some concepts in the text; and the heteroscedasticity in the data of Figure 1E. They have all been changed/addressed/commented.

To summarize, I think the only point that could still be debatable is the generality of the conclusion in terms of replication stress. The authors find that the genetic changes are largely independent on nutrient availability. Would that be true in the presence of other sources of replication stress? Reading the paper, one could predict that this would depend on the strength of the stress. However, a thorough investigation of other sources of stress would require to produce an equal amount of work for every source of stress, which I believe would not be suitable.

Reviewer #3:

I was happy to see the improvements the authors made in an already high-quality manuscript. I have no further revisions.

Rev_Com_number: RC-2024-02762

New_manu_number: MSB-2025-12869R

Corr_author: Fumasoni

Title: Compensatory Evolution to DNA Replication Stress is Robust to Nutrient Availability

Dear Dr. Bheda,

We are glad to hear the reviewers are supportive. Below you find our point-by-point response to your comments and to the remaining suggestions by Reviewer #2.

- 1) Please clearly indicate the corresponding author on the main manuscript title page along with a contact email address.

The corresponding author information is now indicated on the manuscript.

- 2) Please format the Data availability section according to the example below (i.e. please provide a specific URL for PRJEB87420 datasets and please provide a more specific link to the scripts for analysis available on your Github page, as currently the link leads to the general homepage):

"The datasets and computer code produced in this study are available in the following databases:

- Chip-Seq data: Gene Expression Omnibus GSE46748

(<https://www.ncbi.nlm.nih.gov/geo/query/acc.cgi?acc=GSE46748>)

- Modeling computer scripts: GitHub

(<https://github.com/SysBioChalmers/GECKO/releases/tag/v1.0>)

- [data type]: [full name of the resource] [accession number/identifier] ([doi or URL or identifiers.org/DATABASE:ACCESSION])"

The data availability section has now been modify to include specific links:

"The datasets and computer code used in this study are now available in the following databases:

- FASTAQ files: European Nucleotide Archive PRJEB87420

(<https://www.ebi.ac.uk/ena/browser/view/PRJEB87420>)

- Scripts used for data analysis are available at the GitHub repository

(https://github.com/FumaLab/NatalinoFumasoni_2025) "

- 3) Please include a "Disclosure and competing interests statement". We updated our journal's competing interests policy in January 2022 and request authors to consider both actual and perceived competing interests. Please review the policy <https://www.embopress.org/competing-interests> and update your competing interests if necessary.

The section of competing interests was now added to the manuscript.

The authors state they have no competing interests or disclosures

- 4) Our journal encourages inclusion of *data citations in the reference list* to directly cite datasets that were re-used and obtained from public databases. Data citations in the article text are distinct from normal bibliographical citations and should directly link to the database records from which the data can be accessed. In the main text, data citations are formatted as follows: "Data ref: Smith et al, 2001" or "Data ref: NCBI Sequence Read Archive PRJNA342805, 2017". In the Reference list, data citations must be labeled with "[DATASET]". A data reference must provide the database name, accession number/identifiers and a resolvable link to the landing page from which the data can be

accessed at the end of the reference. Further instructions are available at <https://www.embopress.org/page/journal/17574684/authorguide#referencesformat>.

We have now incorporated data citations in reference list and in text

- 5) In the Methods, please take care of the following:
 - The Materials and Methods section should be renamed to "Methods".

The section was renamed in the manuscript

- 6) Please place individual sections of the manuscript in the following order: Title page - Abstract & Keywords - Introduction - Results - Discussion - Methods - Data Availability - Acknowledgements - Disclosure and Competing Interests Statement - References - Figure Legends - Expanded View Figure Legends.

The order and title of the sections has been corrected

- 7) For the figures and figure legends, please take care of the following:
 - Please remove all figures from main manuscript file and leave only main figure legends followed by the expanded view figure legends, both placed after the references.

The figures were removed, and figure legends were placed in their respective section after the references

- Please note that the exact p values are not provided in the legends of figures 1A, 2B, 3A, EV1 A, C; EV2 E.

Exact p-values were added either to the figure or to the figure legend.

- Please note that the box plots need to be defined in terms of minima, maxima, center, bounds of box and whiskers, and percentile in the legends of figures 1C, D; 3C, EV1 C, EV2 C-E; EV3 C

Box plots are now defined as following: "Box plots represent the median (center line), 25th and 75th percentiles (lower and upper bounds of the box), and whiskers extending to the smallest and largest values within 1.5× the interquartile range (IQR) from the lower and upper quartiles, respectively. Data points beyond whiskers are shown as outliers."

Please note that information related to n is missing in the legends of figures 3A, C; EV3 B; EV5 A, B.

The information on *n* was included in the remaining figure legends

Although 'n' is provided, please describe the nature of entity for 'n' in the legends of figures 1A, C, D, E; 2B, C; 5B, D; 6C, D; EV1 A, C; EV2 B-E.

The nature of entity for *n* was included in the figure legends

- 8) Expanded view dataset: There is one dataset uploaded as part of the submission as Dataset EV4 (source file and callout actually read Dataset EV5). Since there does not appear to be Datasets EV1-EV3, this should be updated to Dataset EV1. Please be sure to update the callout for this dataset in the main manuscript and the name of the file in our system.

The Dataset EV5 was now changed to EV1 both in the manuscript and in the submission portal.

- 9) Appendix file: As you have included only the expanded view figures in the Appendix, the Appendix file is actually not needed (since expanded view figures are linked as individual high quality files). The only remaining Appendix file is Appendix Table S1, but this was uploaded separately as Excel file. In this case, since an Appendix file is not really necessary, we would suggest changing this to an Expanded View table as Table EV1.

The Appendix Table S1 was now changed to Table EV1 both in the manuscript and in the submission portal.

- 10) Source Data: Please ensure that a completed Source Data checklist is uploaded as a Related Manuscript File (this file should have been sent to you previously in a separate email). **Source Data should be organized as a single source data file (zipped) per figure for main figures (all EV and/or Appendix figure Source Data can be included in a single folder), with the panels clearly visible in the folder structure instead of a single excel file for all Source Data.** e.g. all the Source data files for figure 1 need to be saved in a single folder and this needs to be zipped and then uploaded as "SD figure 1.zip" file. Currently the Source Data files are not clearly labeled and e.g. Figure 1 and EV1 should not be in the same folder. SD

All source data has been re-organized and we're uploading the updated Source Data Checklist.

- 11) As part of the EMBO Publications transparent editorial process initiative (see our policy here: https://www.embopress.org/transparent-process#Review_Process), Molecular Systems Biology will publish online a Peer Review File (PRF) to accompany accepted manuscripts. This file will be published in conjunction with your paper and will include the anonymous referee reports, your point-by-point response and all pertinent correspondence relating to the manuscript. Let us know whether you agree with the publication of the PRF and as here, if you want to remove or not any figures from it prior to publication. Please note that the Authors checklist will be published at the end of the PRF.

We agree with the publication of the PRF as it is.

- 12) After your paper is published, we may promote it on social media. If you have any handles or hashtags for Bluesky you would like included, please let us know.

Yes. Our handles are the following:

@mariananatalin6.bsky.social

@marcofumasoni.bsky.social

Regarding hashtags, we would like to include: #CompensatoryEvolution #NutrientAvailability #ReplicationStress and #GxE

- 13) Please provide a point-by-point letter INCLUDING my comments as well as the reviewer's reports and your detailed responses (as Word file).

Reviewer #1:

All my previous concerns have been addressed. The manuscript looks great.

Reviewer #2:

The authors have addressed carefully most of my concerns. In particular, they produced new

experimental evidence for addressing the role of the compensatory mutation in MED14 (Figure 5); they have tested a mutant that express all the 'core genetic adaptation genes' (Figure 6); and have tested the interplay between Rnr1 and ixr1 compensatory mutation (Figure EV4). The manuscript is much improved. I have only some minor points:

- Figure EV4 shows that the compensatory effects of the inactivation of IXR1 is lost when a form of RNR1 refractory to feedback inhibition (rnr1-D57N) is overexpressed. The data also show that the simple overexpression of RNR1 does not affect the rescue (why is that?). These other data are shown but not discussed. Moreover, this Figure is only cited in the Discussion (i.e., does not appear in between Figure EV4A and EV5A). I think it could be discussed a bit more thoroughly in the right spot.

We share the same thoughts of the reviewer. We could speculate that the feedback inhibition of *RNR1* could prevent its overexpression to fully boost dNTPs production. However, we feel that adding a discussion of these aspects may distract the reader from the main message of the discussion.

- line 70, not clear to me what 'For example' refers to (as an example of what?).

We removed it as we agreed that it was ambiguous.

- lines 190-193 have become quite difficult to follow, I would suggest to explain better observation and conclusion (why the proportion of fitness effect recovered should remain constant in consequence of the varying initial fitness?).

We rephrased the sentence to improve clarity.

- line 350, 'with the exception of IXR1 deletion', but then in the figure we do not see the statistical significance.

The statistical analysis for the whole panel was present in source data EV5B. We now report the p-value relative to the *ixr1* deletion in *ctf4D* in the figure panel, as well as referring to the source data EV5B for the rest of the statistical analysis.

- Figure EV5A-B: maybe use the same scale on the two plots, to better appreciate the difference between WT and *ctf4*-null?

The plots of WT and *ctf4Δ* are now on the same scale.

- line 372, 'a frequent combination' somehow sounds strange to me (a combination of frequent adaptive mutations?).

We added the suggested clarification.

I also had a look at the comments of referee 4, and here are my thoughts:

I think the authors have convincingly addressed the concerns.

One major concern dealt with the lack of generality of the effect (i.e., whether other forms of replication stress may obtain the same effect). Here the authors argue, convincingly, that a

thorough analysis of other mutants that cause replication stress would be material for several papers.

A second major point was the mechanism by which the med14 mutations is advantageous. This was addressed with new experiments shown in Figure 5. The results do not explain the full molecular mechanism, but they suffice to exclude the hypothesis presented originally and to focus on the pleiotropic effects of the mutation.

The other major points raised by the reviewer lead to a study on the presence in wild strain of the med14 mutation (specific figure 5 for the reviewer not included in the paper); to specify some concepts in the text; and the heteroscedasticity in the data of Figure 1E. They have all been changed/addressed/commented.

To summarize, I think the only point that could still be debatable is the generality of the conclusion in terms of replication stress. The authors find that the genetic changes are largely independent on nutrient availability. Would that be true in the presence of other sources of replication stress? Reading the paper, one could predict that this would depend on the strength of the stress. However, a thorough investigation of other sources of stress would require to produce an equal amount of work for every source of stress, which I believe would not be suitable.

Reviewer #3:

I was happy to see the improvements the authors made in an already high-quality manuscript. I have no further revisions.

5th Jun 2025

Manuscript number: MSB-2025-12869RR

Title: Compensatory Evolution to DNA Replication Stress is Robust to Nutrient Availability

Dear Dr. Fumasoni,

Thank you again for sending us your revised manuscript. We are now satisfied with the modifications made and I am pleased to inform you that your paper has been accepted for publication.

Yours sincerely,

Poonam Bheda, PhD
Scientific Editor
Molecular Systems Biology
